# SAM 3: SEGMENT ANYTHING WITH *Concepts*

**Nicolas Carion**[*], **Laura Gustafson**[*], **Yuan-Ting Hu**[*], **Shoubhik Debnath**[*],
**Ronghang Hu**[*], **Didac Suris**[*], **Chaitanya Ryali**[*], **Kalyan Vasudev Alwala**[*],
**Haitham Khedr**[*], **Andrew Huang**, **Jie Lei**, **Tengyu Ma**, **Baishan Guo**, **Arpit Kalla**,
**Markus Marks**, **Joseph Greer**, **Meng Wang**, **Peize Sun**, **Roman Rädle**,
**Triantafyllos Afouras**, **Effrosyni Mavroudi**, **Katherine Xu**[°], **Tsung-Han Wu**[°], **Yu Zhou**[°],
**Liliane Momeni**[°], **Rishi Hazra**[°], **Shuangrui Ding**[°], **Sagar Vaze**[°], **Francois Porcher**[°],
**Feng Li**[°], **Siyuan Li**[°], **Aishwarya Kamath**[°], **Ho Kei Cheng**[°], **Piotr Dollár**[†],
**Nikhila Ravi**[†], **Kate Saenko**[†], **Pengchuan Zhang**[†], **Christoph Feichtenhofer**[†]

Meta Superintelligence Labs          `https://ai.meta.com/sam3`

## ABSTRACT

We present Segment Anything Model (SAM) 3, a unified model that detects, segments, and tracks objects in images and videos based on *concept prompts*, which we define as either short noun phrases (e.g., "yellow school bus"), image exemplars, or a combination of both. Promptable Concept Segmentation (PCS) takes such prompts and returns segmentation masks and unique identities for all matching object instances. To advance PCS, we build a scalable data engine that produces a high-quality dataset with 4M unique concept labels, including hard negatives, across images and videos. Our model consists of an image-level detector and a memory-based video tracker that share a single backbone. Recognition and localization are decoupled with a presence head, which boosts detection accuracy. SAM 3 *doubles the accuracy* of existing systems in both image and video PCS, and improves previous SAM capabilities on visual segmentation tasks. We open source SAM 3 along with our new Segment Anything with Concepts (SA-Co) benchmark for promptable concept segmentation.

## 1 INTRODUCTION

The ability to find and segment *anything* in a visual scene is foundational for multimodal AI, powering applications in robotics, content creation, augmented reality, data annotation, and broader sciences. The SAM series (Kirillov et al., 2023; Ravi et al., 2024) introduced the promptable segmentation task for images and videos, focusing on *Promptable Visual Segmentation* (PVS) with points, boxes or masks to segment a single object per prompt. While these methods achieved a breakthrough, they did not address the general task of finding and segmenting *all* instances of a concept appearing *anywhere* in the input (e.g., all "cats" in a video).

[*]core contributor, [°]intern, [†]project lead, order is random within groups

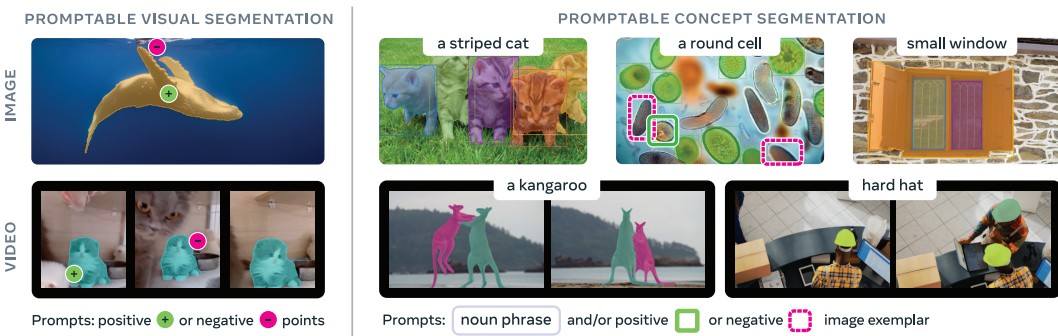

Figure 1: SAM 3 improves over SAM 2 on promptable *visual* segmentation with clicks (left) and introduces the new promptable *concept* segmentation capability (right). Users can segment all instances of a visual concept specified by a short noun phrase, image exemplars (positive or negative), or a combination of both.

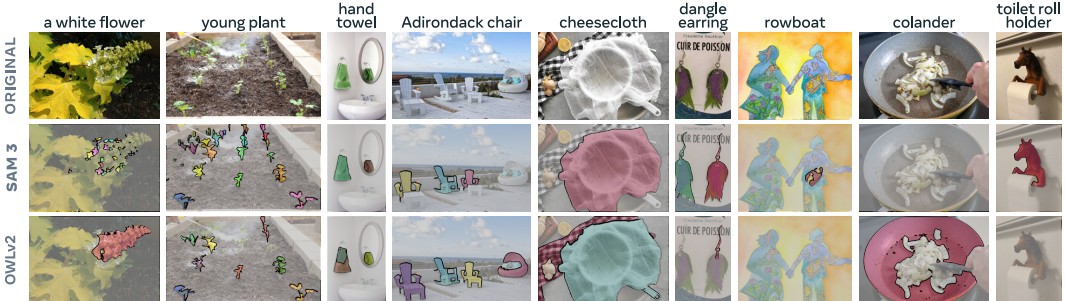

Figure 2: Examples of SAM 3 improving segmentation of open-vocabulary concepts compared to OWLv2 (Minderer et al., 2024), on the SA-Co benchmark. See §G.6.1 for additional SAM 3 outputs.

To fill this gap, we present SAM 3, a model that achieves a step change in promptable segmentation in images and videos, improving PVS relative to SAM 2 and setting a new standard for *Promptable Concept Segmentation (PCS)*. We formalize the PCS task (§2) as taking text and/or image exemplars as input, and predicting instance and semantic masks for every single object matching the concept, while preserving object identities across video frames (see Fig. 1). To focus on recognizing atomic visual concepts, we constrain text to simple noun phrases (NPs) such as "red apple" or "striped cat". While SAM 3 is not designed for long referring expressions or queries requiring reasoning, we show that it can be straightforwardly combined with a Multimodal Large Language Model (MLLM) to handle more complex language prompts. Consistent with previous SAM versions, SAM 3 is fully interactive, allowing users to resolve ambiguities by adding refinement prompts to guide the model towards their intended output.

Our *model* (§3) consists of a detector and a tracker that share a vision encoder (Bolya et al., 2025). The detector is a DETR-based (Carion et al., 2020) model conditioned on text, geometry, and image exemplars. To address the challenge of open-vocabulary concept detection, we introduce a separate *presence head* to decouple recognition and localization, which is especially effective when training with challenging *negative phrases*. The tracker inherits the SAM 2 transformer encoder-decoder architecture, supporting video segmentation and interactive refinement. The decoupled design for detection and tracking avoids task conflict, as the detector needs to be identity agnostic, while the tracker's main objective is to separate identities in the video.

To unlock major performance gains, we build a human- and model-in-the-loop *data engine* (§4) that annotates a large and diverse training dataset. We innovate upon prior data engines in three key ways: (i) *media curation*: we curate more diverse media domains than past approaches that rely on homogeneous web sources, (ii) *label curation*: we significantly increase label diversity and difficulty by leveraging an ontology and multimodal LLMs as "AI annotators" to generate noun phrases and hard negatives, (iii) *label verification*: we double annotation throughput by fine-tuning MLLMs to be effective "AI verifiers" that achieve near-human accuracy.

Starting from noisy media-phrase-mask pseudo-labels, our data engine checks mask quality and exhaustivity using both human and AI verifiers, filtering out correctly labeled examples and identifying challenging error cases. Human annotators then focus on fixing these errors by manually correcting masks. This enables us to annotate high-quality training data with 4M *unique* phrases and 52M masks, and a synthetic dataset with 38M phrases and 1.4B masks. We additionally create the Segment Anything with Concepts (SA-Co) *benchmark* for PCS (§5) containing 207K unique concepts with exhaustive masks in 120K images and 1.7K videos, $> 50\times$ more concepts than existing benchmarks.

Our *experiments* (§6) show that SAM 3 sets a new state-of-the-art in promptable segmentation, e.g., reaching a zero-shot mask AP of 48.8 on LVIS *vs.* the current best of 38.5, surpassing baselines on our new SA-Co benchmark by at least $2\times$ (see examples in Fig. 2), and improving upon SAM 2 on visual prompts. Ablations (§B) verify that the choice of backbone, novel presence head, and adding hard negatives all boost results, and establish scaling laws on the PCS task for both our high-quality and synthetic datasets. We open-source the SA-Co benchmark and release the SAM 3 checkpoints and inference code. On an H200 GPU, SAM 3 runs in 30 ms for a single image with 100+ detected objects. In video, the inference latency scales with the number of objects, sustaining near real-time performance for $\sim 5$ concurrent objects. We review related work in §A; next, we dive into the task.

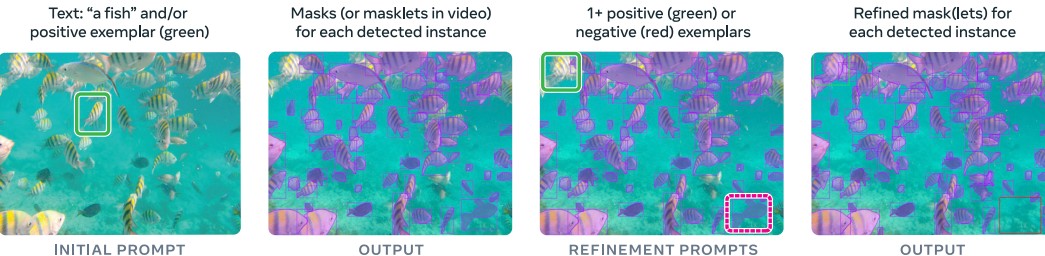

Figure 3: Illustration of supported initial and optional interactive refinement prompts in the PCS task.

## 2 PROMPTABLE CONCEPT SEGMENTATION (PCS)

We define the Promptable Concept Segmentation task as follows: given an image or short video ($\leq$30 secs), detect, segment and track all instances of a visual concept specified by a short text phrase, image exemplars, or a combination of both. We restrict concepts to those defined by simple noun phrases (NPs) consisting of a noun and optional modifiers. Noun-phrase prompts (when provided) are *global* to all frames of the image/video, while image exemplars can be provided on *individual* frames as positive or negative bounding boxes to iteratively *refine* the target masks (see Fig. 3).

All prompts must be consistent in their category definition, or the model's behavior is undefined; e.g., "fish" cannot be refined with subsequent exemplar prompts of just the tail; instead the text prompt should be updated. Exemplar prompts are particularly useful when the model initially misses some instances, or when the concept is rare.

Our vocabulary includes any simple noun phrase groundable in a visual scene, which makes the task intrinsically ambiguous. There can be multiple interpretations of phrases arising from polysemy ("mouse" device *vs.* animal), subjective descriptors ("cozy", "large"), vague or context-dependent phrases that may not even be groundable ("brand identity"), boundary ambiguity (whether 'mirror' includes the frame) and factors such as occlusion and blur that obscure the extent of the object. While similar issues appear in large closed-vocabulary corpora (e.g., LVIS (Gupta et al., 2019)), they are alleviated by carefully curating the vocabulary and setting a clear definition of all the classes of interest. We address the ambiguity problem by collecting test annotations from three experts, adapting the evaluation protocol to allow multiple valid interpretations (§F.3), designing the data pipeline/guidelines to minimize ambiguity in annotation, and an ambiguity module (§D.2).

## 3 MODEL

SAM 3 is a generalization of SAM 2, supporting the new PCS task (§2) as well as the PVS task. It takes *concept* prompts (simple noun phrases, image exemplars) or *visual* prompts (points, boxes, masks) to define the *objects* to be (individually) segmented spatio-temporally. Image exemplars and visual prompts can be *iteratively* added on individual frames to *refine* the target masks—false positive and false negative objects can be *removed* or *added* respectively using image exemplars and an *individual* mask(let) can be refined using PVS in the style of SAM 2. Our architecture is broadly based on the SAM and (M)DETR (Carion et al., 2020; Kamath et al., 2021) series. Fig. 4 shows the SAM 3 architecture, consisting of a dual encoder-decoder transformer—a *detector* for image-level capabilities—which is used in combination with a *tracker* and memory for video. The detector and tracker ingest vision-language inputs from an aligned Perception Encoder (PE) backbone (Bolya et al., 2025). We present an overview below, see §D for details.

**Detector Architecture.** The architecture of the detector follows the general DETR paradigm. The image and text prompt are first encoded by PE and image exemplars, if present, are encoded by an exemplar encoder. We refer to the image exemplar tokens and text tokens jointly as "prompt tokens". The fusion encoder then accepts the unconditioned embeddings from the image encoder and conditions them by cross-attending to the prompt tokens. The fusion is followed by a DETR-like decoder, where learned object queries cross-attend to the conditioned image embeddings from the fusion encoder.

Each decoder layer predicts a classification logit for each object query (in our case, a binary label of whether the object corresponds to the prompt), and a delta from the bounding box predicted by the previous level, following Zhu et al. (2020). We use box-region-positional bias (Lin et al., 2023)

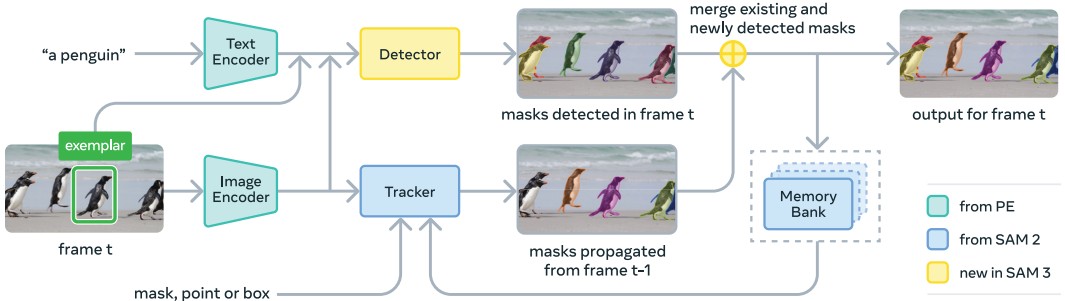

Figure 4: SAM 3 architecture overview. See Fig. 10 for a more detailed diagram.

to help focalize the attention on each object, but unlike recent DETR models, we stick to vanilla attention. During training, we adopt dual supervision from DAC-DETR (Hu et al., 2023), and the Align loss (Cai et al., 2024). The mask head is adapted from MaskFormer (Cheng et al., 2021). In addition, we also have a semantic segmentation head, which predicts a binary label for every pixel in the image, indicating whether or not it corresponds to the prompt. See §D for details.

**Presence Token.** It can be difficult for each of the proposal queries to both recognize (what) and localize (where) an object in the image/frame. For the recognition component, contextual cues from the entire image are important. However, forcing proposal queries to understand the global context can be counterproductive, as it conflicts with the inherently local nature of the localization objective. We decouple the recognition and localization steps by introducing a learned global *presence token*. This token is solely responsible for predicting whether the target concept in the form of a noun phrase (NP) is present in the image/frame, i.e. $p(\text{NP is present in input})$. Each proposal query $q_i$ only needs to solve the localization problem $p(q_i \text{ is a match} \mid \text{NP is present in input})$. The final score for each proposal query is the product of its own score and the presence score.

**Image Exemplars and Interactivity.** SAM 3 supports image exemplars, given as a pair—a bounding box and an associated binary label (positive or negative)—which can be used in isolation or to supplement the text prompt. The model then detects all the instances that match the prompt. For example, given a positive bounding box on a dog, the model will detect *all* dogs in the image. This is different from the PVS task in SAM 1 and 2, where a visual prompt yields only a single object instance. Each image exemplar is encoded separately by the exemplar encoder using an embedding for the position, an embedding for the label, and ROI-pooled visual features, then concatenated and processed by a small transformer. The resulting prompt is concatenated to the text prompt to comprise the prompt tokens. Image exemplars can be *interactively* provided based on errors in current detections to refine the output.

**Tracker and Video Architecture.** Given a video and a prompt $P$, we use the detector and a tracker (see Fig. 4) to detect and track objects corresponding to the prompt throughout the video. On each frame, the detector finds new objects $\mathcal{O}_t$ and the tracker propagates masklets $\mathcal{M}_{t-1}$ (spatial-temporal masks) from frames at the previous time $t-1$ to their new locations $\hat{\mathcal{M}}_t$ on the current frame at time $t$. We use a matching function to associate propagated masklets $\hat{\mathcal{M}}_t$ with new object masks emerging in the current frame $\mathcal{O}_t$,

$$\hat{\mathcal{M}}_t = \text{propagate}\left(\mathcal{M}_{t-1}\right), \quad \mathcal{O}_t = \text{detect}\left(I_t, P\right), \quad \mathcal{M}_t = \text{match\_and\_update}\left(\hat{\mathcal{M}}_t, \mathcal{O}_t\right).$$

**Tracking an Object with SAM 2 Style Propagation.** A masklet is initialized for every object detected on the first frame. Then, on each subsequent frame, the tracker module predicts the new masklet locations $\hat{\mathcal{M}}_t$ of those already-tracked objects based on their previous locations $\mathcal{M}_{t-1}$ through a single-frame propagation step similar to the video object segmentation task in SAM 2. The tracker shares the same image/frame encoder (PE backbone) as the detector. After training the detector, we freeze PE and train the tracker as in SAM 2, including a prompt encoder, mask decoder, memory encoder, and a memory bank that encodes the object's appearance using features from the past frames and conditioning frames (frames where the object is first detected or user-prompted). The memory encoder is a transformer with self-attention across visual features on the current frame and cross-attention from the visual features to the spatial memory features in the memory bank. We describe details of our video approach in §D.3.

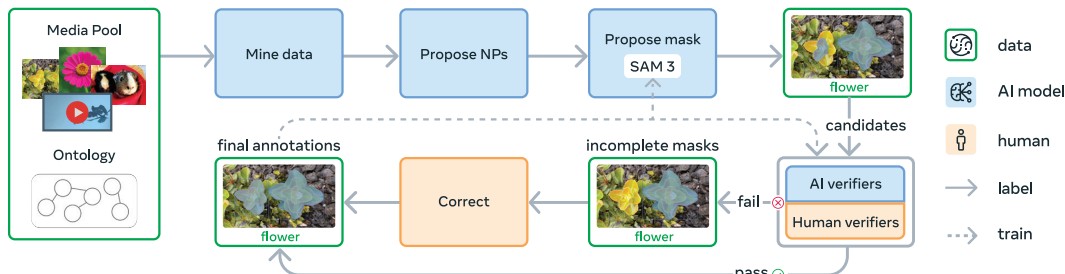

Figure 5: Overview of the final SAM 3 data engine. See §F.1 for details of collected data.

During inference, we only retain frames where the object is confidently present in the memory bank. The mask decoder is a two-way transformer between the encoder hidden states and the output tokens. To handle ambiguity, we predict three output masks for every tracked object on each frame along with their confidence, and select the most confident output as the predicted mask on the current frame.

**Matching and Updating Based on Detections.** After obtaining the tracked masks $\hat{\mathcal{M}}_t$, we match them with the current frame detections $\mathcal{O}_t$ through a simple IoU based *matching function* (§D.3) and add them to $\mathcal{M}_t$ on the current frame. We further spawn new masklets for all newly detected objects that are not matched. The merging might suffer from ambiguities, especially in crowded scenes. We address this with two temporal disambiguation strategies outlined next.

First, we use temporal information in the form of a *masklet detection score* (§D.3) to measure how consistently a masklet is matched to a detection within a temporal window (based on the number of past frames where it was matched to a detection). If a masklet's detection score falls below a threshold, we suppress it. Second, we use the detector outputs to resolve specific failure modes of the tracker due to occlusions or distractors. We periodically *re-prompt* the tracker with high-confidence *detection* masks $\mathcal{O}_t$, replacing the tracker's own predictions $\hat{\mathcal{M}}_t$. This ensures that the memory bank has recent and reliable references (other than the tracker's own predictions).

**Instance Refinement with Visual Prompts.** After obtaining the initial set of masks (or masklets), SAM 3 allows refining individual masks(lets) using positive and negative clicks. Specifically, given the user clicks, we apply the prompt encoder to encode them, and feed the encoded prompt into the mask decoder to predict an adjusted mask. In videos the mask is then propagated across the entire video to obtain a refined masklet.

**Training Stages.** We train SAM 3 in four stages that progressively add data and capabilities: 1) Perception Encoder (PE) pre-training, 2) detector pre-training, 3) detector fine-tuning, and 4) tracker training with a frozen backbone. See §D.4.1 for details.

## 4 DATA ENGINE

Achieving a step change in PCS with SAM 3 requires training on a large, diverse set of concepts and visual domains, beyond existing datasets (see Fig. 12). We build an efficient data engine that iteratively generates annotated data via a feedback loop with SAM 3, human annotators, and *AI annotators*, actively mining media-phrase pairs on which the current version of SAM 3 fails to produce high-quality training data to further improve the model. By delegating certain tasks to AI annotators—models that match or surpass human accuracy—we more than double the throughput compared to a human-only annotation pipeline. We develop the data engine in four phases, with each phase increasing the use of AI models to steer human effort to the most challenging failure cases, alongside expanding visual domain coverage. Phases 1-3 focus only on images, with Phase 4 expanding to videos. We describe the key steps here; details and metrics are in §E.

**Data Engine Components (Fig. 5).** Media inputs (image or video) are mined from a large pool with the help of a curated ontology. An AI model proposes noun phrases (NPs) describing visual concepts, followed by another model (e.g., SAM 3) that generates candidate instance masks for each proposed NP. The proposed masks are verified by a two-step process: first, in *Mask Verification (MV)* annotators accept or reject masks based on their quality and relevance to the NP. Second, in *Exhaustivity Verification (EV)* annotators check if all instances of the NP have been masked in the input. Any media-NP pairs that did not pass the exhaustivity check are sent to a manual correction

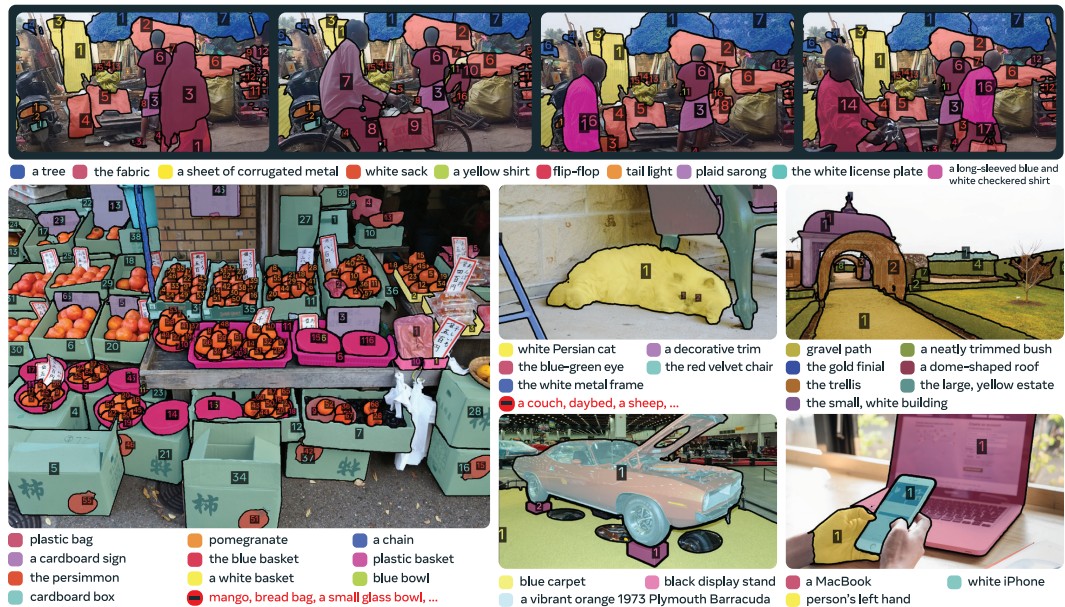

Figure 6: Example video (top) and images (bottom) from SA-Co with annotated phrases and instance masks/IDs.

stage, where humans add, remove or edit masks (using SAM 1 in a browser based tool), or use "group" masks for small, hard to separate objects. Annotators may reject ungroundable or ambiguous phrases.

**Phase 1: Human Verification.** We first randomly sample images and NP proposal with a simple captioner and parser. The initial mask proposal model is SAM 2 prompted with the output of an off-the-shelf open-vocabulary detector, and initial verifiers are human. In this phase, we collected 4.3M image-NP pairs as the initial SA-Co/HQ dataset. We train SAM 3 on this data and use it as the mask proposal model for the next phase.

**Phase 2: Human + AI Verification.** In this next phase, we use human accept/reject labels from the MV and EV tasks collected in Phase 1 to fine-tune Llama 3.2 (Dubey et al., 2024) to create AI verifiers that *automatically* perform the MV and EV tasks. These models receive image-phrase-mask triplets and output multiple-choice ratings of mask quality or exhaustivity. This new auto-verification process allows our human effort to be focused on the most challenging cases. We continue to re-train SAM 3 on newly collected data and update it 6 times. As SAM 3 and AI verifiers improve, a higher proportion of labels are auto-generated, further accelerating data collection. The introduction of AI verifiers for MV and EV roughly doubles the data engine's throughput *vs.* human annotators. We refer to §B.4 for detailed analysis of how AI verifiers improve the data engine's throughput. We further upgrade the NP proposal step to a Llama-based pipeline that also proposes hard negative NPs adversarial to SAM 3. Phase 2 adds 122M image-NP pairs to SA-Co/HQ.

**Phase 3: Scaling and Domain Expansion.** In the third phase, we use AI models to mine increasingly challenging cases and broaden domain coverage in SA-Co/HQ to 15 datasets (Fig. 15). A *domain* is a unique distribution of text and visual data. In new domains, the MV AI verifier performs well zero-shot, but the EV AI verifier needs to be improved with modest domain-specific human supervision. We also expand concept coverage to long-tail, fine-grained concepts by extracting NPs from the image alt-text where available and by mining concepts from a 22.4M node *SA-Co ontology* (§E.2) based on Wikidata (17 top-level categories, 72 sub-categories). We iterate SAM 3 training 7 times and AI verifiers 3 times, and add 19.5M image-NP pairs to SA-Co/HQ.

**Phase 4: Video Annotation.** This phase extends the data engine to video. We use a mature image SAM 3 to collect targeted quality annotations that capture video-specific challenges. The data mining pipeline applies scene/motion filters, content balancing, ranking, and targeted searches. Video frames are sampled (randomly or by object density) and sent to the image annotation flow (from phase 3). *Masklets* (spatio-temporal masks) are produced with SAM 3 (now extended to video) and post-processed via deduplication and removal of trivial masks. Because video annotation is more difficult, we concentrate humans on likely failures by favoring clips with many crowded objects and tracking failures. The collected video data SA-Co/VIDEO consists of 52.5K videos and 467K masklets. See §E.6 for details.

| Model | Instance Segmentation | | | | | | Box Detection | | | | | | | | Semantic Segmentation | | |
|---|---|---|---|---|---|---|---|---|---|---|---|---|---|---|---|---|---|
| | LVIS | | SA-Co | | | | LVIS | | COCO | | SA-Co | | | | ADE-847 | PC-59 | Cityscapes |
| | $cgF_1$ | AP | Gold $cgF_1$ | Silver $cgF_1$ | Bronze $cgF_1$ | Bio $pmF_1$ | $cgF_1$ | AP | AP | $AP_o$ | Gold $cgF_1$ | Silver $cgF_1$ | Bronze $cgF_1$ | Bio $pmF_1$ | mIoU | mIoU | mIoU |
| Human | – | – | 72.8 | – | – | – | – | – | – | – | 74.0 | – | – | – | – | – | – |
| OWLv2 | 20.1 | – | 17.3 | 7.6 | 3.9 | 0.64 | 19.9 | 35.2 | 38.2 | 42.4 | 16.9 | 7.1 | 4.1 | 0.95 | – | – | – |
| OWLv2$^\star$ | 29.3 | 43.4 | 24.6 | 11.5 | 11.7 | 0.04 | 30.2 | 45.5 | 46.1 | 23.9 | 24.5 | 11.0 | 12.0 | 0.08 | – | – | – |
| gDino-T | 14.7 | – | 3.3 | 2.7 | 7.0 | 0.34 | 15.1 | 20.5 | 45.7 | 35.3 | 3.4 | 2.5 | 7.6 | 0.35 | – | – | – |
| LLMDet-L | 35.1 | 36.3 | 6.5 | 7.1 | 12.5 | 0.15 | 39.3 | 42.0 | 55.6 | 49.8 | 6.8 | 6.7 | 14.0 | 0.17 | – | – | – |
| APE-D$^\star$ | – | 53.0$^\dagger$ | 16.4 | 7.3 | 12.4 | 0.00 | – | 59.6$^\dagger$ | 58.3$^\dagger$ | – | 17.3 | 7.7 | 14.3 | 0.00 | 9.2$^\dagger$ | 58.5$^\dagger$ | 44.2$^\dagger$ |
| DINO-X | – | 38.5$^\dagger$ | 21.3$^\delta$ | – | – | – | – | 52.4$^\dagger$ | 56.0$^\dagger$ | – | 22.5$^\delta$ | – | – | – | – | – | – |
| Gemini 2.5 | 13.4 | – | 13.0 | 8.3 | 7.3 | 10.7 | 16.1 | – | – | – | 14.4 | 9.4 | 8.2 | 12.4 | – | – | – |
| **SAM 3** | **37.2** | **48.5** | **54.1** | **49.6** | **42.6** | **55.4** | **40.6** | **53.6** | **56.4** | **55.7** | **55.7** | **50.0** | **47.1** | **56.3** | **13.8** | **60.8** | **65.2** |

Table 1: Evaluation on image concept segmentation with text. $AP_o$ corresponds to COCO-O accuracy, $\star$: partially trained on LVIS, $\dagger$: from original papers, $\delta$: from DINO-X API. Gray numbers indicate usage of respective closed set training data (LVIS/COCO). See §G.1 for more results and §F.4 for details of human performance.

# 5  Segment Anything with Concepts (SA-Co) Dataset

**Training Data.** We collect three *image datasets* for the PCS task: (i) SA-Co/HQ, the high-quality image data collected from the data engine in phases 1-4, (ii) SA-Co/SYN, a synthetic dataset of images labeled by a mature data engine (phase 3) without human involvement, and (iii) SA-Co/EXT, 15 external datasets that have instance mask annotations, enriched with hard negatives using our ontology pipeline. Notably in the SA-Co/HQ dataset we annotate 5.2M images and 4M unique NPs, making it the largest high-quality open-vocab segmentation dataset. We also annotate a *video dataset*, SA-Co/VIDEO, containing 52.5K videos and 24.8K unique NPs, forming 134K video-NP pairs. The videos on average have 84.1 frames at 6 fps. See §F.1 for details including full statistics, comparison with existing datasets and the distribution of concepts.

**SA-Co Benchmark.** The SA-Co evaluation benchmark has 207K unique phrases, 121K images and videos, and over 3M media-phrase pairs with hard negative labels to test open-vocabulary recognition. It has 4 splits: SA-Co/Gold has seven domains and each image-NP pair is annotated by three different annotators (used to measure human performance); SA-Co/Silver has ten domains and only one human annotation per image-NP pair; SA-Co/Bronze and SA-Co/Bio are nine existing datasets either with existing mask annotations or masks generated by using boxes as prompts to SAM 2. The SA-Co/VEval benchmark has three domains and one annotator per video-NP pair. See Tab. 28 for dataset statistics and Fig. 6 for example annotations.

**Metrics.** We aim to measure the usefulness of the model in downstream applications. Detection metrics such as average precision (AP) do not account for calibration, which means that models can be difficult to use in practice. To remedy this, we only evaluate predictions with confidence above 0.5, effectively introducing a threshold that mimics downstream usages and enforces good calibration. The PCS task can be naturally split into two sub-tasks, *localization* and *classification*. We evaluate localization using *positive micro F1* ($pmF_1$) on positive media-phrase pairs with at least one ground-truth mask. Classification is measured with *image-level Matthews Correlation Coefficient* (IL_MCC) which ranges in $[-1, 1]$ and evaluates binary prediction at the image level ("is the object present?") without regard for mask quality. Our main metric, *classification-gated F1* ($cgF_1$), combines these as follows: $cgF_1 = 100 * pmF_1 * \text{IL\_MCC}$. Full definitions are in §F.3.

**Handling Ambiguity.** We collect 3 annotations per NP on SA-Co/Gold. We measure *oracle* accuracy comparing each prediction to all ground truths and selecting the best score. See §F.3.

# 6  Experiments

We evaluate SAM 3 across image and video segmentation, few-shot adaptation to detection and counting benchmarks, and segmentation with complex language queries with SAM 3 + MLLM. We also show a subset of ablations, with more in §B. References, more results and details are in §G.

**Image PCS with Text.** We evaluate instance segmentation, box detection, and semantic segmentation on external and our benchmarks. SAM 3 is prompted with a single NP at a time, and predicts instance masks, bounding boxes, or semantic masks. As baselines, we evaluate OWLv2, GroundingDino (gDino), and LLMDet on box detection, and prompt SAM 1 with their boxes to evaluate segmentation.

| Model | ODinW13 AP$_0$ | ODinW13 AP$_{10}$ | RF-100VL AP$_0$ | RF-100VL AP$_{10}$ |
|---|---|---|---|---|
| Gemini2.5-Pro | 33.7 | – | 11.6 | 9.8 |
| gDino-T | 49.7 | – | **15.7** | 33.7 |
| gDino1.5-Pro | 58.7 | 67.9 | – | – |
| **SAM 3** | **61.0** | **71.8** | 15.2 | **36.5** |

Table 2: Zero-shot and 10-shot transfer on in-the-wild datasets.

| Model | COCO AP T | COCO AP$^+$ T | COCO AP$^+$ I | COCO AP$^+$ T+I | LVIS AP T | LVIS AP$^+$ T | LVIS AP$^+$ I | LVIS AP$^+$ T+I | ODinW13 AP T | ODinW13 AP$^+$ T | ODinW13 AP$^+$ I | ODinW13 AP$^+$ T+I |
|---|---|---|---|---|---|---|---|---|---|---|---|---|
| T-Rex2 | 52.2 | – | 58.5 | – | 45.8 | – | 65.8 | – | 50.3 | – | 61.8 | – |
| **SAM 3** | **56.4** | **58.8** | **76.8** | **78.1** | **52.4** | **54.7** | **76.0** | **78.4** | **61.1** | **63.1** | **82.2** | **81.8** |

Table 3: Prompting with 1 exemplar on COCO, LVIS and ODinW13. Evaluation per prompt type: T (text-only), I (image-only), and T+I (combined text and image). AP$^+$ is evaluated only on positive examples.

We also compare to APE, DINO-X, and Gemini 2.5 Flash, a generalist LLM. Tab. 1 shows that zero-shot, SAM 3 sets a new state-of-the-art on closed-vocabulary COCO, COCO-O and on LVIS boxes, and is significantly better on LVIS masks. On open-vocabulary SA-Co/Gold SAM 3 achieves *more than double* the cgF$_1$ score of the strongest baseline OWLv2$^\star$, and 74% of the estimated human performance. The improvements are even higher on the other SA-Co splits. Open vocabulary semantic segmentation results on ADE-847, PascalConcept-59, and Cityscapes (val set) show that SAM 3 outperforms APE, a strong specialist baseline. See §G.1 for details.

**Few-Shot Adaptation.** We evaluate zero- and few-shot transfer of SAM 3 on ODinW13 and RF100-VL, with their original labels as prompts. We *do not* perform any prompt tuning. We fine-tune SAM 3 without mask loss, and report average bbox mAP in Tab. 2. SAM 3 achieves state-of-the-art 10-shot performance, surpassing in-context prompting in Gemini and object detection experts (gDino); more details in §G.3. RF-100VL contains domains with specialized prompts that are out of SAM 3's current scope, but SAM 3 adapts through fine-tuning more efficiently than baselines.

**PCS with 1 Exemplar.** We first evaluate image exemplars using a single input box sampled at random from the ground truth. This can be done only on "*positive*" data, where each prompted object appears in the image. We report the corresponding AP$^+$ in Tab. 3 across three settings: text prompt (T), exemplar image (I), and both text and image (T+I); SAM 3 outperforms prior state-of-the-art T-Rex2 by a healthy margin on COCO (+18.3), LVIS (+10.3), and ODinW (+20.5). See §G.2 for more details and results on SA-Co/Gold.

**PCS with K Exemplars.** Next, we evaluate SAM 3 in an interactive setting, simulating collaboration with a human annotator. Starting with a text prompt, we iteratively add one exemplar prompt at a time: missed ground truths are candidate positive prompts, false positive detections are candidate negative prompts. Results (Fig. 7) are compared to a perfect PVS baseline, where we simulate the user manually fixing errors using ideal box-to-mask corrections. SAM 3's PCS improves cgF$_1$ more quickly, as it generalizes from exemplars (e.g., detecting or suppressing similar objects), while PVS only corrects individual instances. After 3 clicks, interactive PCS outperforms text-only by +21.6 cgF$_1$ points and PVS refinement by +2.0. Performance plateaus after 4 clicks, as exemplars cannot fix poor-quality masks. Simulating a *hybrid* switch to PVS at this point yields gains, showing *complementary*.

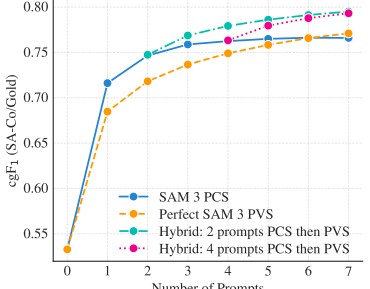

Figure 7: cgF$_1$ *vs.* # of interactive box prompts for SAM 3 compared to the ideal PVS baseline, averaged over SA-Co/Gold phrases.

**Object Counting.** We evaluate on object counting benchmarks CountBench and PixMo-Count to compare with several MLLMs using Accuracy (%) and Mean Absolute Error (MAE) from previous technical reports and our own evaluations. See Tab. 4 for results and §G.4 for more evaluation details. Compared to MLLMs, SAM 3 not only achieves good object counting accuracy, but also provides object segmentation that most MLLMs cannot provide.

| Model | CountBench MAE ↓ | CountBench Acc ↑ | PixMo-Count MAE ↓ | PixMo-Count Acc ↑ |
|---|---|---|---|---|
| DINO-X | 0.62 | 82.9 | **0.21** | 85.0 |
| Qwen2-VL-72B | 0.28 | 86.7 | 0.61 | 63.7 |
| Molmo-72B | 0.27 | 92.4 | 0.17 | 88.8 |
| Gemini 2.5 Pro | 0.24 | 92.4 | 0.38 | 78.2 |
| **SAM 3** | **0.12** | **93.8** | 0.21 | **86.2** |

Table 4: Accuracy on counting benchmarks. Gray indicates usage of training sets.

**Video PCS with Text.** We evaluate video segmentation with text prompts on both our SA-Co/VEval benchmark and existing public benchmarks. For SA-Co/VEval, we report cgF$_1$ and pHOTA metrics (defined in §G.5) across its subsets (SA-V, YT-Temporal-1B, SmartGlasses). For public benchmarks, we use their official metrics. Baselines include GLEE, an open-vocabulary image and video segmentation model, "LLMDet + SAM 3 Tracker" (replacing our detector with LLMDet), and "SAM

| | SA-Co/VEval benchmark test split | | | | | | Public benchmarks | | | |
| | **SA-V** (2.0K NPs) | | **YT-Temporal-1B** (1.7K NPs) | | **SmartGlasses** (2.4K NPs) | | **LVVIS** (1.2K NPs) | **BURST** (482 NPs) | **YTVIS21** (40 NPs) | **OVIS** (25 NPs) |
| Model | cgF$_1$ | pHOTA | cgF$_1$ | pHOTA | cgF$_1$ | pHOTA | test mAP | test HOTA | val mAP | val mAP |
|---|---|---|---|---|---|---|---|---|---|---|
| Human | 53.1 | 70.5 | 71.2 | 78.4 | 58.5 | 72.3 | – | – | – | – |
| GLEE[†] (all NPs at once) | 0.1 | 8.7 | 1.6 | 16.7 | 0.0 | 4.7 | 20.8 | 28.4 | **62.2** | 38.7 |
| GLEE[†] (one NP at a time) | 0.1 | 11.8 | 2.2 | 18.9 | 0.1 | 5.6 | 9.3 | 20.2 | 56.5 | 32.4 |
| LLMDet[†] + **SAM 3** Tracker | 2.3 | 30.1 | 8.0 | 37.9 | 0.3 | 18.6 | 15.2 | 33.3 | 31.3 | 20.4 |
| **SAM 3** Detector + T-by-D | 25.7 | 55.7 | 47.6 | 68.2 | 29.7 | 60.0 | 35.9 | 39.7 | 56.5 | 55.1 |
| **SAM 3** | **30.3** | **58.0** | **50.8** | **69.9** | **36.4** | **63.6** | **36.3** | **44.5** | 57.4 | **60.5** |

Table 5: Video PCS from a text prompt (open-vocabulary video instance segmentation) on SA-Co/VEval and public benchmarks (see Tab. 39 for more results and analyses). SAM 3 shows strong performance, especially on benchmarks with a large number of NPs. †: GLEE and LLMDet do not perform well zero-shot on SA-Co/VEval.

3 Detector + T-by-D" (replacing our tracker with an association module based on the tracking-by-detection paradigm). In Tab. 5, SAM 3 largely outperforms these baselines, especially on benchmarks with a very large number of noun phrases. On SA-Co/VEval it reaches over 80% of human pHOTA. See §G.5 for more details.

**PVS.** We evaluate SAM 3 on a range of visual prompting tasks, including Video Object Segmentation (VOS) and interactive image segmentation. Tab. 6 compares SAM 3 to recent state-of-the-art methods on the VOS task. SAM 3 achieves significant improvements over SAM 2 on most benchmarks, particularly on the challenging MOSEv2 dataset, where SAM 3 outperforms prior work by 6.5 points. For the interactive image segmentation task, we evaluate SAM 3 on the 37 datasets benchmark introduced in Ravi et al. (2024). As shown in Tab. 7, SAM 3 outperforms SAM 2 on average mIoU. See also §G.6 and Fig. 21 for interactive video segmentation.

| | $\mathcal{J}\&\mathcal{F}$ | | | | | $\mathcal{G}$ | $\mathcal{J}\&\dot{\mathcal{F}}$ |
| | **MOSEv1** | **DAVIS17** | **LVOSv2** | **SA-V** | **SA-V** | **YTVOS19** | **MOSEv2** |
| Model | val | val | val | val | test | val | val |
|---|---|---|---|---|---|---|---|
| SAMURAI | 72.6 | 89.9 | 84.2 | 79.8 | 80.0 | 88.3 | 51.1 |
| SAM2Long | 75.2 | 91.4 | 85.9 | 81.1 | 81.2 | 88.7 | 51.5 |
| SeC | 75.3 | 91.3 | 86.5 | 82.7 | 81.7 | 88.6 | 53.8 |
| SAM 2.1 L | 77.9 | 90.7 | 79.6 | 77.9 | 78.4 | 89.3 | 47.9[†] |
| **SAM 3** | **78.4** | **92.2** | **88.5** | **83.5** | **84.4** | **89.7** | **60.3** |

Table 6: SAM 3 improves over SAM 2 in VOS. †: Zero-shot.

| | Avg. mIoU | | | |
| Model | 1-click | 3-clicks | 5-clicks | FPS |
|---|---|---|---|---|
| SAM 1 H | 58.5 | 77.0 | 82.1 | 41.0 |
| SAM 2.1 L | **66.4** | 80.3 | 84.3 | **93.0** |
| **SAM 3** | 66.1 | **81.3** | **85.1** | 43.5 |

Table 7: Interactive image segmentation on the SA-37 benchmark.

**SAM 3 Agent.** We experiment with an MLLM that uses SAM 3 as a tool to segment more complex text queries (see Fig. 25). The MLLM proposes noun phrase queries to prompt SAM 3 and analyzes the returned masks, iterating until the masks are satisfactory. Tab. 8 shows that this "SAM 3 Agent" evaluated zero-shot on ReasonSeg and OmniLabel surpasses prior work without training on any referring expression segmentation or reasoning segmentation data. SAM 3 Agent also outperforms previous zero-shot results on RefCOCO+ and RefCOCOg. SAM 3 can be combined with various MLLMs, with the same set of the system prompts for all those MLLMs, showing SAM 3's robustness. See §H for more details.

| | | ReasonSeg (gIoU) | | | | Omnilabel (AP) | | | |
| | | val | test | | | val 2023 | | | |
| Model | MLLM | All | All | Short | Long | descr | descr-S | descr-M | descr-L |
|---|---|---|---|---|---|---|---|---|---|
| X-SAM | Phi-3-3.8B | 56.6 | 57.8 | 47.7 | 56.0 | 12.0* | 17.1* | 11.4* | 8.8* |
| SegZero | Qwen2.5-VL 7B | 62.6 | 57.5 | – | – | 13.5* | 20.7* | 12.4* | 9.1* |
| RSVP | GPT-4o | 64.7 | 55.4 | 61.9 | 60.3 | – | – | – | – |
| Overall state-of-the-art[†] | | 65.0 | 61.3 | 55.4 | 63.2 | 36.5 | 54.4 | 33.2 | 25.5 |
| **SAM 3** Agent | Qwen2.5-VL 7B | 62.2 | 63.0 | 59.4 | 64.1 | 36.7 | 52.6 | 34.3 | 26.6 |
| **SAM 3** Agent | Llama4 Maverick | 68.5 | 67.1 | 66.8 | 67.2 | 32.8 | 43.7 | 30.9 | 27.5 |
| **SAM 3** Agent | Qwen2.5-VL 72B | 74.6 | 70.8 | 70.3 | 71.0 | 42.0 | **56.0** | 40.4 | 33.2 |
| **SAM 3** Agent | Gemini 2.5 Pro | **77.0** | **74.0** | **75.8** | 73.4 | **45.3** | 53.8 | **45.1** | **37.7** |

Table 8: SAM 3 Agent results. Gray indicates fine-tuned results on ReasonSeg (train), * indicates reproduced results, underline indicates the main metric. †: LISA-13B-LLaVA1.5 for ReasonSeg; REAL for OmniLabel.

**(a) Presence head.**

| | cgF$_1$ | IL_MCC | pmF$_1$ |
|---|---|---|---|
| × | 50.7 | 0.77 | **65.4** |
| ✓ | **52.2** | **0.82** | 63.4 |

**(b) Hard Negatives.**

| #/img | cgF$_1$ | IL_MCC | pmF$_1$ |
|---|---|---|---|
| 0 | 28.3 | 0.44 | 62.4 |
| 5 | 39.4 | 0.62 | **62.9** |
| 15 | 41.8 | 0.67 | 62.4 |
| 30 | **43.0** | **0.68** | 62.8 |

**(c) Training data.**

| EXT | SYN | HQ | cgF$_1$ | IL_MCC | pmF$_1$ |
|---|---|---|---|---|---|
| ✓ | × | × | 23.7 | 0.46 | 50.4 |
| ✓ | ✓ | × | 32.8 | 0.57 | 56.9 |
| ✓ | × | ✓ | 45.5 | 0.71 | **64.0** |
| ✓ | ✓ | ✓ | **47.4** | **0.74** | 63.8 |

**(d) SAM 3 + AI verifiers.**

| Model | cgF$_1$ | IL_MCC | pmF$_1$ |
|---|---|---|---|
| Human | 72.8 | 0.94 | 77.0 |
| SAM 3 | 54.0 | 0.82 | 65.9 |
| + EV AI | 61.2 | 0.86 | 70.8 |
| + MV AI | **62.3** | **0.87** | **71.1** |

Table 9: Selected model and data ablations on SA-Co/Gold. Numbers *across* tables are not directly comparable.

**Selected Ablations.** In Tab. 9 we report a subset of the more extensive ablations from §B. Note that the ablated models are from different, shorter training runs than the model evaluated above. The presence head boosts cgF$_1$ by +1.5 (9a), improving image-level recognition measured by IL_MCC by +0.05. Tab. 9b shows that adding hard negatives significantly improves the model performance, most notably the image-level IL_MCC from 0.44 to 0.68. Tab. 9c shows that synthetic (SYN) training data improves over the external (EXT) by +8.8 cgF$_1$ and our high-quality (HQ) annotations add +14.6 cgF$_1$ on top of this baseline. We present detailed data scaling laws of both types of data in §B.2, showing their effectiveness on both in-domain and out-of-domain test sets. In Tab. 9d, we show how AI verifiers can improve pseudo-labels. Replacing the presence score from SAM 3 with that score from the exhaustivity verification (EV) AI verifier boosts cgF$_1$ by +7.2. Using the mask verification (MV) AI verifier to remove bad masks adds another 1.1 points. Overall, AI verifiers close half of the gap between SAM 3's and human performance.

**Domain adaptation ablation.** With domain-specific synthetic data generated by SAM 3 + AI verifiers, we show that one can significantly improve performance on a new domain *without any human annotation*. We hold out one of the SA-Co domains, "Food&drink", from training SAM 3 and AI verifiers. We then use three variants of training data for the *novel* "Food&drink" domain: high-quality AI+human annotations as in SA-Co/HQ (referred to as **SA-Co/HQ-Food**), synthetic annotations as in SA-Co/SYN, using AI but no humans (**SA-Co/SYN-Food**), and pseudo-labels generated before the AI verification step, i.e. skipping both AI verifiers and humans (**PL-Food**). Fig. 8 plots performance on the "Food&drink" test set of the SA-Co/Gold benchmark as each type of training data is scaled up. We mix the domain specific data and high-quality general domain data at a 1:1 ratio. PL-Food provides some

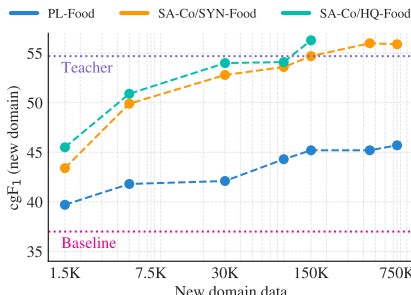

Figure 8: **Domain adaptation via synthetic data.** Synthetic (SYN) data generated by SAM 3 + AI verifiers (teacher system) achieves similar scaling behavior as *human-annotated* (HQ) data.

improvement compared to the baseline SAM 3 (zero-shot), but is far below the other variants due to its lower quality. HQ-Food and SYN-Food show similar scaling behavior, with SYN-Food slightly lower but eventually catching up, without incurring any human annotation cost. This points to a scalable way to improve performance on new data distributions. More details are in §B.3.

## 7 CONCLUSION

We present Segment Anything with *Concepts*, enabling open-vocabulary text and image exemplars as prompts in interactive segmentation. Our principal contributions are: (i) introducing the PCS task and SA-Co benchmark, (ii) an architecture that decouples recognition, localization and tracking and extends SAM 2 to solve concept segmentation while retaining visual segmentation capabilities, (iii) a high-quality, efficient data engine that leverages the complimentary strengths of human and AI annotators. SAM 3 achieves state-of-the-art results, doubling performance over prior systems for PCS on SA-Co in images and videos. That said, our model has several limitations. For example, it struggles to generalize to out-of-domain terms, which could be mitigated by automatic domain expansion but requires extra training. We discuss this and other limitations of our model in §C. We believe SAM 3 and the SA-Co benchmark will be important milestones and pave the way for future research and applications in computer vision.

ACKNOWLEDGMENTS

We would like to thank the following people for their contributions to the SAM 3 project: Alex He, Alexander Kirillov, Alyssa Newcomb, Ana Paula Kirschner Mofarrej, Andrea Madotto, Andrew Westbury, Ashley Gabriel, Azita Shokpour, Ben Samples, Bernie Huang, Carleigh Wood, Ching-Feng Yeh, Christian Puhrsch, Claudette Ward, Daniel Bolya, Daniel Li, Facundo Figueroa, Fazila Vhora, George Orlin, Hanzi Mao, Helen Klein, Hu Xu, Ida Cheng, Jake Kinney, Jiale Zhi, Jo Sampaio, Joel Schlosser, Justin Johnson, Kai Brown, Karen Bergan, Karla Martucci, Kenny Lehmann, Maddie Mintz, Mallika Malhotra, Matt Ward, Michelle Chan, Michelle Restrepo, Miran Heo, Miranda Hartley, Muhammad Maaz, Nisha Deo, Peter Park, Phillip Thomas, Raghu Nayani, Rene Martinez Doehner, Robbie Adkins, Ross Girshik, Sasha Mitts, Shashank Jain, Spencer Whitehead, Ty Toledano, Valentin Gabeur, Vincent Cho, Vivian Lee, William Ngan, Xuehai He, Yael Yungster, Ziqi Pang, Ziyi Dou, Zoe Quake. We also thank the IDEA team for granting us DINO-X and T-Rex2 access to benchmark them on the SA-Co/Gold dataset.

APPENDIX

## A   RELATED WORK

**Promptable and Interactive Visual Segmentation.** SAM (Kirillov et al., 2023) introduces "promptable" image segmentation with interactive refinement. While the original task definition included text prompts, they were not fully developed. SAM 2 (Ravi et al., 2024) extended the promptable visual segmentation task to video, allowing refinement points on any frame. SAM 3 inherits geometry-based segmentation while extending to include text and image exemplar prompts to segment all instances of a concept in images and videos.

**Open-Vocabulary Detection and Segmentation in Images** exhaustively labels every instance of an open-vocabulary object category with a coarse bounding box (detection) or a fine-grained pixel mask (segmentation). Recent open-vocabulary (OV) detection  (Gu et al., 2021; Minderer et al., 2022) and segmentation (Ding et al., 2022; Liang et al., 2023) methods leverage large-scale vision-language encoders such as CLIP (Radford et al., 2021) to handle categories described by arbitrary text, even those never seen during training. While DETR (Carion et al., 2020) is limited to a closed set of categories seen during training, MDETR (Kamath et al., 2021) evolves the approach to condition on raw text queries. Image exemplars used as prompts to specify the desired object category (e.g., DINOv (Li et al., 2023a), T-Rex2 (Jiang et al., 2024)) present a practical alternative to text, but fall short in conveying the abstract concept of objects as effectively as text prompts. We introduce a new benchmark for OV segmentation with $> 100\times$ more unique concepts than prior work.

**Visual Grounding** localizes a language expression referring to a region of the image with a box or mask. (Plummer et al., 2020) introduces phrase detection as both deciding whether the phrase is relevant to an image and localizing it. GLIP (Li et al., 2022b) and GroundingDino (Liu et al., 2023) formulate object detection as phrase grounding, unifying both tasks during training. MQ-GLIP (Xu et al., 2023) adds image exemplars to text as queries. Building on this trend toward models supporting multiple tasks and modalities, GLEE (Wu et al., 2024a) allows text phrases, referring expressions, and visual prompts for category and instance grounding in both images and videos. Unlike SAM 3, GLEE does not support exemplars or interactive refinement. LISA (Lai et al., 2024) allows segmentation that requires reasoning, while OMG-LLaVa (Zhang et al., 2024a) and GLaMM (Rasheed et al., 2024) generate natural language responses interleaved with corresponding segmentation masks, with GLaMM accepting both textual and optional image prompts as input. Some general-purpose MLLMs can output boxes and masks (Gemini2.5 (Comanici et al., 2025)) or points (Molmo (Deitke et al., 2025)). SAM 3 can be used as a "vision tool" in combination with an MLLM (§6).

**Multi-Object Tracking and Segmentation** methods identify object instances in video and track them, associating each with a unique ID. In tracking-by-detection methods, detection is performed independently on each frame to produce boxes and confidence scores, followed by association of boxes using motion-based and appearance-based matching as in SORT (Bewley et al., 2016; Wojke et al., 2017), Tracktor (Bergmann et al., 2019), ByteTrack (Zhang et al., 2022c), SAM2MOT (Jiang et al., 2025), or OC-SORT (Cao et al., 2023). An alternative is an end-to-end trainable architecture that jointly detects and associates objects, e.g., TrackFormer (Meinhardt et al., 2022), TransTrack (Sun et al., 2020), or MOTR (Zeng et al., 2022). TrackFormer uses a DETR-like encoder-decoder that initializes new tracks from static *object queries* and auto-regressively follows existing tracks with identity-preserving *track queries*. A challenge with joint models is the conflict between detection and tracking  (Feichtenhofer et al., 2017; Yu et al., 2023a), where one needs to focus on semantics while

the other on disentangling identities, even if their spatial locations overlap over time. SAM 3 is a strong image detector tightly integrated into a tracker to segment concepts in videos.

# B    ABLATIONS

## B.1    MODEL ABLATIONS

**Presence Token.** We first ablate the impact of the presence token and the approach to its training. The presence token is included in the decoder (discussed further in §D.2), together with the object queries, and predicts a *concept* presence score. The presence score receives gradients *only* on the PCS task during joint training and is *always* supervised with the presence (or absence) of the concept in the image using a binary cross-entropy loss. Using a presence token to *decouple* presence and localization brings significant gains in performance, particularly on IL_MCC, see Tab. 9a.

When used with a presence score, we found that it is better for the box/mask object scores to *not* receive gradients when a concept is an image-level negative, see Setting (a) in Tab. 10. Note that this is in contrast to the approach in typical DETR variants, where all individual object scores are supervised *negatively* to reflect the absence of the concept in the image, see Setting (b) in Tab. 10. We find that (b) works worse than (a) when used with the presence score. When a concept is *present* in the image, individual object queries always receive classification supervision based on Hungarian matching. Setting (a) is consistent with our recognition-localization decoupled design, where the presence score is responsible for recognition (existence in the image) and the object scores are responsible for localization (i.e., rank the best match to the *positive* ground-truth highest among all the proposals).

During inference, we use the product of the global presence score and the object score as the total object score. In Setting (c), we explored directly supervising the *total* object scores (instead of the typical object scores) as positive or negative (as determined by matching); this setting can slightly improve the overall $cgF_1$, but is less *flexible* as the presence and object scores are *jointly* calibrated, e.g. such a model is less amenable to conditioning on a concept known to be present in the image. Finally, Setting (d) in Tab. 10 investigates detaching the presence score from the computation graph while supervising the total scores, but this does not improve over (c).

|     | Supervise mask scores only when concept present | Supervise total score | Sup. total score, detach presence | SA-Co/Gold | | |
|-----|:---:|:---:|:---:|:---:|:---:|:---:|
|     |     |     |     | $cgF_1$ | IL_MCC | $pmF_1$ |
| a. | ✓ | ✗ | ✗ | 54.0 | 0.82 | 65.5 |
| b. | ✗ | ✗ | ✗ | 52.2 | 0.81 | 64.2 |
| c. | ✓ | ✓ | ✗ | **54.9** | **0.83** | **66.0** |
| d. | ✓ | ✗ | ✓ | 53.6 | **0.83** | 64.9 |

Table 10: **Supervision strategy for object/mask scores for a model with a presence token.** We find the best supervision strategy is to supervise mask scores only for positive concepts and to supervise the presence and mask scores separately, although their product is used as the total object score during inference.

Training with presence can be considered as a form of post-training and occurs in Stage 3 (see §D.4.1) of our training pipeline. By default, *ablations* do *not* undergo this stage unless otherwise mentioned.

**Vision and Text Encoder.** While SAM 2 uses an MAE (He et al., 2022) pre-trained Hiera (Ryali et al., 2023) vision encoder for its strong localization capability and efficiency for the more *geometric* PVS task, SAM 3 also needs strong *semantic* and *linguistic* understanding with broad coverage. We adapted PE (Bolya et al., 2025) for the vision and text encoders of SAM 3, so that a large and diverse set of concepts is seen in Stage 1 of training, while producing *aligned* image and text encoders. In Tab. 11, we compare performance with Hiera and DINOv2 (Oquab et al., 2024); since these vision encoders lack an aligned text encoder, we use DistilRoBERTa-base (Sanh et al., 2019). We find PE to be the best overall choice of vision backbone, and using its own aligned text encoder provides further gains over PE with an unaligned text baseline. Use of PE enables strong robustness in SAM 3 (here measured by AP on COCO-O, demonstrating good object detection across various domain shifts, e.g. "sketch", "cartoon", "painting", etc).

*Implementation Details.* The image resolution is set to 1008 px, 1008 px, 1152 px for PE, DINOv2, Hiera, respectively, ensuring the same number of tokens in the detector due to their differences in

| Encoder (patch size) | SA-Co/Gold ($cgF_1$) | COCO-O (AP) |
|---|---|---|
| PE-L+ (14) | **43.2** | **42.5** |
| PE-L+ (14) w/ DistilRoBERTa | 38.1 | 39.6 |
| DINOv2-L (14) | 35.3 | 31.9 |
| Hiera-L (16) | 32.8 | 22.0 |

Table 11: **Choice of encoders**. As SAM 3 needs both semantic visual and linguistic understanding, we find PE's aligned image and text encoders work well.

patch size. All vision encoders used global attention in only a subset of the layers, using windowed ($24 \times 24$ tokens) attention otherwise. Since Hiera is a hierarchical multiscale encoder, we set the window size to $24 \times 24$ in stage 3 of the encoder, which has most of the FLOPs. Since PE is capable of using relative positional information via RoPE (Su et al., 2021; Heo et al., 2024), we include relative positional embeddings in global layers for Hiera and DINOv2 following Bolya et al. (2024). All models are trained using SA-Co/HQ viewing 5 million samples over the course of training. Recipe is separately optimized for each choice of encoder. Tokens from the respective vision encoders are downsampled by $2 \times 2$ to 1296 tokens before being passed to the fusion encoder and detector.

## B.2 IMAGE TRAINING DATA ABLATIONS

*Setup*. We adopt a simplifed, lighter model and training strategy for ablations in this section. Specifically, we use (i) a stride-28 (instead of 14) variant of SAM 3 using $4\times$ fewer tokens in the detector, (ii) limit to 45% of the entire SA-Co/SYN dataset and adopt, (iii) shorter training schedules and do not run "presence post-training" (see §B), (iv) evaluations are on an internal version of SA-Co/Gold, which has slightly lower human performance than the public version ($cgF_1$: internal 70.8 vs public 72.8). This allows running ablations more efficiently (but results in lower absolute accuracy *vs.* SAM 3). We observed similar trends when training at scale.

**SAM 3 Training Data.** Tab. 9c analyzes the impact of various SA-Co training data subsets. Training with even with *just* SA-Co/EXT shows comparable performance with *best* external models on SA-Co/Gold (see OWLv2's and DINO-X's performance in Tab. 1), indicating a strong base model. Adding synthetic data SA-Co/SYN into the training mix results in significantly improved performance. The performance further increases after adding the high-quality SA-Co/HQ data due to its quality and distributional similarity with SA-Co/Gold. Although SA-Co/HQ is large-scale and in-domain with SA-Co/Gold, SA-Co/SYN shows further gains on SA-Co/Gold when added on top of SA-Co/HQ.

| Training data | SA-Co/Gold (All) (in-domain) | | | SA-Co/Gold-MetaCLIP (in-domain) | | | SA-Co/Gold-Wiki-Food&Drink (in domain) | | |
|---|---|---|---|---|---|---|---|---|---|
| | $cgF_1$ | IL_MCC | $pmF_1$ | $cgF_1$ | IL_MCC | $pmF_1$ | $cgF_1$ | IL_MCC | $pmF_1$ |
| SA-Co/EXT | 23.7 | 0.46 | 50.4 | 21.5 | 0.45 | 47.7 | 20.5 | 0.45 | 45.4 |
| + 1% SA-Co/HQ | 34.0 | 0.57 | 59.6 | 30.8 | 0.56 | 54.6 | 33.4 | 0.55 | 60.7 |
| + 4% SA-Co/HQ | 37.3 | 0.62 | 59.6 | 35.4 | 0.65 | 54.7 | 39.2 | 0.66 | 58.9 |
| + 10% SA-Co/HQ | 40.0 | 0.65 | 60.9 | 37.5 | 0.67 | 55.7 | 46.6 | 0.71 | 65.5 |
| + 20% SA-Co/HQ | 42.2 | 0.68 | 61.8 | 38.5 | 0.69 | 56.1 | 50.3 | 0.74 | 67.6 |
| + 100% SA-Co/HQ | **45.5** | **0.71** | **64.0** | **40.3** | **0.71** | **57.1** | **53.3** | **0.77** | **68.9** |
| Teacher (Human) | 70.8 | 0.944 | 71.5 | 63.3 | 0.936 | 67.7 | 77.3 | 0.964 | 80.2 |

Table 12: **SA-Co/HQ scaling.** SA-Co/EXT data alone is not enough to solve SA-Co/Gold, training on SA-Co/HQ scales well with increasing amount of data. Human performance given as an estimated range where applicable, see §F.4 for details. Ablations use a lighter model and training setting *vs.* SAM 3.

**SA-Co/HQ Scaling Law.** Tab. 12 investigates scaling behavior of the SA-Co/HQ training data. For this ablation, the data mix is sampled randomly from the entire SA-Co/HQ (collected from the three phases in §4) at a fixed percentage. We also report scaling behavior on two specific subsets of SA-Co/Gold: the MetaCLIP Xu et al. (2024b) subset annotated with generic caption-derived NPs, and Wiki-Food&Drink subset annotated with fine-grained NPs from SA-Co Ontology nodes. SA-Co/HQ improves performance on both subsets as expected, since they are from the same distribution (in-domain). We also report the Teacher (Human) performance in the last row. Due to the simplified setting, the gap between SAM 3 and Human is larger than that of the best SAM 3 model.

| Training data | SA-Co/Gold (All) | | | SA-Co/Gold-MetaCLIP (in-domain) | | | SA-Co/Gold-Wiki-Food&Drink (out-of-domain) | | |
|---|---|---|---|---|---|---|---|---|---|
| | $cgF_1$ | IL_MCC | $pmF_1$ | $cgF_1$ | IL_MCC | $pmF_1$ | $cgF_1$ | IL_MCC | $pmF_1$ |
| SA-Co/EXT | 23.7 | 0.46 | 50.4 | 21.5 | 0.45 | 47.7 | 20.5 | 0.45 | 45.4 |
| + 1% SA-Co/SYN | 30.1 | 0.52 | 57.3 | 31.0 | 0.58 | 53.7 | 26.2 | 0.44 | 59.1 |
| + 4% SA-Co/SYN | 30.7 | 0.53 | 56.9 | 31.8 | 0.59 | 53.6 | 27.8 | 0.47 | 59.7 |
| + 15% SA-Co/SYN | 32.1 | 0.56 | 56.6 | 33.3 | 0.62 | 53.4 | 29.5 | 0.50 | 59.5 |
| + 45% SA-Co/SYN | **32.8** | **0.57** | 56.9 | **34.5** | **0.64** | 53.6 | **30.1** | **0.51** | **59.6** |
| Teacher (SAM 3 + AI verifiers) | 55.4 | 0.84 | 65.3 | 48.3 | 0.83 | 58.5 | 59.0 | 0.87 | 68.1 |

Table 13: **SA-Co/SYN scaling.** SAM 3 benefits from increasing SA-Co/SYN data, *both* on MetaCLIP which is *in-domain* with the synthetic data, and on Wiki concepts which are *out-of-domain* of the synthetic data. The teacher that generated the SA-Co/SYN data consists of an older version of SAM 3, and AI verifiers from the SAM 3 data engine. Ablations use a lighter model and training setting *vs.* SAM 3.

**SA-Co/SYN Scaling Law.** Tab. 13 shows that SAM 3 scales well with SA-Co/SYN data on SA-Co/Gold benchmark as it benefits from the large scale concepts captured from image captions generated by Llama4 and alt-text associated with the images, for *both* the *in-domain* MetaCLIP subset and the *out-of-domain* Wiki-Food&Drink subset within the SA-Co/Gold benchmark. The last row shows the Teacher performance (an older version of SAM 3 and AI verifiers) is much better than the student, and explains why SA-Co/SYN is useful. When comparing the SA-Co/SYN in Tab. 13 and SA-Co/HQ in Tab. 12, the lower in-domain performance gap on MetaCLIP (42.5 *vs.* 49.0) comes from the relatively weaker annotation quality of SA-Co/SYN, due to lacking of the human correction step. The gap is larger on the out-of-domain Wiki-Food&Drink set (37.4 *vs.* 59.9), because SA-Co/SYN only covers the MetaCLIP images and noun phrases from a captioning model; see Table 26. We also show in Fig. 9 that with additional *in-domain* synthetic data, we can close the performance gap on SA-Co/Gold-Wiki-Food&Drink subset without *any* human involvement.

**Hard Negatives.** We ablate the number of hard negative noun phrases in SA-Co/HQ per image in Tab. 9b. We show that increasing the number of negatives improves SAM 3 performance across all metrics, most notably IL_MCC. Hard negatives are phrases that are not present in the image but that (a previous generation of) SAM 3 predicts masks for, i.e., they are adversarial to (a previous generation of) SAM 3. Training on such difficult distractors helps improve the image-level classification performance captured by the IL_MCC metric.

**SAM 3 and AI Verifiers.** AI verifiers improve performance over the final SAM 3 model alone on the PCS task, as shown in Tab. 9d, with per-domain results in Tab. 14. We first replace the presence score from SAM 3 with a presence score from the Exhaustivity Verification (EV) AI verifier (given the image and noun phrase with no objects as input, the probability of *not exhaustive*, defined in Tab. 22). This results in a +7.2 point gain in $cgF_1$, from both IL_MCC and $pmF_1$. The reason why EV presence score can even improve $pmF_1$ is because it serves as a better calibration of object scores. Then we apply the Mask Verification (MV) AI verifier to each mask, and remove the rejected masks. This results in a further +1.1 point gain in $cgF_1$. The system closes nearly half the gap between SAM 3 and human performance, which indicates potential further improvements of SAM 3 by scaling up the SA-Co/SYN data and SAM 3 model size.

| | Average | | | Metaclip | | | SA-1B | | | Crowded | | | Food&Drink | | | Sports Equip. | | | Attributes | | | Wiki-Common | | |
|---|---|---|---|---|---|---|---|---|---|---|---|---|---|---|---|---|---|---|---|---|---|---|---|---|
| | $cgF_1$ | IL_MCC | $pmF_1$ | $cgF_1$ | IL_MCC | $pmF_1$ | $cgF_1$ | IL_MCC | $pmF_1$ | $cgF_1$ | IL_MCC | $pmF_1$ | $cgF_1$ | IL_MCC | $pmF_1$ | $cgF_1$ | IL_MCC | $pmF_1$ | $cgF_1$ | IL_MCC | $pmF_1$ | $cgF_1$ | IL_MCC | $pmF_1$ |
| SAM 3 | 54.0 | 0.82 | 65.9 | 46.9 | 0.81 | 58.8 | 53.8 | 0.85 | 63.4 | 60.3 | 0.90 | 67.0 | 56.1 | 0.81 | 69.2 | 63.0 | 0.89 | 71.0 | 54.2 | 0.76 | 71.0 | 42.9 | 0.70 | 60.9 |
| SAM 3 +EV | 61.2 | 0.86 | 70.8 | 54.2 | 0.85 | 64.0 | 56.0 | 0.89 | 62.9 | 61.3 | 0.88 | 69.8 | 67.6 | 0.86 | 78.5 | 67.5 | 0.89 | 75.6 | 71.1 | 0.91 | 77.8 | 51.1 | 0.76 | 67.1 |
| SAM 3 +EV+MV | 62.3 | 0.87 | 71.1 | 56.5 | 0.88 | 64.3 | 58.0 | 0.90 | 64.2 | 62.7 | 0.89 | 70.6 | 68.0 | 0.86 | 78.9 | 67.2 | 0.88 | 76.0 | 70.9 | 0.92 | 77.4 | 52.3 | 0.79 | 66.5 |
| Human | 72.8 | 0.94 | 77.0 | 64.1 | 0.94 | 68.5 | 64.3 | 0.97 | 66.6 | 70.4 | 0.94 | 75.3 | 78.3 | 0.96 | 81.2 | 80.4 | 0.97 | 83.1 | 80.2 | 0.95 | 84.4 | 71.6 | 0.89 | 80.1 |

Table 14: Per-domain results of SAM 3 + AI verifiers in Tab. 9d.

## B.3 AUTOMATIC DOMAIN ADAPTATION

With domain-specific synthetic data generated by SAM 3 + AI verifiers, we show that one can significantly improve performance on a new domain *without any human annotation*. We select "Food & drink" concepts with MetaCLIP images as the new domain. We generated three variants of synthetic training data on this "Food & drink" domain, while ensuring that no data from the new domain was used in training the AI annotators (including SAM 3 and AI verifiers):

- **PL-Food**: We select "Food&drink" Wiki nodes and mine images from MetaCLIP (refer to *Concept Selection*, *Offline Concept Indexing* and *Online Mining* steps in §E.4 for more details on data mining). For pseudo-annotating fine-grained "Food&drink" concepts, we use Wiki ontology to identify relevant *coarse*-grained concepts that SAM 3 works well on and prompt SAM 3 with them to generate masks. This data is similar to typical pseudo-labeled data used in prior work for detection self-training (e.g. Minderer et al. (2022)).

- **SA-Co/SYN-Food**: PL-Food is cleaned by AI verifiers: MV AI verifier to remove bad masks, and EV AI verifier to verify exhaustivity/negativity of (image, noun phrase) pairs, as the AI verification step in Fig. 5.

- **SA-Co/HQ-Food**: PL-Food is cleaned by human verifiers for both MV and EV tasks. For non-exhaustive datapoints after EV, human annotators further manually correct them, as the "Correct" step in Fig. 5.

We study the data scaling law of these three variants by evaluating their performance on the Wiki-Food&Drink subset of the SA-Co/Gold benchmark.

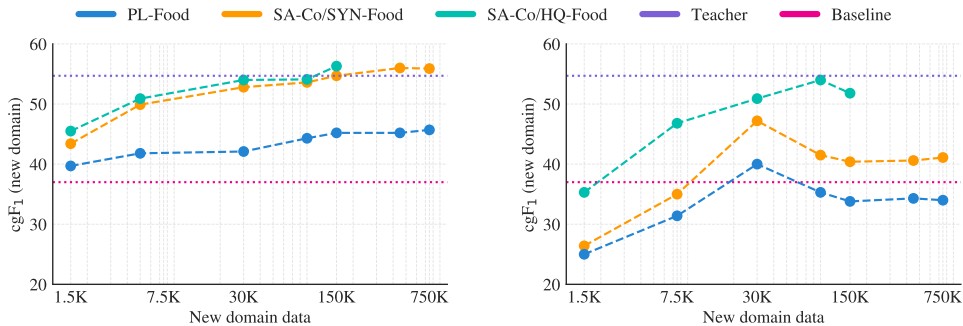

(a) Data scaling mixing pre-training data at a 1:1 ratio.   (b) Data scaling without mixing pre-training data.

Figure 9: **Domain adaptation via synthetic data.** (a) SAM 3 + AI verifiers (teacher system) can annotate synthetic (SYN) data in *new* domains (e.g., fine-grained food concepts) and achieve similar scaling behavior as with *human-annotated* (HQ) data. (b) Not mixing in high-quality pre-training data can limit performance gains when fine-tuning on new domains, particularly when using synthetic data.

We train the models in 2 steps to isolate the impact of the data from the new domain from other data as well as to amortize training costs. We first pre-train a base model using "SA-Co/HQ *minus* SA-Co/HQ-Food" to establish base capability and a common starting point. Next, we fine-tune the same base model with the three data variants in two settings: with or without mixing the pre-training data.

Fig. 9a shows the scaling law when mixing the synthetic data for the new domain with the pre-training data in a 1:1 ratio. We observe some improvement with PL-Food compared to baseline, but there is a large gap between the other variants due to its lower quality. SA-Co/HQ-Food and SA-Co/SYN-Food have similar data scaling behavior, with SA-Co/SYN-Food slightly lower but eventually catching up, without incurring any human annotation cost. The model trained on SA-Co/SYN-Food eventually surpassed the performance of its teacher system, thanks to the high quality pre-training data mixed during the fine-tuning.

Fig. 9b shows the scaling law when fine-tuned with only synthetic data for the new domain. All three data variants result in poorer performance than that in Fig. 9a. In this setting, there is a larger gap between SA-Co/HQ-Food and SA-Co/SYN-Food reflecting the lower quality of SA-Co/SYN-Food (mainly lack of exhaustivity due to no human correction). Comparing Fig. 9a and 9b, it is beneficial to include high-quality general-domain data when fine-tuning SAM 3 on new domains, particularly when using synthetic data.

### B.4 IMAGE DATA ENGINE ANNOTATION SPEED

Tab. 15 measures the speedup in the SAM 3 data engine from adding AI verifiers when collecting data on a new domain with fine-grained concepts. We use the same setup as Fig. 9, annotating

Wiki-Food&Drink data generated with a data engine where neither SAM 3 nor AI verifiers have been trained on Wiki-Food&Drink data. We annotate the same set of image-NP pairs in four settings:

- **Human (NP Input).** A human annotator is given a single image noun-phrase pair from SA-Co/HQ-Food, and is required to manually annotate all instance masks. No mask proposals or AI-verifiers are used in the loop.

- **Human (Mask Input).** The same annotation task as "NP input" but in this setting, the human annotators starts with PL-Food, i.e., image noun-phrase pairs with mask proposals generated by SAM 3.

- **Engine (All Human)** Similar to Phase 1 in the SAM 3 data engine, humans start with PL-Food, and sequentially perform 3 tasks: Mask Verification, Exhaustivity Verification and Correction. All three tasks are performed by humans.

- **Engine (Full)** Similar to Phase 3 in the SAM 3 data engine, Mask Verification and Exhaustivity Verification tasks are completed by AI verifiers, and Correction is done by humans i.e human annotators in the manual annotation task start with SA-Co/SYN-Food.

| Task | Human from NP | Human from masks | Engine - all human | Engine - full |
|---|---|---|---|---|
| Time for datamix (sec) | 90 | 86 | 50 | **23** |
| Time for positive NP (sec) | 236 | 205 | 207 | **152** |
| Time for negative NP (sec) | 71 | 70 | 30 | **6** |

Table 15: **Data engine efficiency - Image.** AI verifiers significantly increase throughput, allowing humans to focus on challenging cases and the manual correction task. AI verifiers allow for a 5x speed up on negative phrases and a 36% speed up for positive phrases. The time for datamix is calculated based on 88.5% negatives in SA-Co/HQ, see Tab. 24 for dataset composition. Timing is calculated based on the Wiki-Food&Drink domain. Compared to captioner-based domains, fine-grained domains require more research by annotators to understand the concept, leading to much higher annotation times for negative phrases and amortized time per mask.

Tab. 15 shows that a version of the SAM 3 model and AI verifiers that were never trained in this new domain *double* the throughput of the data engine. AI verifiers also allow verifying generated hard negative NPs at scale with close to no human-annotator involvement. As SAM 3 and AI verifiers are updated with the collected data and improve, human annotators need to manually correct fewer errors. This leads to increasingly higher throughput and the collection of more challenging data for a given amount of human annotation time.

In Tab. 23, we show that AI verifiers achieve a similar even better performance on the MV and EV tasks than human verifiers, so the quality of annotations from these four settings are similar.

### B.5 VIDEO DATA ENGINE ANNOTATION SPEED

Using the same settings as described in Section B.4, we evaluate annotation speed in the video data engine by comparing Human (NP Input) and Engine (All Human) on positive video-NP pairs from SA-Co/VEval - SA-V. In contrast to the image data engine, we observe that starting with PL increases annotation time, but also improves exhaustivity by providing annotators with more visual cues and candidate masklets.

| Task | Human (NP Input) | Engine (All Human) human |
|---|---|---|
| Time for positive NP (sec) | 2307 | 3221 |
| Number masklets per video-NP pair | 2.52 | 2.76 |
| Total masklets | 1700 | 1860 |

Table 16: **Data engine efficiency - Video.** While the Engine - all human is 40% slower than Human (NP Input) annotation, it yields 9% more masklets. PL helps annotators focus on regions where masklets potentially exist, but the additional time required for human verification and correction increases the overall annotation time. This experiment was run before the final cleaning of the SA-Co/VEval data, so the number of masklets might differ from the released version.

## B.6 Video Training Data Ablations

We analyze how much the SAM 3 model benefits from the videos and annotations in SA-Co/VIDEO obtained through the video data engine, which are used in Stage 4 (video-level) training (described further in §D.4.1). Specifically, we train the model with a varying amount of masklets from SA-Co/VIDEO as VOS training data, and evaluate the resulting checkpoints on SA-Co/VEval under the VOS task with the $\mathcal{J\&F}$ metric. The results are shown in Tab. 17, where adding masklets collected with noun phrases through the video data engine (as additional Stage 4 training data) improves the $\mathcal{J\&F}$ performance on both SA-Co/VEval and public benchmarks such as DAVIS17 (Pont-Tuset et al., 2017a) and SA-V (Ravi et al., 2024).

| Stage-4 training data | SA-Co/VEval YT-1B val $\mathcal{J\&F}$ | SA-Co/VEval SA-V val $\mathcal{J\&F}$ | DAVIS17 val $\mathcal{J\&F}$ | SA-V val $\mathcal{J\&F}$ | SA-V test $\mathcal{J\&F}$ |
|---|---|---|---|---|---|
| using SAM 2 video data only | 80.7 | 84.9 | 91.6 | 77.0 | 77.1 |
| + 25% SA-Co Train videos | 80.9 | 85.3 | 91.3 | 75.2 | 76.9 |
| + 50% SA-Co Train videos | 81.2 | 85.3 | 91.5 | 76.7 | 77.0 |
| + 75% SA-Co Train videos | **81.4** | 85.9 | **91.7** | 76.5 | **78.5** |
| + 100% SA-Co Train videos | **81.4** | **86.5** | 91.5 | **77.4** | 78.0 |

Table 17: Scaling analysis on SA-Co/VEval under Stage 4 (video-level) training, evaluated on multiple benchmarks through the Video Object Segmentation (VOS) task under the $\mathcal{J\&F}$ metric. Note that "SA-Co/VEval YT-1B" and "SA-Co/VEval SA-V" refer to the subset of SA-Co/VEval built upon YT-Temporal-1B videos and SA-V videos respectively, while "SA-V" referred to the VOS evaluation dataset released in Ravi et al. (2024).

## C Limitations

SAM 3 shows strong performance on the PCS task in images and videos but has limitations in many scenarios.

SAM 3 struggles to generalize to fine-grained out-of-domain concepts (e.g., aircraft types, medical terms) in a zero-shot manner, especially in niche visual domains (e.g., thermal imagery). Concept generalization for PCS is inherently more challenging than the class-agnostic generalization to new visual domains for the PVS task, with the latter being the key that enables SAM and SAM 2 to be successfully applied zero-shot in diverse settings. Our experiments show that SAM 3 is able to quickly adapt to new concepts and visual domains when fine-tuned on small quantities of human-annotated data (Tab. 2). Further, we show that we can improve the performance in a new domain without any human involvement (Fig. 9), using domain-specific synthetic data generated using our data engine.

From our formulation of the PCS task, SAM 3 is constrained to simple noun phrase prompts and does not support multi-attribute queries beyond one or two attributes or longer phrases including referring expressions. We show that when combined with an MLLM, SAM 3 is able to handle more complex phrases (§6 and §H).

In the video domain, SAM 3 tracks every object with a SAM 2 style masklet, which means the cost of SAM 3 inference scales linearly with the number of objects being tracked. To support real-time inference (30 FPS) on videos in practical applications (e.g., a web demo), we parallelize the inference over multiple GPUs: up to 10 objects on 2 H200s, up to 28 objects on 4 H200s, and up to 64 objects on 8 H200s. Further, under the current architecture, there is no shared object-level contextual information to aid in resolving ambiguities in multi-object tracking scenarios. Future developments could address this through shared global memory across multiple objects, which would also improve inference efficiency.

Supporting concept-level interactivity for PCS, alongside instance-level interactivity for PVS, poses several challenges. To support instance-level modifications without affecting all other instances of the concept, we enforce a hard "mode-switch" within the model from concept to instance mode. Future work could include more seamlessly interleaving concept and instance prompts.

# D  MODEL DETAILS

## D.1  MODEL ARCHITECTURE

Our architecture is broadly based on the SAM series (Ravi et al., 2024; Kirillov et al., 2023) and DETR (Carion et al., 2020) and uses a (dual) encoder-decoder transformer architecture, see Fig. 10 for an overview. SAM 3 is a generalization of SAM 2, supporting the new Promptable Concept Segmentation (PCS) task as well as the Promptable Visual Segmentation (PVS) task (Ravi et al., 2024). The design supports *multimodal prompts* (e.g., text, boxes, points) and *interactivity*, in images and videos.

SAM 3 has ∼850M parameters, distributed as follows: ∼450M and ∼300M for the vision and text encoders (Bolya et al., 2025), and ∼100M for the detector and tracker components. We next discuss the detector architecture for images followed by the tracker components built on top of it for video.

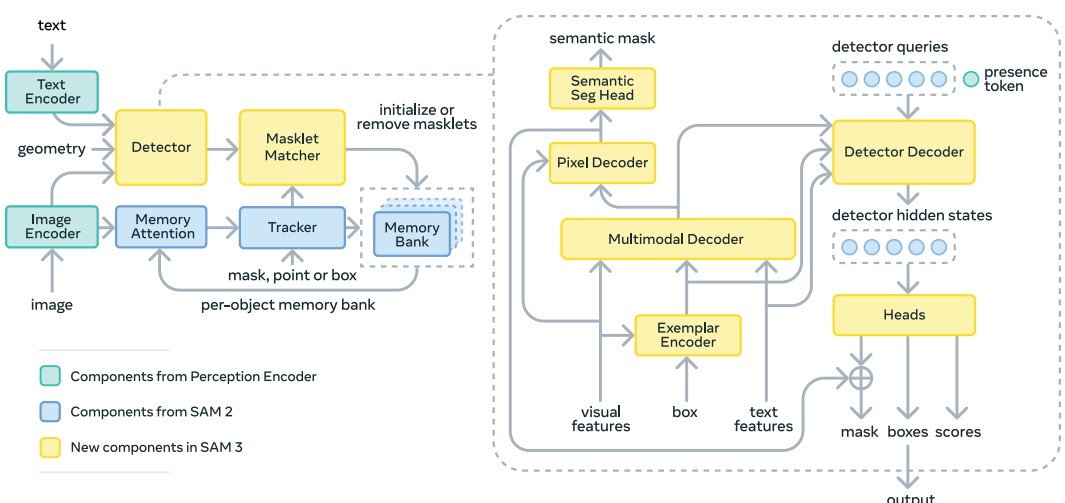

Figure 10: SAM 3 architecture. New components are in yellow, SAM 2 (Ravi et al., 2024) in blue and PE (Bolya et al., 2025) in cyan.

## D.2  IMAGE IMPLEMENTATION DETAILS

The image detector is an encoder-decoder transformer architecture. We describe its details in this section.

**Image and Text Encoders.** The image and text encoders are Transformers (Vaswani et al., 2017) trained using constrastive vision language training using 5.4 billion image-text pairs following Perception Encoder (PE) (Bolya et al., 2025), see §D.4.1 for training details. As in SAM 2, the vision encoder uses windowed attention (Ryali et al., 2023; Li et al., 2022d) and global attention in only a small subset of layers (4 out of 32), where an image of 1008 pixels is divided into 3×3 non-overlapping windows of 336 pixels each. The vision encoder uses RoPE (Su et al., 2021; Heo et al., 2024) in each layer and windowed absolute positional embeddings as in Bolya et al. (2024). The text encoder is causal, with a maximum context length of 32.

As in Ravi et al. (2024), we use a streaming approach, ingesting new frames as they become available. We run the PE backbone only *once* per frame for the entire interaction, which can span multiple forward/backward propagation steps through a video. The backbone provides unconditioned tokens (features/embeddings) representing each frame to the dual-encoder consisting of the fusion encoder described below and memory attention for video.

**Geometry and Exemplar Encoder.** The geometry and exemplar encoder is *primarily* used to encode image *exemplars* (if present) for the PCS task. It is additionally used to encode *visual prompts* for the PVS task on images as an *auxiliary* functionality that is primarily used to include pre-training data for the PVS task in stages-2,-3 of training (see §D.4.1), to enable a more modular training approach.

Each individual image exemplar is encoded using positional embedding, label embedding (positive or negative) and ROI-pooled visual features that are concatenated (comprising "exemplar tokens") and processed by a small transformer. Visual prompts (points, boxes) for auxiliary training are encoded in a similar manner, comprising "geometry tokens". It is possible for neither "geometry tokens" nor "exemplar tokens" to be present (e.g. when only a text prompt is used). The geometry or exemplar tokens attend to each other via self-attention and also cross-attend to the frame-embeddings of the corresponding (unconditioned) frame from the image encoder.

**Fusion Encoder.** The text and geometry/exemplar tokens together constitute the *prompt tokens*. The fusion encoder accepts the unconditioned frame-embeddings and conditions on prompt tokens using a stack of 6 transformer blocks with self- and cross-attention (to prompt tokens) layers followed by an MLP. We use vanilla self-attention operations. The output of the fusion encoder are the *conditioned* frame-embeddings.

**Decoder.** The decoder architecture follows Carion et al. (2020); Kamath et al. (2021) as a starting point and is a stack of 6 transformer blocks. $Q$ learned *object queries* (not to be confused with *prompts*) self-attend to each other and cross attend to the prompts tokens (made up of text and geometry/exemplar tokens) and conditioned frame-embeddings, followed by an MLP. We use box-to-pixel relative position bias (Lin et al., 2023) in the cross-attention layers attending to the conditioned frame-embeddings.

Following standard practice in stronger DETR variants, we use iterative box refinement (Zhu et al., 2020), look-forward-twice (Zhang et al., 2022a) and hybrid matching (Jia et al., 2022) and Divide-And-Conquer (DAC) DETR (Hu et al., 2023). By default, we use $Q = 200$ object queries. Bounding boxes and scores are predicted using dedicated MLPs and accept the object queries as input.

**Presence Head.** Classifying each object in isolation is often difficult, due to insufficient information, and may require contextual information from the rest of the image. Forcing each object query to acquire such global awareness is however detrimental, and can conflict with the localization objectives that are by nature very local. To address this, we propose decomposing the classification problem into two complementary components: a global-level classification that determines object presence within the entire image, and a local-level localization that functions as foreground-background segmentation while preventing duplicate detections. Formally, we add the following structure: instead of predicting $p(\text{query}_i \text{ matches NP})$ directly, we break it down as

$$p(\text{query}_i \text{ matches NP}) = p(\text{query}_i \text{ matches NP} \mid \text{NP appears in image}) \cdot p(\text{NP appears in image}).$$

To compute $p(\text{NP appears in image})$, we use a *presence token*, which is added to our decoder and then fed through an MLP classification head. Crucially, the presence score is shared by all object queries. The per-query classification loss is kept as usual, but to account for the decomposition, we only compute it when the NP is present in the image (see §B.1 for ablations on supervision strategy). The same decomposition is applied to the semantic segmentation head, where we reuse the same presence score, and train the binary mask head only on the positive examples.

Besides being more robust to false positives, decomposing the prediction in this manner is also more *flexible*, e.g. in typical counting tasks, we already know the NP is present in the image and instead want to know how many instances are present - in this case we can simply set $p(\text{NP is present in frame}) = 1$. The presence token is concatenated with the object queries in all operations, but is excluded from DAC.

We also learn 4 *geometric* queries. Their function is similar to the 4 geometric queries in SAM 1 and 2 (where they were called "output tokens") and are used to perform the PVS on individual image or video frames during the stags-2,-3 of training, see §D.4.1. The prompts are provided by the "geometry tokens" in the form of visual prompts. The presence score is set to 1 when performing the PVS task on a single frame, as the target is *known* to be present in the frame.

**Segmentation Head.** The segmentation head is adapted from MaskFormer (Cheng et al., 2021). Semantic segmentation and instance segmentation share the same segmentation head. The *conditioned* features from the fusion encoder are used to produce semantic segmentation masks, while instance segmentation additionally uses the decoder's output object queries. "Multi-scale" features are provided to the segmentation head using SimpleFPN (Li et al., 2022d), since the vision encoder is a (single-scale) ViT.

**Handling Ambiguity.** Experimentally, if we train a SAM 3 model without handling ambiguities as described in §2 in any way, we observe that the model tends to predict several valid but conflicting interpretations of the phrase. This is expected; if in our training dataset a given phrase has two distinct interpretations, and roughly half the data is annotated assuming the first one, while the other half follows the second one, then the solution that minimizes the training loss is to output both interpretations with 50% confidence. However, this behavior is undesirable for end-users, because it produces conflicting, sometimes overlapping masks.

To address this issue, we add an ambiguity head to our model. Similar to SAM 1 and 2, this head is a mixture of experts, where we train in parallel $K$ experts, and only supervise the expert that gets the lowest loss (winner-takes-all). We find that $K = 2$ performs the best and that it is more difficult to train $K > 3$ experts due to mode collapse.

For a mixture of $K$ experts, each producing an output $y_k$ with loss $\mathcal{L}_k$, the mixture loss is a weighted average:

$$\text{Loss:} \quad \mathcal{L}_{\text{MoE}} = \sum_{k=1}^{K} p_k \, \mathcal{L}_k \qquad \text{Gradient:} \quad \frac{\partial \mathcal{L}_{\text{MoE}}}{\partial \theta_j} = p_j \, \frac{\partial \mathcal{L}_j}{\partial \theta_j}.$$

In our winner-takes-all variant, only the expert with the lowest loss receives gradient:

$$\text{Loss:} \quad k^\star = \arg\min_k \mathcal{L}_k, \qquad \mathcal{L}_{\text{WTA}} = \mathcal{L}_{k^\star}$$

$$\text{Gradient:} \quad \frac{\partial \mathcal{L}_{\text{WTA}}}{\partial \theta_j} = \begin{cases} \dfrac{\partial \mathcal{L}_{k^\star}}{\partial \theta_j}, & \text{if } j = k^\star, \\ 0, & \text{otherwise.} \end{cases}$$

Backpropagating the loss only through the expert which received the minimal loss allows each expert to specialize to one kind of interpretation. This behavior is illustrated in Fig. 11.

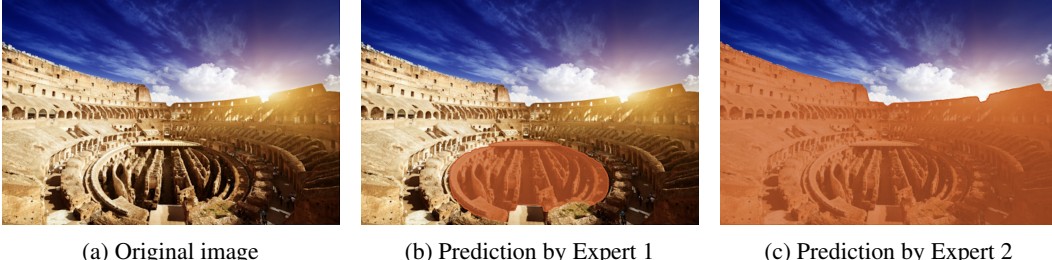

|       (a) Original image       |   (b) Prediction by Expert 1   |   (c) Prediction by Expert 2   |

Figure 11: Two interpretations of the noun phrase "large circular shape" learned by two Experts (SA-1B image).

While this strategy allows experts to specialize, it does not explicitly select which expert should be used at inference time. To resolve this, we train a classification head that predicts the expert that has the highest probability of being correct. The classification head is trained in a supervised fashion with a cross entropy loss, by predicting which expert obtained the minimal loss during training. The Ambiguity head adjusts only the classification logits, leaving masks, boxes, and presence scores unchanged. We train it on top of a frozen SAM 3 model.

Finally, to detect overlapping instances, we compute the Intersection-over-Minimum (IoM) between masks. IoM is more effective than Intersection-over-Union (IoU) for identifying nested instances. With the ambiguity head, we obtain a *15% reduction* in overlapping instances.

### D.3    VIDEO IMPLEMENTATION DETAILS

The tracker architecture follows Ravi et al. (2024), which we briefly describe for convenience followed by a discussion of the disambiguation strategies we introduce.

**Tracker.** The image encoder is the *shared* PE (Bolya et al., 2025) with the detector and provides unconditioned tokens to the memory attention using a *separate* neck. The memory attention receives these unconditioned PE tokens stacks self- and cross- attention layers that condition the current frame's tokens on spatial memories and corresponding object pointers in the memory bank. Memories are encoded by fusing a frame's mask prediction with the unconditioned PE tokens from the image encoder and placed in the memory bank.

As in Ravi et al. (2024), the decoder includes an occlusion head to indicate the likelihood of the object of interest being visible in the current frame. During inference, the occlusion score may also be used to select frames to place in the memory bank adaptively.

SAM introduced the ability to output multiple valid masks when faced with ambiguity about the object being segmented in an image. For example, when a person clicks on the tire of a bike, the model can interpret this click as referring to only the tire or the entire bike and output multiple predictions. In videos, this ambiguity can extend across video frames. For example, if in one frame only the tire is visible, a click on the tire might relate to just the tire, or as more of the bike becomes visible in subsequent frames, this click could have been intended for the entire bike. To handle this ambiguity, SAM 2 predicts multiple masks at each step of the video. If further prompts do not resolve the ambiguity, the model selects the mask with the highest predicted IoU for the current frame for further propagation in the video although other strategies are possible.

**Disambiguation Strategy.** As outlined in §3, tracking in videos can suffer from ambiguities in mask propagation, false predictions from the detector, or limitations of IoU-based matching in crowded scenes with highly overlapping objects. In this section, we present the details of the temporal disambiguation strategies used to address these challenges. We begin by introducing the notation used throughout this section.

Let $\mathcal{D}_\tau$ and $\hat{\mathcal{M}}_\tau$ denote the set of detector outputs and the set of tracker's predicted masks on frame $\tau$ respectively. We define a frame-wise matching function $\Delta_i(\tau)$ for a masklet $i$ on frame $\tau$ as

$$\Delta_i(\tau) = \begin{cases} +1, & \text{if } \exists\, d \in \mathcal{D}_\tau \text{ s.t. } \text{IoU}(d, \hat{\mathcal{M}}_\tau^i) > \text{iou\_threshold} \\ -1, & \text{otherwise,} \end{cases}$$

where $\hat{\mathcal{M}}_\tau^i$ is the predicted output mask of object $i$ on frame $\tau$. In addition, we define a Masklet Detection Score (MDS) over an interval $[t, t']$ as $S_i(t, t') = \sum_{\tau=t}^{t'} \Delta_i(\tau)$. This score measures how a masklet is consistently matched to a detection within a temporal window. The first frame in which object $i$ appears is denoted $t_{\text{first}}^i$.

**Track Confirmation Delay.** To reduce spurious and duplicate masklets, we delay the output of the model slightly. Specifically, the output at frame $\tau$ is shown only after observing frame $\tau + T$. This delay provides temporal context for validating candidate masklets before outputting their masks. By default, we use $T = 15$ which achieves good accuracy at a slight delay cost of around half a second for 30 frames per second videos. During the delay, we apply the following two criteria to remove *unconfirmed* or *duplicate* masklets as follows.

**Removal of Unconfirmed Masklets.** A candidate masklet is considered unconfirmed within the confirmation window $[t, t + T]$ if their MDS is below a threshold, $S_i(t, t + T) < V$, and the masklet first appears within the window $t_{\text{first}}^i \geq t$. If both conditions are satisfied within the confirmation delay, we remove the masklet from the tracker's state. We choose $V = 0$, requiring that the masklet has to be matched to a detection for at least half of the frames within the confirmation delay period to be confirmed. This strategy helps reject some false positive detections and not track them.

**Removal of Duplicate Masklets.** If the tracker temporarily fails to predict a mask for an object in some frames, but the detector continues to detect the object during those frames, this can lead to the creation of a new masklet for the same object. As a result, two masklets may end up tracking the same object: the original (older) masklet, and a new masklet that is initiated during the period when the tracker missed the object. To resolve this issue, during the confirmation delay period, if two masklets consistently overlap with the same detection, we remove the one that started later. Specifically, two masklets $i, j$ are considered duplicates on frame $\tau$ if there exists a detection $d \in \mathcal{D}_\tau$ such that $\text{IoU}(\hat{\mathcal{M}}_\tau^i, d) \geq \text{iou\_threshold}$ and $\text{IoU}(\hat{\mathcal{M}}_\tau^j, d) \geq \text{iou\_threshold}$. If the two masklets $i$ and $j$ are found to be duplicates for at least $\lceil T/2 \rceil$ frames, we remove the one with the latest first

appearance $t_{\text{first}}$ only if it first appeared within the confirmation window $[t, t + T]$. Empirically, we find that using $\text{iou\_threshold} = 0.1$ gives the best results.

**Masklet Suppression.** For confirmed masklets that were not removed during the confirmation delay, we apply an additional suppression step: if a masklet's MDS over its entire lifetime falls below zero at any frame $\tau$ (i.e. $S_i(t_{\text{first}}^i, \tau) < 0$), we suppress its output by zeroing out its mask. However, we retain the masklet in the tracker's state, allowing for the possibility that the object may be confirmed in future frames. This strategy primarily addresses ambiguous detections, such as objects entering the scene near the boundary. For example, if only a person's hands are visible as they enter the frame, the detector may be unable to determine whether the object matches the text prompt (e.g., impossible to distinguish between a man and a woman). In such cases, if the detector subsequently fails to detect the object after it fully enters the scene, the masklet suppression criterion ensures that these masklets are suppressed, unless they are consistently matched with new detections.

**Periodic Re-Prompting.** In challenging scenarios involving occlusions or visually similar distractor objects, the tracker may lose track of the target object. To address such tracking failures, we periodically re-prompt the tracker using the latest detection outputs. Specifically, on every $N$-th frame $\tau$, we compare each detection $d \in \mathcal{D}_\tau$ with the tracker's current predictions $\hat{\mathcal{M}}_\tau$. If a detection $d$ has a high overlap with the tracker's prediction (i.e., $\text{IoU}(d, \hat{\mathcal{M}}_\tau^i) \geq 0.8$) and both the detection score and the masklet prediction score exceed a confidence threshold of 0.8, we re-initialize the tracker for that object using the detection output mask. We observed that re-prompting is most effective on frames where objects are not occluded and are fully visible, which motivates our choice of high confidence thresholds. In our experiments, we set $N = 16$ by default. This periodic re-prompting helps the tracker recover from temporary failures and maintain accurate object tracking throughout the video.

**Detection-Guided Re-Prompting.** In cases where the tracker's predictions may drift and its predicted masks become leaky, we employ the detectors' outputs. For each frame $\tau$, we compare every detection $d \in \mathcal{D}_\tau$ with the tracker's current predictions $\hat{\mathcal{M}}_\tau$. If the highest-matching detection $d$ has a low bounding box IoU (i.e., $\text{IoU}_{\text{bbox}}(d, \hat{\mathcal{M}}_\tau^i) < 0.85$) with the corresponding tracker prediction $\hat{\mathcal{M}}_\tau^i$, we recondition the tracker for that object using the latest detector output. This approach ensures that the tracker remains synchronized with reliable detection results.

The impact of these strategies is ablated in Tab. 39, and they show quantitative improvements across our evaluation sets.

## D.4 Model Training

### D.4.1 Training Stages

SAM 3 is trained in 4 stages, with each stage introducing new capabilities or refining existing capabilities. Tab. 18 lists the data used in each stage.

**Stage 1.** Perception Encoder (PE) pre-training (Bolya et al., 2025), which pre-trains the image and text encoders with 5.4 billion image-text pairs. In addition to broad concept coverage, this stage is key for robustness (see §B.1). Since the vision encoder has to support multiple tasks (while also not being too large) we opt for an "L+" size; The vision and text encoders are transformers with 450M and 300M parameters respectively. We largely follow Bolya et al. (2025), but do not use distillation and do not perform video fine-tuning in this stage.

**Stage 2.** This stage is for detector pre-training and trains the (image-level) detector as well as the vision and text encoders with large-scale image segmentation data (including video frames as images). This stage uses both pseudo-labelled and human-annotated data, see Tab. 18. The main goal of this stage is broad concept coverage of (image, noun phrase, masks) tuples. At the end of this stage, the model is able to do open-vocabulary object detection, instance and semantic segmentation across many domains fairly well.

An additional goal of this stage is to prepare the base model for tasks in subsequent stages. To prepare for the PCS task, (image, noun phrase) pairs are randomly ($p = 0.2$) converted into visual queries (i.e. noun phrase is dropped) or augmented with input bounding boxes ($p = 0.2$).

Besides training for the PCS task, in this stage, the model is also pre-trained on the *visually prompted* PVS task. This is done by adding 4 decoder queries specific to this task following the design of SAM 1 & 2. Training data includes images (e.g., SA-1B) and videos frames (e.g, SA-V), see Tab. 18; the number of interactivity steps is restricted to 4 for efficiency. We largely follow the settings from Ravi et al. (2024), but use the Align loss (Cai et al., 2024) in lieu of the IoU prediction loss, co-opting the classification head for object queries for this task.

We train for $\sim$95k iterations with a batch size of 896 with 5k warm up and cooldown steps using AdamW (Loshchilov & Hutter, 2019). We apply layer-wise learning rate decay (Clark et al., 2020) of 0.9 to the vision encoder. We use a reciprocal square-root schedule (Zhai et al., 2022) and weight decay of 0.1. We use an initial learning rate of $5e$-4, $1e$-4 for vision and text encoder and $1e$-3 for all other components. For boxes, we use $L_1$ and gIoU losses with weights of 5 and 2. Classification loss uses a weight of 100 and focal and dice losses use weights of 200 and 10 respectively. The encoder and decoder use a dropout of 0.1.

**Stage 3.** This stage further trains the model with the highest-quality human annotated image segmentation data, expands the interactivity capabilities and introduces post-training to improve detection performance.

Specifically, in terms of interactivity, (a) in the PVS task, the number of interactivity steps is increased to 7 and (b) interactivity is introduced into the PCS task, where *positive or negative exemplars* are provided based on model error, as described next. We iteratively sample box prompts to mimic the real user policy. Positive boxes are sampled from false negative errors, and we prompt their corresponding ground-truth boxes. Negative boxes are sampled from high-confidence false positive predictions that do not have significant overlap with ground truths. At each iteration, the box inputs are added on top of the previous ones. If both a valid positive and negative box exist, we randomly select one; if no valid candidates are available, no additional prompt is given. The process is repeated for 5 iterations.

The expanded interactivity in the PCS and PVS in this stage significantly slows down training, but the extensive pretraining with limited interactivity for the PVS and no interactivity for PCS (but using image exemplars together with text prompts) prepares the model well to ensure that a short stage 3 is sufficient.

This stage retains only the highest quality, exhaustivity verified data (e.g., SA-Co/SYN is dropped) and introduces a presence token (and presence loss) to better model *presence* of target segments and their location *location* greatly increasing the precision of the model. The presence loss is a binary cross-entropy loss with weight of 20. All learning rates are lowered by a factor of 0.025. We train for $\sim$5k iterations with a batch size of 512, with other settings identical to stage 2.

**Stage 4.** For video, the tracker decoder is trained on top of the frozen backbone. Freezing the backbone at this stage is made possible by pre-training on VOS data in previous stages at the video frame level. This stage retains the strong spatial grounding of the previous stage and focuses on spatial-temporal tracking without degrading other capabilities. We use a batch size of 512, train for $\sim$190k iterations using a cosine schedule with a peak learning rate of $5.0e^{-4}$ and a linear warmup of 1k iterations. We supervise the model's outputs using a weighted sum of losses: a linear combination of focal and dice losses for mask prediction, mean absolute error (MAE) loss for IoU prediction, and cross-entropy loss for object occlusion prediction, with respective weights of 20:1:1:1. For multi-mask predictions, we only apply supervision to the mask with the lowest segmentation loss. If a frame's ground truth does not include a mask, we do not supervise any mask outputs for that frame; however, we always supervise the occlusion prediction head, which determines whether a mask should be present. As in Ravi et al. (2024), we further fine-tune the tracker with a longer temporal context using 16-frame and 32-frame videos for 60k iterations, while scaling the learning rate by a factor of 0.1.

### D.4.2    Additional Training Settings

**Data augmentation.** For the PCS task, we apply the following transformations:

- **Geometric**: We use some cropping and resizing to vary the aspect ratios and help with small objects. The input resolution of our model is always a fixed square (usually $1008 \times 1008$). During evaluation, the images are resized to this size, without respecting their aspect ratio. During training,

| Dataset | Ingested As | Train | Test | Stage 1 | Stage 2 | Stage 3 | Stage 4 |
|---|---|---|---|---|---|---|---|
| *Promptable Visual Segmentation- In Images* | | | | | | | |
| SA-1B | Image | ✓ | | | ✓ | ✓ | ✓ |
| SA-Co/VIDEO | Frames | ✓ | | | ✓ | ✓ | ✓ |
| SA-Co/VIDEO-EXT | Frames | ✓ | | | ✓ | ✓ | ✓ |
| SA-37 | Image | | ✓ | | ✓ | ✓ | ✓ |
| *Promptable Visual Segmentation- In Videos* | | | | | | | |
| SA-Co/VIDEO | Video | ✓ | | | | | ✓ |
| SA-Co/VIDEO-EXT | Video | ✓ | | | | | ✓ |
| SA-V val | Video | | ✓ | | | | ✓ |
| LVOSv2 | Video | | ✓ | | | | ✓ |
| *Promptable Concept Segmentation- In Images* | | | | | | | |
| SA-Co/SYN | Image | ✓ | | | ✓ | | |
| SA-Co/HQ | Image | ✓ | | | ✓ | ✓ | |
| SA-Co/EXT | Image | ✓ | | | ✓ | ✓ | |
| SA-Co/VIDEO | Frames | ✓ | | | ✓ | ✓ | |
| SA-Co/Gold | Image | | ✓ | | ✓ | ✓ | |
| SA-Co/Silver | Image | | ✓ | | ✓ | ✓ | |
| SA-Co/Bronze, SA-Co/Bio | Image | | ✓ | | ✓ | ✓ | |
| *Promptable Concept Segmentation- In Videos* | | | | | | | |
| SA-Co/VEval | Video | | ✓ | | | | ✓ |

Table 18: Dataset usage across different tasks and training stages.

we apply our augmentations, and pad if the resulting size is smaller than $1008 \times 1008$. We found it important to randomly distribute the padding on all sides, to avoid creating biases towards one particular region of the image. If the dataset does not contain notions of left and right, we also apply random horizontal flips.

- **Semantic**: When training on datasets with a closed vocabulary, we leverage our mapping to wikidata to further enhance the training. There are three main ways we can leverage the ontology: (i) to sample synonyms, which expand the vocabulary of the model; (ii) to sample negatives (typically, if the dataset is exhaustively annotated, we can sample any node in the graph that corresponds to a category and is not present in the image); and (iii) to ensure the hierarchy closure of the concepts (for example, if we have some annotations for "canoe" and "boat" in the same image, we need to make sure that all the "canoe" objects are also labeled as "boat" since a canoe is a type of boat).

- **Safety:** To prevent the model from randomly making predictions for unsafe concepts, we randomly sample some of them at train time and add them as negatives. These concepts mainly include slurs of all kinds. We also try to prevent the model from making predictions for subjective and non-visual adjectives, especially when applied to a person. This includes flattering ones (such as "a smart person") as well as derogatory ones (such as "a dull person").

- **Mosaics:** On some datasets, we further increase the complexity of the images by doing mosaics (Bochkovskiy et al., 2020). The maximal grid size of our mosaics is $3 \times 3$, and we sample any configuration that is at most that, including irregular ones, as long as the constituents are still square. For example, in a $3 \times 3$ regular grid, we can have a large image that effectively covers a $2 \times 2$ area, and use $1 \times 1$ for the remaining 5 slots. Unifying different images can be tricky in an open vocabulary setting, since there is no guarantee that concepts are exhaustively annotated. For example, if one image has a car annotated, but the second does not (neither as positive nor negative), then we do not know if the second image has a car or not, and thus could create some labeling noise. To avoid this, we only mosaic datasets that have low chance of such missing annotations (either closed vocabulary ones, or some created with specific mining patterns). To merge annotations, we again rely on the wikidata mapping if available, otherwise rely on plain-text queries to merge appropriately.

## E    DATA ENGINE DETAILS

The overview of the SAM 3 data engine's components is shown in Fig. 5. In this section we provide further details of how each component is implemented in the image (phases 1-3) and video (phase 4)

versions of the engine. The datasets collected in each phase and the performance improvements are in Tab. 19.

## E.1 MEDIA POOL

The media (image and video) pool consists of many sources with varying visual domains, from web-scraped data to datasets collected for specialized domains such as art, food, or driving. Tab. 26 lists the datasets used to mine media for each subset of the SA-Co training data. The web-scraped images and alt captions are sourced from MetaCLIP (Xu et al., 2024b), a curated version of CommonCrawl. We further expand coverage by mining media from a large pool with the help of a curated ontology. Compared to previous works such as OWLv2 (Minderer et al. (2024)) which mainly rely on uncurated web-scraped data, our target mining strategy resulted in coverage of 12 media domains.

## E.2 SA-CO ONTOLOGY

To track and improve the coverage and overall distribution of concepts in our data, we build a custom SA-Co ontology of visual concepts from Wikidata (Vrandečić & Krötzsch, 2014), which covers a comprehensive set of entities and offers hierarchical information with its graph data structure. We manually select high-level Wikidata nodes (e.g., Human, Mammals) and recursively include all of their descendants. The resulting 22.4 million nodes are classified into 17 top-level categories (e.g. animal, furnishing & home) and 72 sub-categories (e.g., birds, home appliance). The full list of categories and Wikidata node counts are shown in Tab. 20. We further develop a mapping process that can map an arbitrary NP to a node in the SA-Co ontology by leveraging a retrieval model (Sentence-BERT) to source candidate nodes and an AI annotator as judge (Llama 3.2) to select the closest match. This mapping is used to track the distribution of nodes in the dataset (see Fig. 12) as well as to create negative phrases (see below for details).

## E.3 PHASE 1: HUMAN VERIFICATION

**Data Mining.** During this phase, we randomly sample images from MetaCLIP.

**Proposing NPs.** We generate image-level captions using the BLIP-2 captioner (Li et al., 2023b) followed by the spaCy parser (Honnibal et al., 2020) to parse the caption into NPs.

**Proposing Masks.** We prompt an off-the-shelf open-vocabulary detector, FIBER (Dou et al., 2022) or OWLv2 (Minderer et al., 2024) with the noun phrase and use the resulting boxes to prompt SAM 2 to generate mask proposals.

**Verification (Human).** Verification of mask proposals consists of two tasks which can be performed by human or AI annotators: mask quality verification and mask exhaustivity verification. In Phase 1, verification is done by humans only. Each human verifier works exclusively on one task type.

- *Mask Verification (MV).* Given a triplet of an image, a noun phrase and a set of candidate masks for that phrase, the task is to accept or reject each of the masks. A mask is accepted if it matches the given noun phrase and is high quality (no holes, coverage issues, etc.) If the mask is unrelated to the phrase, or low quality, it is rejected.

- *Exhaustivity Verification (EV).* All accepted masks from the verification task are sent to an exhaustivity check. Given an image, noun phrase, and any accepted masks that passed the previous mask verification for that phrase, the task is to decide whether or not the accepted masks (if any)

| | SA-Co/HQ | | | SA-Co/SYN | | SA-Co/EXT | | SA-Co/VIDEO | | SAM 3 performance | | |
|---|---|---|---|---|---|---|---|---|---|---|---|---|
| | #img | #img-NP | #annotation domains | #img | #img-NP | #img | #img-NP | #vid | #vid-NP | SA-Co/Gold (cgF$_1$) | SA-Co/Silver | SA-Co/VEval (test pHOTA) |
| Phase 1 | 1.2M | 4.3M | 1 | 0 | 0 | 0 | 0 | 0 | 0 | - | - | - |
| Phase 2 | 2.4M | 122.2M | 5 | 0 | 0 | 0 | 0 | 0 | 0 | 21.4 | 18.9 | - |
| Phase 3 | 1.6M | 19.5M | 15 | 39.4M | 1.7B | 9.3M | 136.6M | - | - | 54.4 | 50.5 | - |
| Phase 4 | - | - | - | - | - | - | - | 52.5K | 134.3K | 54.5 | 50.1 | 63.9 |

Table 19: Data engine phases and SAM 3 progress.

| Category | Count | Category | Count | Category | Count |
|---|---|---|---|---|---|
| **1. animals** | **2.3M** | **6. electronics** | **10.2K** | **12. object parts** | **101.9K** |
| insects & crustaceans | 1.7M | electronics | 6.9K | body parts | 75.8K |
| molluscs | 188.4K | cameras | 3.3K | other object parts | 26.1K |
| other animals | 166.5K | **7. equipments** | **14.9K** | **13. other products** | **3.5K** |
| fish & other chordates | 85.7K | military equipments | 10.2K | other products | 2.7K |
| birds | 52.4K | sport equipments | 2.0K | celebration supplies | 384 |
| mammals | 38.3K | safety equipments | 1.2K | animal-related products | 359 |
| reptiles | 28.2K | medical equipments | 1.1K | tobacco products | 51 |
| echinoderms | 23.0K | agricultural machinery | 458 | **14. patterns & material** | **896.6K** |
| amphibians | 14.2K | **8. fashion & beauty** | **7.5K** | material | 885.6K |
| **2. art, history & religion** | **3.1M** | fashion | 3.9K | patterns & shapes | 10.9K |
| artworks | 3.1M | beauty & healthcare products | 3.7K | **15. plants & fungi** | **1.5M** |
| collectibles | 10.2K | **9. food & drinks** | **33.1K** | plants | 1.1M |
| religious objects | 9.1K | dishes | 12.9K | fungi | 376.4K |
| flags | 8.3K | other food | 6.7K | **16. tools & appliances** | **14.5K** |
| musical instruments | 4.9K | fruits | 6.6K | other appliances | 6.5K |
| gemstones | 526 | drinks | 6.3K | toys | 3.0K |
| art material | 438 | vegetables | 621 | tools | 1.8K |
| **3. buildings & locations** | **2.7M** | **10. furnishing & home** | **2.7K** | kitchenware | 1.6K |
| places | 2.4M | furnishing | 1.3K | containers | 945 |
| geographical features | 343.2K | home appliances | 486 | light sources | 672 |
| **4. celestial** | **9.2K** | stationery | 472 | **17. transportation** | **258.1K** |
| meteorological phenomena | 5.3K | household supplies | 417 | watercraft | 178.9K |
| space related | 3.1K | **11. human** | **11.3M** | land vehicles | 41.6K |
| light related | 734 | humans | 11.0M | aircraft | 27.4K |
| **5. documents & ocr** | **201.3K** | occupations | 140.8K | other vehicles | 6.4K |
| glyphs | 173.4K | fictional characters | 87.8K | transport infrastructures | 3.7K |
| logos | 21.0K | gestures & expressions | 158 | construction machines | 100 |
| documents | 6.0K | | | | |
| cards | 435 | | | | |
| infographics | 324 | | | | |
| GUI & layout elements | 135 | | | | |
| maps | 23 | | | | |

Table 20: SA-Co ontology top-level categories and sub-categories with corresponding node counts in Wikidata.

exhaustively cover all instances of the phrase in the image. If there are unmasked instances of the phrase, annotators decide whether or not at least one of the remaining instances is separable, or if the remaining instances are too crowded together to separate. Phrases that are annotated as non-exhaustive from this step are sent to the correction task. Phrases that are annotated as exhaustive are directly sent to final annotations.

**Correction.** Human annotators are given the same input as the exhaustivity task: an image, noun phrase, and any (0 or more) accepted masks from the mask verification task. Annotators manually add individual masks for the unmasked instances of the noun phrase by prompting SAM 1 with clicks in a browser based tool. If there are non-separable occurrences of the phrase, annotators use special group masks to indicate that the mask covers more than a single instance. The output of the task is a complete set of instance and/or group masks covering all pixels in the image corresponding to the noun phrase. Noun phrases that are not present are submitted with no masks. If it is not possible to reach a complete set of masks due to mask complexity, the annotator rejects the job.

In each task, annotators are given the ability to reject the image-NP pairing if they decide the phrase is un-maskable as a set of objects (e.g "it", "blue") or if after research they are still unsure if it is present (e.g., fine-grained species of animals). Filtering out vague phrases and allowing annotators to be unsure increases the consistency and agreement in the resulting annotations.

### E.4 PHASE 2: HUMAN + AI VERIFICATION

**Data Mining.** We use a retrieval model (including Perception Encoder, DINOv2, and MetaCLIPv2) for mining concepts that are challenging and not prevalent in the caption NPs from Phase 1. We leverage our SA-Co ontology to determine the list of candidate concepts, followed by offline concept indexing and online mining from MetaCLIP.

- *Concept Selection.* We use a taxonomy-guided mining strategy to balance the overall ontological distribution, expand concept coverage and enhance performance on long-tail and fine-grained phrases. Two groups of concepts are selected from the SA-Co Ontology for targeted mining: *Wiki-Common* are nodes judged by an LLM to be common concepts, *Wiki-FG* are all nodes from the "sports equipment" and "food and drink" sub-graphs, chosen to test the model's ability to generalize to very fine-grained concepts like "kefir", "pastille", "kettlebell".

- *Offline Concept Indexing.* For every new concept, we collect reference images from Wikimedia and compute their K-dimensional embedding offline. We aggregate the embeddings from all reference images resulting in a single embedding per concept. We repeat the process across all N concepts resulting in an N*K dimensional offline index.

- *Online Mining.* Relevant images for each concept are retrieved using both image and text based mining. With image-based retrieval, we compute the embedding on every image, run KNN on the offline concept index followed by top-k sampling, and apply a threshold before mapping it to a specific concept. With text-based retrieval, we compute CLIP based similarity scores between the text embedding from input concepts and image embeddings from the corpus and apply a threshold before mapping the image to a specific concept.

The following additional mining strategies are used to further refine the selection.

- *Image-Type Balancing.* Web datasets are usually dominated by a few types of images such as ads or product photos. To avoid over-representation of certain image types, we use a MLLM (Llama 3.2) and prompt it zero-shot to classify an image into image types (such as ads, product photos, indoor and outdoor scenes, infographics), and sample based on a type-agnostic probability.

**Proposing NPs.** We improve this step to generate higher-quality and more diverse noun phrases.

- *Image-Level Captioner and Parser.* We use an image captioning model (Llama 3.2) to generates image-level captions and a phrase parser (Llama 3.1) that proposes noun phrases given the caption. The Llama 3.2 captioning model improved concept recall compared to BLIP-2 from Phase 1. The phrase parser is fine-tuned for this task and significantly outperforms its zero-shot model variant and spaCy parser.

- *Removing Non-Groundable Phrases.* The parser can generate non-specific phrases such as "it", *"them"* or hard to segment phrases such as *"middle"*. To address this, we use another AI verifier (MLLM) that is fine-tuned to classify such cases and remove them from the rest of the pipeline.

- *NP Balancing.* We employ heuristics to avoid collecting too many frequent or easy objects. We remove NPs if the data engine has already annotated enough instances, if the SAM 3 has high accuracy when prompted with the NP, and based on a fixed list (e.g. that occur frequently, are harmful). From Phase 3 we rely on AI verifiers to remove easy cases.

- *Cleaning NPs.* We singularize noun phrases, deduplicate nearly-identical ones, and remove possessives.

- *Hard Negative Proposal.* A hard negative phrase generator proposes image-level negative phrases, i.e. those that do not exist in the image and are adversarial to SAM 3. Given verified positive NPs (i.e. that exist in the image), negative NPs are *proposed* and then *checked for adversariality*.

  - *Proposal.* The proposal of hard negatives is done in two ways. The first approach maps every positive NP to a node in the SA-Co ontology, then navigates the ontology graph to find sibling, cousin, or uncle nodes corresponding to different but related concepts. For example, the noun phrase "gray Siamese cat" maps to the node "Siamese cat", which could result in negative candidates like "tabby cat" (sibling), "dog" (uncle), or "Chihuahua" (cousin). The second approach relies on an MLLM (Llama 4), which proposes visually similar negatives for every positive NP.

– *Check for Adversariality.* Once the negative NPs are proposed, they are filtered to retain only those adversarial to the current SAM 3 version. For each negative NP candidate, predictions from SAM 3 are obtained. If the set of predictions is empty, the candidate is discarded. If the model predicts one or more objects, these predictions are compared to the original segmentation masks of the corresponding positive NP. If the overlap between the negative NP predictions and the positive NP annotations exceeds a certain threshold, the negative NP is retained as a hard negative. This final check is necessary because initial proposals may not be true negatives and instead may be only negatives relative to the existing positive NPs (i.e. the object could still be present somewhere else in the image).

**Proposing Masks.** We prompt SAM 3 with the set of positive and negative phrases to produce candidate instance and semantic masks for the image. For pseudo-annotating domains with fine-grained concepts that SAM 3 fails on (e.g., *Zanclus cornutus*), we identify the relevant coarse-grained concept that SAM 3 works well on (e.g., *frog*), and use this as the prompts to generate masks. We deduplicate masks generated per NP based on a IoU metric. These noisy pseudo-labels undergo further cleaning by both human and AI annotators.

**Verification (Human+AI).** We train "AI verifiers" to perform the mask verification (MV) and exhaustivity verification (EV) tasks. More specifically, we fine-tune Llama 3.2 Dubey et al. (2024) on human annotated data collected during Phase 1 of the data engine for both tasks.

- *Task Formulation.* Tab. 21 provides an example data point of the mask verification task: given an (image, phrase, mask) triplet, we render the mask on top of the image as the image prompt, provide the task guidance as text prompt, and use the human annotation (1 out of 5 choices) as output. Each mask's quality is evaluated independently from other masks for the same image-phrase pair. Rendering tricks are used to better visualize small objects, and to avoid color confusion from mask overlay. Tab. 22 provides an example data point of the exhaustivity verification task: given the (image, phrase, masks) triplet, we render the bounding boxes of the masks on top of the image and use this as the image prompt, provide the task guidance as the text prompt, and use the human annotation (1 out of 6 choices) as the output.

- *Evaluation.* We construct test sets for "AI verifiers" from jobs that were reviewed by multiple human annotators for all SA-Co test sets. We leave one human annotation as human prediction, and use the majority vote of the remaining human annotations as ground truth. This allows us to compare human and AI verifiers' accuracy.

- *Training.* The training data of each task comes from not only the task itself, but also from the Correction task. For example, each manually added mask is a good data point in the mask verification task. Each exhaustively finished job in the Correction task results in a good data point in exhaustivity verification task. We merge all training data for these two tasks together (over 200M image-text pairs) to pre-train a foundational AI verifier, and then only use high quality human annotated data from the task itself (around 10M scale) to fine-tune two AI verifiers, one for each task.

- *Result.* Thanks to the simplicity of these two tasks (MCQ tasks on image-text pairs) and the large volume of training data from Phase 1, AI verifiers reach and even surpass human performance on these two tasks, as shown in Tab. 23. We also evaluate the system of SAM 3 and AI verifiers end-to-end on the PCS task, and the system always performs better than the single SAM 3 model, as shown in Tab. 9d.

- *Generalization to new domains.* We also study the generalization ability of AI verifiers. For a given new domain, the MV AI verifier is typically on par with human verifiers without any domain specific data; the EV AI annotator is typically worse than human in a zero-shot evaluation, but can reach human performance with only thousands of domain specific data points.

As discussed in §B.4, using AI verifiers is effective and allows human annotators to focus on the most challenging data points, i.e. those that have poor mask quality or missing masks. This approach more than doubles the throughput of the SAM 3 data engine. As both SAM 3 and AI verifier models improve, more data can be exhaustively annotated using only SAM 3 and AI verifiers. This leads to increasingly higher throughput and ensures that human annotators only work on SAM 3 failure cases.

**Correction.** We perform manual correction wherever needed as described in phase 1.

| Input | 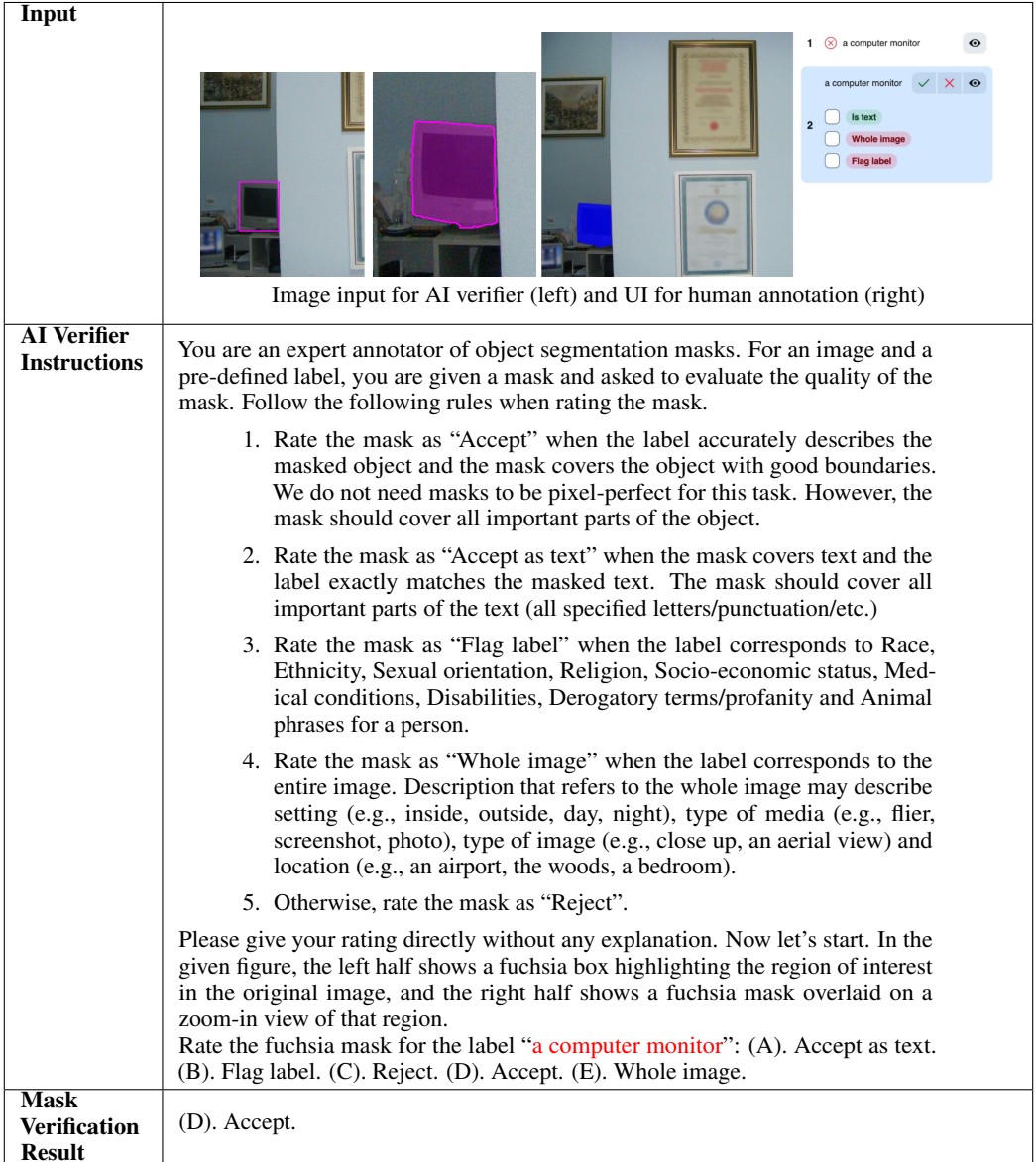 |
|---|---|
| | Image input for AI verifier (left) and UI for human annotation (right) |
| **AI Verifier Instructions** | You are an expert annotator of object segmentation masks. For an image and a pre-defined label, you are given a mask and asked to evaluate the quality of the mask. Follow the following rules when rating the mask.

1. Rate the mask as "Accept" when the label accurately describes the masked object and the mask covers the object with good boundaries. We do not need masks to be pixel-perfect for this task. However, the mask should cover all important parts of the object.

2. Rate the mask as "Accept as text" when the mask covers text and the label exactly matches the masked text. The mask should cover all important parts of the text (all specified letters/punctuation/etc.)

3. Rate the mask as "Flag label" when the label corresponds to Race, Ethnicity, Sexual orientation, Religion, Socio-economic status, Medical conditions, Disabilities, Derogatory terms/profanity and Animal phrases for a person.

4. Rate the mask as "Whole image" when the label corresponds to the entire image. Description that refers to the whole image may describe setting (e.g., inside, outside, day, night), type of media (e.g., flier, screenshot, photo), type of image (e.g., close up, an aerial view) and location (e.g., an airport, the woods, a bedroom).

5. Otherwise, rate the mask as "Reject".

Please give your rating directly without any explanation. Now let's start. In the given figure, the left half shows a fuchsia box highlighting the region of interest in the original image, and the right half shows a fuchsia mask overlaid on a zoom-in view of that region.
Rate the fuchsia mask for the label "a computer monitor": (A). Accept as text. (B). Flag label. (C). Reject. (D). Accept. (E). Whole image. |
| **Mask Verification Result** | (D). Accept. |

Table 21: **An example data point of Mask Verification (either human or AI verifier).** The AI-verifier is given two crops of the image, a zoomed-out view where the object is highlighted via a box and a zoomed-in view where the mask is highlighted. This allows better visualization of small objects, and avoids color confusion from mask overlay.The AI-verifier instructions are a condensed version of the annotation guidelines given to human annotators. Human annotators are also given options to reject the phrase at the image-level as vague. For AI verifier, we use the model output logits of the choice tokens (e.g., A/B/C/D/E) as the logits for the corresponding labels, and soft-max over the logits to get their probabilities.

| Input | 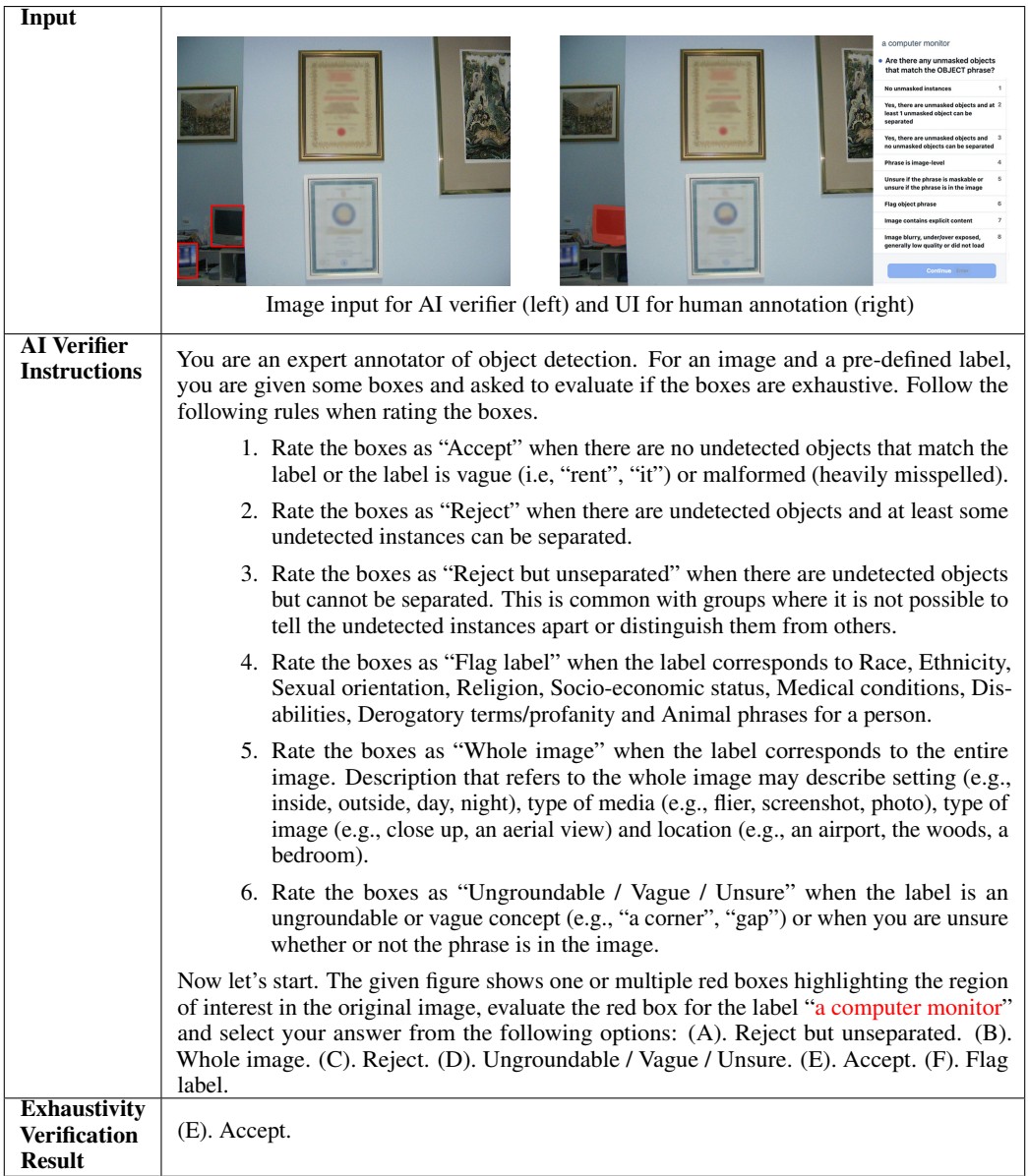 |
|---|---|
| | Image input for AI verifier (left) and UI for human annotation (right) |
| **AI Verifier Instructions** | You are an expert annotator of object detection. For an image and a pre-defined label, you are given some boxes and asked to evaluate if the boxes are exhaustive. Follow the following rules when rating the boxes. 

 1. Rate the boxes as "Accept" when there are no undetected objects that match the label or the label is vague (i.e, "rent", "it") or malformed (heavily misspelled). 

 2. Rate the boxes as "Reject" when there are undetected objects and at least some undetected instances can be separated. 

 3. Rate the boxes as "Reject but unseparated" when there are undetected objects but cannot be separated. This is common with groups where it is not possible to tell the undetected instances apart or distinguish them from others. 

 4. Rate the boxes as "Flag label" when the label corresponds to Race, Ethnicity, Sexual orientation, Religion, Socio-economic status, Medical conditions, Disabilities, Derogatory terms/profanity and Animal phrases for a person. 

 5. Rate the boxes as "Whole image" when the label corresponds to the entire image. Description that refers to the whole image may describe setting (e.g., inside, outside, day, night), type of media (e.g., flier, screenshot, photo), type of image (e.g., close up, an aerial view) and location (e.g., an airport, the woods, a bedroom). 

 6. Rate the boxes as "Ungroundable / Vague / Unsure" when the label is an ungroundable or vague concept (e.g., "a corner", "gap") or when you are unsure whether or not the phrase is in the image. 

 Now let's start. The given figure shows one or multiple red boxes highlighting the region of interest in the original image, evaluate the red box for the label "a computer monitor" and select your answer from the following options: (A). Reject but unseparated. (B). Whole image. (C). Reject. (D). Ungroundable / Vague / Unsure. (E). Accept. (F). Flag label. |
| **Exhaustivity Verification Result** | (E). Accept. |

Table 22: **An example data point of Exhaustivity verification (either human or AI verifier).** For AI verifier, objects to are highlighted via boxes in the image. If there are no candidate objects, the original image is used and "one or multiple red boxes" is replaced by "zero red box" in the text prompt. For AI verifier, we use the model output logits of the choice tokens (e.g., A/B/C/D/E/F) as the logits for the corresponding labels, and soft-max over the logits to get their probabilities. The presence score from the EV AI verifier is defined as 1 - $Prob$(Accept|no boxes as input). $Prob$(Accept|no masks as input) is the probability of Accept (no missing objects) given zero detections as input, which is equivalent to the probability of NO presence.

| | Attributes | | Crowded | | Food&Drinks | | Sports Equip. | | MetaCLIP | | SA-1B | | Wiki-Common | | Average | |
|---|---|---|---|---|---|---|---|---|---|---|---|---|---|---|---|---|
| | MV | EV | MV | EV | MV | EV | MV | EV | MV | EV | MV | EV | MV | EV | MV | EV |
| Human | 72.3 | 81.2 | 72.9 | **82.4** | 76.8 | **76.7** | 79.2 | **87.3** | 72.4 | **79.5** | 72.3 | 73.8 | 79 | **91.5** | 75 | **81.8** |
| AI verifier | **77.1** | **82.6** | **74.6** | 81.3 | **79.4** | 75.1 | **80.1** | 84.7 | **75.3** | 78.8 | **75.9** | **76.8** | **81.3** | 88.4 | **77.7** | 81.1 |

Table 23: Human/AI verifier accuracy on mask verification (MV) and exhaustivity verification (EV) tasks

### E.5 PHASE 3: SCALING AND DOMAIN EXPANSION

**Data Mining.** We continue the data mining approaches from Phase 2 and scale to more novel domains. In addition, we target cases that are rare in web datasets and challenging for the model: crowded scenes with high object counts and images with very small objects. To mine such images, we rely on the SA-1B dataset with mask annotations and compute the "crowdedness" metric i.e. calculate IoU between pair of masks and then aggregate it over all pairs of masks. We also use statistics of the number of masks and mask area to identify images with high object counts and very small objects.

**Proposing NPs.** We continue leveraging the approach from phase 2. We also expand concept coverage to long-tail, fine-grained concepts by extracting NPs from each image's alt-text where available and by mining concepts from the SA-Co ontology.

**Proposing Masks.** Unchanged from Phase 2.

**Verification (Human+AI).** We continue to use both human and AI verifiers as described in Phases 1 and 2 respectively, but primarily rely on AI verifiers to increase the data engine throughput.

**Correction (Human).** We perform manual correction wherever needed, as described in Phase 1. Annotators are asked to correctly mask all occurrences of the given concept in the image.

### E.6 PHASE 4: VIDEO ANNOTATION

In Phase 4, we extend the data engine to video. We use the same high-level stages as the image version, but with video-specific implementation details which are described next.

**Media Pool.** We curate a pool of O(1M) hours of video from SA-V, SA-V internal, YouTube-1B and SA-FARI (wildlife cameras) datasets that covers diverse domains and a range of video quality.

**Data Mining.** To efficiently utilize human annotation resources, we developed aggressive data mining filters and selected only videos that presented the most challenging object tracking scenarios. The mining pipeline finds challenging single-shot video clips that are 5-30s long. Focusing on single-shot clips largely reduces annotation time and ambiguity originating from attempting to track objects across camera shots in edited videos. The mining pipeline consists of the following steps:

- *Scene and Motion Filters.* First, we leverage scene boundary detection and VMAF motion scores from FFmpeg (FFmpeg developers) to identify non-static single-shot camera clips from the video pool. To further improve the precision of single-shot clip selection, we also use Shot Boundary Detection from the PySceneDetect (PySceneDetect Developers) library;

- *Content Balancing.* We use a video-specific ontology to balance content distribution. We build the taxonomy by combining 1) frequent NPs annotated in the image data engine that tend to be associated with higher motion scores, and 2) a taxonomy that emphasizes human activities, animals and transportation. We then generate a set of text queries based on the video ontology and leverage PE Bolya et al. (2025) embeddings to retrieve video candidates for each text query. We propose text queries that elicit grouped objects and crowded scenes, for example "group of dogs" is a text query based on the concept "dog";

- *Challenging Track Filter.* We use an MLLM (PLM (Cho et al., 2025)) as a judge to select videos with challenging tracking scenarios. This is achieved by performing video-QA on a set of questions regarding the existence of various difficult scenarios, and selecting videos that receive more affirmative responses to these questions;

- *Targeted Semantic Search.* Lastly, we enhance the search for challenging scenarios by performing a video similarity search (using PE embeddings) using known challenging videos identified in human annotation as seeds.

**Proposing NPs.** We obtain candidate noun phrases for objects in the video.

- *Frame-level captioner and parser.* We apply the Phase 3 captioner and parser on each video frame, as opposed to video level, to maximize the diversity and volume of candidate noun phrases.

- *NP Filtering*. To keep only relevant phrases, we implement a series of filters. First, we filter out noun phrases that correspond to the overall scene, such as *room*, using a fine-tuned Llama 3.1 model. Similarly, we filter out noun phrases that are too ambiguous to be masked, using the previously trained EV AI Verifier, which has been trained to classify such cases. Next, we remove noun phrases if they are present in a given list of restricted noun phrases. This list contains 1) phrases that have been annotated as non-maskable in previous annotation rounds, 2) phrases for which we already have a lot of annotations, and 3) phrases that correspond to "background" concepts, as our focus is on challenging moving objects. Next, we optionally filter out phrases that do not belong to certain pre-specified super-categories, such as "animal" or "vehicle" to further focus on moving objects. We determine the super-category of a given noun phrase using a Llama 3.1 model.

- *NP Cleaning*. The same cleaning is applied as in previous phases.

**Proposing Masklets.** We use the latest iteration of SAM 3 to generate instance masklets by prompting it with the proposed noun phrases.

- *Masklet Generation.* Initially, we use SAM 3 at the image level to process frames independently, and then propagate the masks using SAM 2. If masks detected in non-propagated frames are not encompassed by the propagated masklets, they are used as starting points for new SAM 2 masklet propagations. Once SAM 3 video performance surpassed the decoupled system, the pipeline was updated to use SAM 3 alone.

- *Masklet Deduplication.* After the masklets are obtained, we deduplicate them based on their IoU.

- *Masklet Filtering.* We filter out the noun phrases that result in masklets containing the whole scene.

- *Filtering Out Easy Cases.* We target challenging multi-object scenarios, namely videos that are relatively crowded and contain multiple objects of the same category. The last step of the pseudo-labeling pipeline filters out all noun phrases with fewer than N=3 objects, and videos that contain fewer than M=2 such noun phrases.

**Verification and Correction (Human).**

- *Verification.* Human annotators check if the video is well pre-processed, e.g., no scene cuts, split screen, or explict content. Then they check if the noun phrase is groundable throughout the video, e.g., there are no comparison or size attributes that might be unclear, and no action attributes which might change across the timeline. Finally, they check that the masklet is challenging to track yet possible to annotate i.e. focus on fast motion and highly occluded objects but which are still identifiable by human annotators and not too blurry to annotate properly.

- *Correction.* Another annotator reviews the proposed masklets, removing those that are incorrect (improving precision), and using online SAM 2 in the loop to correct those that can be improved. Next, they check for any missing masklets, and use SAM 2 to add them if needed (improving recall). This annotation task results in two types of data: fully exhaustive tracking data where every object that matches the noun phrase is annotated, or partially exhaustive tracking data, where some masklets might be missing because they are impossible to annotate (e.g., inseparable background objects that match the noun phrase).

- *Exhaustivity Confirmation.* To ensure data quality, a final round of exhaustivity checking is performed. If there are any remaining missing masklets, they are added as necessary.

**Sampling Frame Annotations.** To increase the diversity and volume of the annotated video data, we also sample video frames and annotate them using the image data engine (Phase 3), where they are treated the same way as other images. The sampling follows two separate strategies. The first one is just random sampling of a frame within a video. This guarantees we cover the distribution of frames. The second strategy consists of first running the video data engine pipeline, and using the results to determine frames that contain many objects.

## F   SA-Co DATASET AND METRIC DETAILS

### F.1   SA-Co TRAINING DATA

SAM 3 training data includes images and videos from many diverse sources, including existing datasets with box or mask annotations. The training data consists of three image datasets and one

video dataset. Fig. 12 visualizes statistics on these subsets in comparison with existing open-source image and video detection and instance segmentation datasets as well as the distribution of top-level SA-Co ontology categories on image datasets. More detailed statistics for each subset and comparison with open-source datasets are shown in Tab. 24 and Tab. 25. The original dataset sources by subset are listed in Tab. 26. .

**SA-Co/HQ: High quality.** This image dataset is generated by the data engine in Phases 1-3 with high quality annotations verified either by human annotators or by AI verifiers that have accuracy on par with humans.

**SA-Co/SYN: Synthetic.** We generate this synthetic dataset via the data engine in Phase 3, relying only on AI annotators. We use MetaCLIP images as the media pool and extract NPs from two sources: 1) alt-text captions associated with the images, 2) captions generated by Llama4. We prompt SAM 3 using the extracted NPs to generate mask proposals. The image-NP-mask proposals are then verified by MV and EV AI verifiers resulting in high-quality synthetic data. We also generate hard negatives proposals (§E.4) and verify them using the EV AI verifier resulting in exhaustive image-level negatives. This scalable system enabled large-scale synthetic data generation, resulting in 39M images, 1.7B image-NPs and 1.4B masks.

**SA-Co/EXT: External.** This dataset includes eighteen external datasets with existing instance mask or bounding boxes annotations. For datasets with only bounding boxes, we generate instance masks with SAM 2. We further enrich these external datasets by mapping the original label to SA-Co ontology and propose additional negative labels using Wikidata hierarchy.

**SA-Co/VIDEO: Video.** The video dataset is collected via the data engine in phase 4 with high quality annotations. All the data in SA-Co/VIDEO is verified by human annotators.

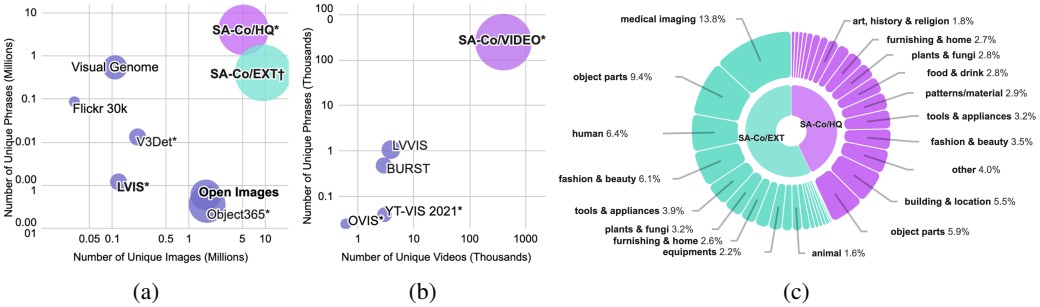

Figure 12: **(a)** SAM 3 image training data statistics and comparison with existing open-source image detection and instance segmentation datasets. Bubble size denotes total number of NP-mask/bbox pairs. Bolded datasets are annotated with masks, others are bboxes only. Datasets with ∗ are exhaustively annotated, datasets with † are partially exhaustively annotated. **(b)** SAM 3 video training data statistics and comparison with existing open-source video instance segmentation datasets. Bubble size denotes total number of NP-masklet pairs. Datasets with ∗ are exhaustively annotated. **(c)** Instance masks distribution among SA-Co ontology top-level categories in SA-Co/HQ and SA-Co/EXT. SA-Co/EXT incorporates several medical and microscopy image datasets, we categorize them under medical imaging in addition to categories in Tab. 20.

| Dataset | # NPs | # Images | # Image-NP | % Negatives | # NP-bbox | # NP-mask | # masks per pair |
|---|---|---|---|---|---|---|---|
| Flickr 30k | 86.4K | 30.1K | 193.0K | - | 312.2K | - | - |
| LVIS∗ | 1.2K | 120.0K | 1.6M | 72.7% | 1.5M | 1.5M | 3.51 |
| V3Det∗ | 13.2K | 213.2K | 737.7K | - | 1.6M | - | - |
| Visual Genome | 542.6K | 108.1K | 4.3M | - | 6.3M | - | - |
| Open Images | 600 | 1.7M | 4.1M | - | 13.3M | 2.7M | 2.79 |
| Object365∗ | 365 | 1.7M | 10.1M | - | 22.9M | - | - |
| **SA-Co/HQ**∗ | 4.0M | 5.2M | 146.1M | 88.5% | 52.3M | 52.3M | 3.10 |
| **SA-Co/EXT**† | 497.4K | 9.3M | 136.6M | 71.8% | 70.5M | 70.5M | 1.83 |
| **SA-Co/SYN**∗ | 38.0M | 39.4M | 1.7B | 74.0% | 1.4B | 1.4B | 3.17 |

Table 24: Detailed statistics for image training datasets and comparison with existing open-source image detection and instance segmentation datasets. Datasets with ∗ are exhaustively annotated, datasets with † are partially exhaustively annotated. % Negatives denotes percentage of Image-NPs that are negatives.

| Dataset | # NPs | # Videos | # Video-NP | % Negatives | # NP-masklet | # masklets per pair |
|---------|-------|----------|------------|-------------|--------------|---------------------|
| OVIS* | 25 | 607 | 886 | - | 3.6K | 4.04 |
| YTVIS 2021* | 40 | 3.0K | 3.9K | - | 6.3K | 1.61 |
| BURST | 482 | 2.9K | 6.9K | - | 16.0K | 2.33 |
| LVVIS | 1.2K | 3.9K | 13.8K | - | 19.7K | 1.40 |
| **SA-Co/VIDEO*** | 24.8K | 52.5K | 134.3K | 26.7% | 467.1K | 4.75 |

Table 25: Detailed statistics for video training dataset and comparison with existing open-source video instance segmentation datasets. Datasets with ∗ are exhaustively annotated. % Negatives denotes percentage of Video-NPs that are negatives.

## F.2 SA-CO EVALUATION BENCHMARK

We create the Segment Anything with Concepts (SA-Co) Benchmark for evaluating promptable concept segmentation (PCS) in images and videos. Our benchmark contains images and videos paired with text labels, each annotated exhaustively with masks on all object instances that match the label. The dataset is federated, meaning that not all labels are annotated for all images, but only a handful of positive and negative labels are verified as ground-truth per image. We add a large volume of challenging hard negative label annotations to test models' ability to handle large, open vocabularies. In particular, the SA-Co/Gold benchmark has $\sim 50\times$ more unique phrases compared to existing exhaustively annotated mask dataset LVIS-test. The SA-Co benchmark covers a diverse array of sub-domains including common objects, fine-grained concepts, food, art, robotics, etc. See Tab. 28 for detailed benchmark statistics and Tab. 27 for the list of sub-domains and their original sources.

In particular, the SA-Co/Gold benchmark has seven sub-domains as shown in Fig. 14 and Tab. 27, created to test different aspects of the concept and image distributions:

- **MetaCLIP** MetaCLIP images (web-scraped) annotated with captioner-proposed noun phrases.

- **SA-1B** SA-1B images (stock photos, more objects per image than MetaCLIP) annotated with captioner-proposed noun phrases.

- **Attributes** MetaCLIP images annotated with attribute phrases. To better test attribute understanding, we also annotate phrases with swapped nouns, e.g., "pink rose"→"pink flamingo", and adjectives, e.g., "pink rose"→"red rose."

- **Crowded Scenes** SA-1B images filtered to select very crowded scenes, annotated with noun phrases proposed by MLLM.

- **Wiki-Common** MetaCLIP images annotated with labels corresponding to 1K nodes from the SA-Co ontology judged to be common by an LLM. These concepts are meant to expand the vocabulary beyond frequent terms like "car", but still be recognizable to non-experts, e.g., "Jeep", "bunk bed", "ballot box."

- **Wiki-Food&Drink** MetaCLIP images annotated with labels corresponding to nodes from the Food&Drink branch of the SA-Co ontology. Many are very fine-grained concepts like "kefir", "pastille".

| SA-Co/HQ | SA-Co/SYN | SA-Co/EXT | SA-Co/VIDEO | SA-Co/VIDEO-EXT |
|----------|-----------|-----------|-------------|-----------------|
| MetaCLIP (Xu et al., 2024b;a) | MetaCLIP | Objects365 (Shao et al., 2019) | SA-V (Ravi et al., 2024) | DAVIS2017 (Pont-Tuset et al., 2017b) |
| SA-1B (Kirillov et al., 2023) | | OpenImages (Kuznetsova et al., 2020) | SA-V internal (Ravi et al., 2024) | MOSEv2 (Ding et al., 2025) |
| Armbench (Mitash et al., 2023) | | ImageNet (Russakovsky et al., 2015) | YT-Temporal-1B (Zellers et al., 2022) | YTVOS2019 (Xu et al., 2018) |
| National Gallery of Art (National Gallery of Art) | | VisualGenome (Krishna et al., 2017) | SA-FARI (Wasmuht et al., 2025) | |
| Ego4d (Grauman et al., 2022) | | Sapiens Body-Parts (Khirodkar et al., 2024) | | |
| MyFoodRepo-273 (Mohanty et al., 2021) | | EDEN (Le et al., 2021) | | |
| GeoDE (Ramaswamy et al., 2023) | | Fashionpedia (Jia et al., 2020) | | |
| DROID (Khazatsky et al., 2024) | | Fathomnet (Katija et al., 2021) | | |
| BDD100k (Yu et al., 2020) | | iNaturalist-2017 (Horn et al., 2017) | | |
| SA-V (Ravi et al., 2024) | | BDD100k (Yu et al., 2020) | | |
| SA-V internal (Ravi et al., 2024) | | Livecell (Edlund et al., 2021) | | |
| YT-Temporal-1B (Zellers et al., 2022) | | PanNuke (Gamper et al., 2019; 2020) | | |
| | | MedSAM2 (Ma et al., 2025) | | |
| | | SNOW (Ding et al., 2023) | | |
| | | Visdrone (Zhu et al., 2021) | | |
| | | WCS Camera Traps (LILA BC) | | |
| | | HierText (Long et al., 2023; 2022) | | |
| | | FSC-147 (Ranjan et al., 2021) | | |

Table 26: Media pool used to construct each SA-Co train subset. See Figs. 13 and 15 to 18 for examples of each domain and annotations.

- **Wiki-Sports Equipment** MetaCLIP images annotated with labels corresponding to nodes from the Sports Equipment branch of the SA-Co ontology, with many fine-grained concepts like "kettlebell."

All of the above sub-domains also have a high number of hard negative annotations, see Tab. 28.

| SA-Co/Gold | SA-Co/Silver | SA-Co/Bronze | SA-Co/Bio | SA-Co/VEval |
|---|---|---|---|---|
| MetaCLIP | BDD100k | BDD100k | Livecell | SA-V |
| SA-1B | DROID | EDEN | PanNuke | YT-Temporal-1B |
| Attributes (MetaCLIP) | Ego4D | Fashionpedia | MedSAM2 | SmartGlasses |
| Crowded Scenes (SA-1B) | MyFoodRepo-273 | Visdrone | SNOW | |
| Wiki-Common (MetaCLIP) | GeoDE | WCS Camera Traps | | |
| Wiki-Food&Drink (MetaCLIP) | iNaturalist-2017 | | | |
| Wiki-Sports Equipment (MetaCLIP) | National Gallery of Art | | | |
| | SA-V | | | |
| | YT-Temporal-1B | | | |
| | Fathomnet | | | |

Table 27: Domains in each SA-Co evaluation benchmark subset. See Figs. 13, 14 and 16 to 18 for examples of each domain and annotations.

| Dataset | # NPs | # Images | # Image-NP | % Negatives | # NP-masks | % 0-shot NPs |
|---|---|---|---|---|---|---|
| LVIS test | 1.2K | 19.8K | - | - | - | - |
| COCO test2017 | 80 | 40.7K | - | - | - | - |
| ODinW-35 test | 290 | 15.6K | 26.1K | - | 131.1K | - |
| **SA-Co/Gold** | 51.8K | 15.8K | 168.9K | 84.4% | 126.9K | 6.98% |
| **SA-Co/Silver** | 54.6K | 66.1K | 1.8M | 94.0% | 219.8K | 8.00% |
| **SA-Co/Bronze** | 105.3K | 32.5K | 1.0M | 84.9% | 261.5K | 57.25% |
| **SA-Co/Bio** | 166 | 5.4K | 35.9K | 71.8% | 264.6K | - |

(a)

| Dataset | # NPs | # Videos | # Video-NP | % Negatives | # NP-masklets | % 0-shot NPs |
|---|---|---|---|---|---|---|
| LVVIS test | 1.2K | 908 | - | - | 5.7K | - |
| BURST test | 482 | 1.4K | 3.4K | - | 8.0K | - |
| **SA-Co/VEval** | 5.2K | 1.7K | 10.3K | 75.4% | 11.2K | 6.37% |

(b)

Table 28: **(a)** Summary statistics of SA-Co image evaluation benchmark by subsets and comparison with existing image instance segmentation benchmarks. **(b)** Summary statistics of SA-Co/VEval benchmark and comparison video instance segmentation benchmarks. % Negatives denotes percentage of Image-NPs or Video-NPs that are negative. Percentages of zero-shot NPs in each subset are reported. A zero-shot NP is defined as a phrase that has not been seen in the combined set of SA-Co/HQ, SA-Co/EXT and SA-Co/VIDEO.

### F.3 METRICS

We introduce the *classification-gated F1* ($cgF_1$) to evaluate the PCS task on images. The traditional AP (Average Precision) metric designed for closed-vocabulary detection tasks (e.g., COCO), breaks down when applied to open-vocabulary detection with very large label spaces. While averaging AP over 80 classes is feasible, with tens of thousands most appear just once in the test set and the average is dominated by noise. Computing full precision-recall curves for all labels is also computationally infeasible and unnecessary for practical use cases. AP also does not account for the model calibration, which means that high-scoring models can be difficult to use in practice. F1 at a fixed confidence threshold presents a good alternative, however it is sensitive to high ratios of negative annotations: no extra credit is given for correctly predicting nothing for a negative, but the score is lowered by predicting false positives.

To remedy these issues we design new metrics for the PCS task. Given datapoints consisting of predicted and ground truth (media, phrase, masks) triplets we compute the following metrics to measure localization and classification separately:

- *Localization.* We measure this only on *positive* datapoints with at least one ground-truth mask. For one sample, assume we have $N$ predicted masks $m_1, \cdots, m_N$ and $M$ ground-truth masks

$\hat{m}_1, \cdots, \hat{m}_M$. We compute the IoU matrix $\text{iou}_{i,j} = \text{iou}(m_i, \hat{m}_j)$, then deduce the optimal bipartite matching $\hat{\sigma} = \text{argmax}_\sigma \sum_i \text{iou}_{i,\sigma(i)}$. We fix an IoU threshold $\tau$, then for every prediction $i$, if it is matched and $\text{iou}_{i,\sigma(i)} \geq \tau$, then it is counted as TP (true positive), otherwise FP (false positive). Unmatched ground truths are counted as FN (false negative). We compute $F_1^\tau = \frac{2\text{TP}^\tau}{2\text{TP}^\tau + \text{FP}^\tau + \text{FN}^\tau}$ for each datapoint, known as the *local F1 score*. We accumulate the counts of TP, FP and FN over all data points with at least one groundtruth mask, and compute "positive micro F1" score $\text{pmF}_1^\tau$. We compute $\text{pmF}_1^\tau$ for all $\tau \in [0.5, 0.95]$ with increments of $0.05$, then average to obtain the final $\text{pmF}_1$:

$$\text{pmF}_1^\tau = \frac{2\text{TP}_{total}^\tau}{2\text{TP}_{total}^\tau + \text{FP}_{total}^\tau + \text{FN}_{total}^\tau}, \quad pmF_1 = \frac{1}{10} \sum_{\tau \in \text{np.linspace}(0.5, 0.95, 10)} \text{pmF}_1^\tau. \quad (1)$$

We also compute the average of the local F1 scores over all data points with at least one groundtruth mask, and obtain the "positive macro F1" score. We report both the positive micro and macro F1 scores in our score chart, and choose the positive micro score $\text{pmF}_1$ as the main metric for localization.

- *Classification.* This metric between $[-1, 1]$ computes the ability of the model to predict one or several masks, *if and only if* the datapoint is positive. This can be seen as a binary prediction task at the image level ("is the object present or not?"), and crucially, in this metric we do not care about the quality of the predicted masks. If the datapoint is *positive*, and if the model has predicted *any* mask (with confidence greater than 0.5), then it is an IL_TP (image level TP), otherwise an IL_FN. If the datapoint is *negative*, and if the model has predicted *any* mask, then it is an IL_FP, otherwise an IL_TN. We summarize this confusion matrix into a single metric, and measure potential imbalances with the Matthews Correlation Coefficient (MCC) as:

$$\text{IL\_MCC} = \frac{\text{IL\_TP} \cdot \text{IL\_TN} - \text{IL\_FP} \cdot \text{IL\_FN}}{\sqrt{(\text{IL\_TP} + \text{IL\_FP}) \cdot (\text{IL\_TP} + \text{IL\_FN}) \cdot (\text{IL\_TN} + \text{IL\_FP}) \cdot (\text{IL\_TN} + \text{IL\_FN})}}. \quad (2)$$

As our main metric, we combine these two metrics to compute $\text{cgF}_1$ ("classification-gated F1"), defined as

$$\text{cgF}_1 = 100 \cdot \text{pmF}_1 \cdot \text{IL\_MCC}. \quad (3)$$

The PCS task is quite ambiguous in many cases, and to alleviate this issue our SA-Co/Gold subset contains three independent ground-truth annotations for each datapoint. To adapt our metric, we use an *oracle* setting, where we compare the model's predictions to each ground-truth for each datapoint, and select the one that yields the best local F1 score.

## F.4 Human Performance on SA-Co

As described in §2, the PCS task is intrinsically ambiguous. Given an image-NP or video-NP pair, even trained annotators can have different interpretations that are all valid. When the phrase is vague, annotators can even disagree on the presence of the NP. Hence when evaluating on the SA-Co benchmark, disagreement with ground truth does not necessarily mean the prediction is wrong. To this end, it is important to study the human-level performance (i.e. the agreement among skilled annotators) on the PCS task to facilitate interpreting model performance.

**Human Performance on SA-Co/Gold.** On the image benchmark, we provide three sets of annotations by different annotators. Fig. 14 shows examples of the three independent annotations per (phrase, image) pair for each domain in the benchmark. These annotations are done from scratch, meaning that the annotators create masks (using SAM 1) without seeing any SAM 3 model interpretations. We define the "oracle" metric as follows to measure the upper bound of human performance. For each image-NP, the best pair (out of all three pairs of annotations) is selected by maximizing the local F1 score or minimizing the sum of false negatives (FN) and false positives (FP) when there is a tie in local F1 scores. We then report the $\text{cgF}_1$ metric based on these selected best pairs using one annotation as ground truth and the other as prediction. To make the model performance comparable, the "oracle" model performance is calculated by comparing model predictions to all three annotations and selecting the best pairs.

Alternative to the "Oracle" protocol, human performance can also be measured on randomly selected pairs. Specifically, we adopt the following protocol to compute "Random Pair" human performance

| Evaluation Protocol | SA-Co/Gold | | | SA-Co/VEval | | | | | | | |
| | 3-Reviewed Subset | | | YT-Temporal-1B test | | | | SmartGlasses test | | | |
| | $cgF_1$ | IL_MCC | $pmF_1$ | $cgF_1$ | pHOTA | pDetA | pAssA | $cgF_1$ | pHOTA | pDetA | pAssA |
| Oracle | 76.2 | 0.99 | 77.0 | 85.1 | 88.4 | 84.7 | 92.6 | 73.5 | 83.1 | 75.4 | 92.0 |
| Random Pair | 55.5 | 0.87 | 63.4 | 75.9 | 82.0 | 73.8 | 91.3 | 53.4 | 70.2 | 54.9 | 90.2 |

Table 29: Human instance segmentation performance comparison between the Oracle and Random Pair protocols on SA-Co benchmark. The comparison is shown on subsets of the benchmark with three human annotations.

on SA-Co benchmark with three sets of annotations: 1) randomly choosing a pair of annotations for each image/video-NP, then aggregate over all image/video-NPs to get the metric values, 2) repeating the process a thousand times and reporting the 0.5 quantile for each metric. As shown in Tab. 29, there is a noticeable gap between Oracle and Random Pair performance on both image and video benchmarks, suggesting that the PCS task is inherently ambiguous.

The image benchmark has a large portion of hard negatives. These phrases go through human verification, but as it is costly to collect three sets of human annotations on the entire dataset due to the large volume, the negative noun phrases only have one ground-truth label. The human performance on these phrases is estimated by collecting additional human annotations on a subsample of phrases and comparing them with the initial annotation (i.e., the ground truth). Specifically, we collect additional human annotations on about one thousand image-NPs for each domain in SA-Co/Gold. Since the ground truths are all negatives, these phrases only contribute to the IL_MCC metric. We compute counts of IL_TN and IL_FP on these samples, and then extrapolate these results to estimate the corresponding counts for the entire set of hard negatives. These estimated counts are then combined with image-level counts from the rest of the benchmark where NPs have three annotations to get the final IL_MCC.

Typically, our annotation protocol allows annotators to mark NPs as ambiguous if they are unsure. In this additional human review for the hard negatives, we remove the unsure option and prompt them to make a choice between positive and negative, thus reduce uncertainty and potential bias that could arise from ambiguous data.

**Human Performance on SA-Co/VEval.** Annotating videos is much more expensive than static images, so we collect only one set of annotations per NP on the video benchmark. To guarantee annotation quality, these ground-truth annotations undergo multiple rounds of human refinement. To measure human performance in a way that is directly comparable to model evaluation in video PCS, we collect one additional from-scratch human annotation for every NP in the test set across all sub-domains. Human performance on video PCS task is then reported by comparing this additional annotation to the ground truth, using the same metrics as for model evaluation ($cgF_1$ and pHOTA).

Additionally, to study the gap between the Random Pair and the Oracle protocols, we collect two further human annotations (for a total of three) on the YT-Temporal-1B and SmartGlasses test splits of the SA-Co/VEval dataset. This allows us to verify that the gap observed in the image domain also exists in the video domain (see Tab. 29).

## F.5 ADDITIONAL DATASET EXAMPLES

Figs. 13 and 15 to 18 show examples of each visual domain in our image and video datasets. Fig. 14 illustrates the domains in our SA-Co/Gold evaluation benchmark and the three independent annotations per sample. Fig. 19 show an example image from our synthetic data set SA-Co/SYN, with its positive noun phrases in the figure and negative noun phrases in the caption.

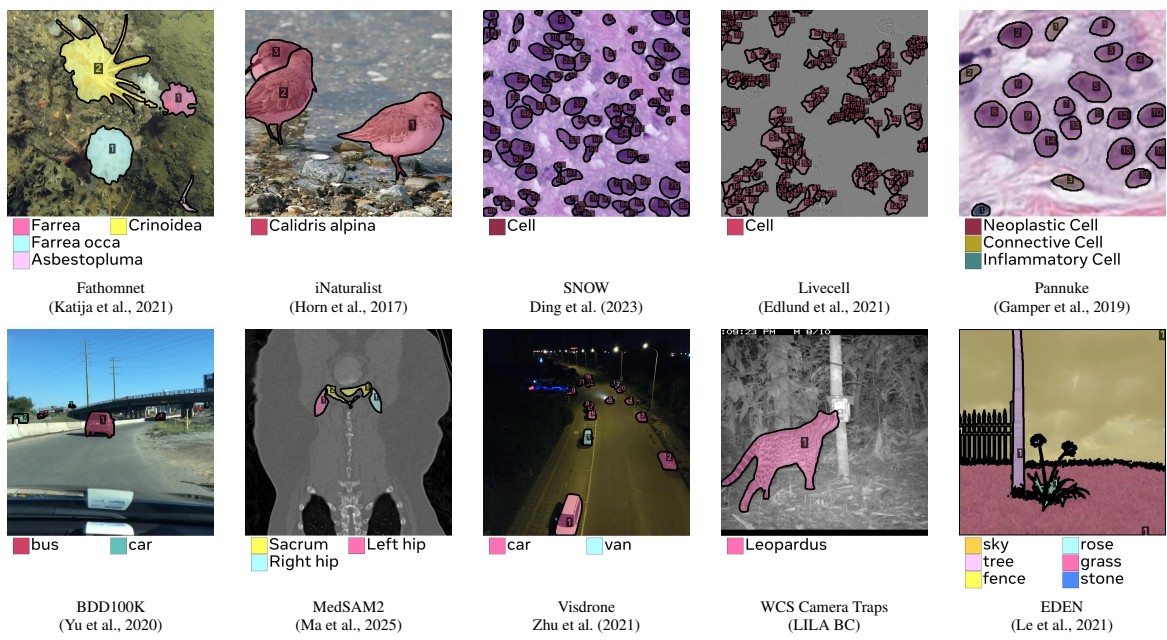

Figure 13: Per-domain examples in SA-Co/EXT. Shown with annotated phrases and masks overlaid.

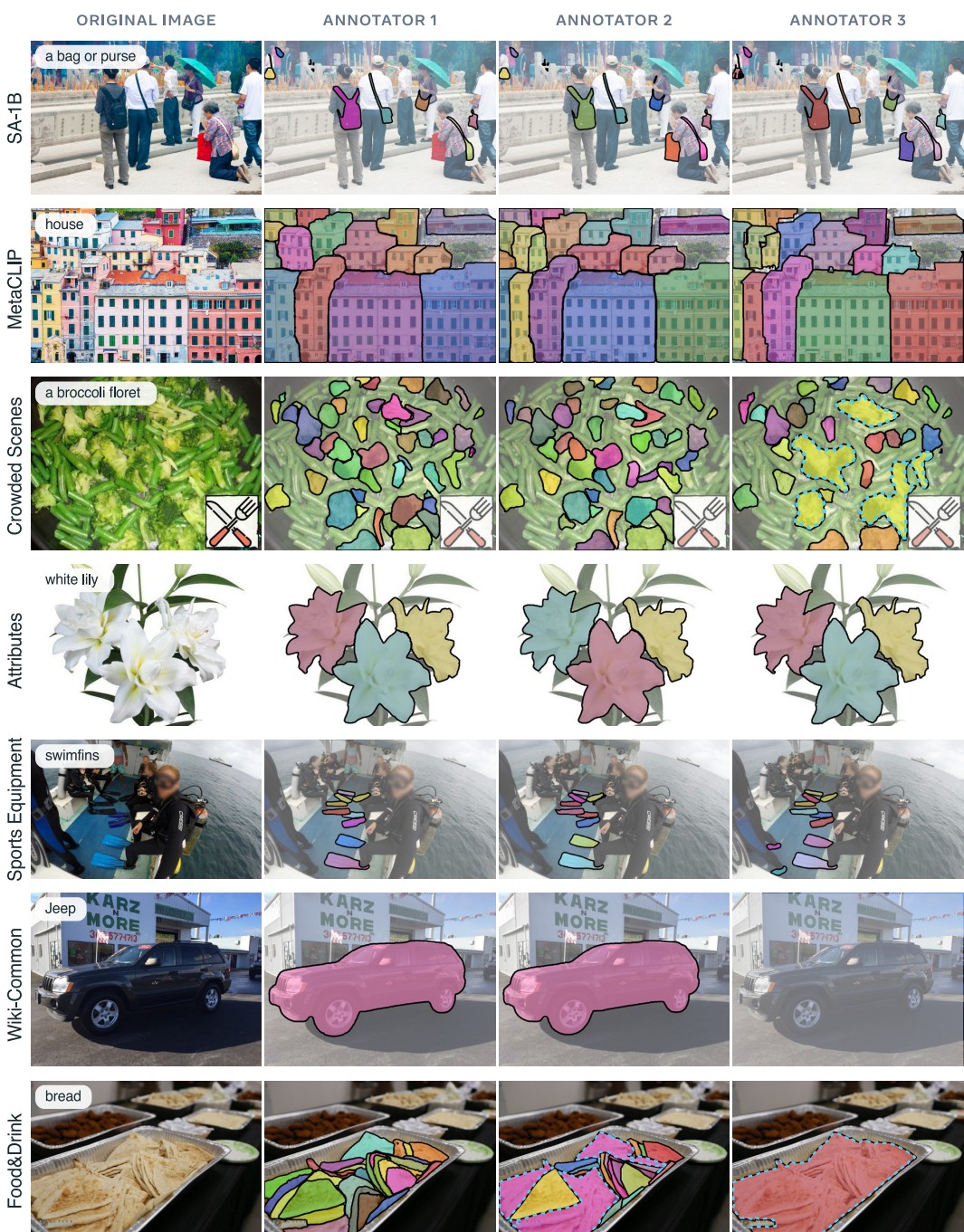

Figure 14: Annotations in our SA-Co/Gold benchmark. Each row shows an example (image, noun phrase) pair from one of the seven domains, with masks from three independent annotators overlayed. No mask means the annotator decided that the phrase is not in the image. Dashed borders indicate special group masks that cover more than a single instance, used when separating into instances is deemed difficult. Annotators sometimes disagree on precise mask borders, the number of instances (e.g., house, broccoli, swimfins) or on whether the phrase applies (Jeep). We use the 3 GT annotations to measure the human performance on the task, to serve as an upper bound for model performance.

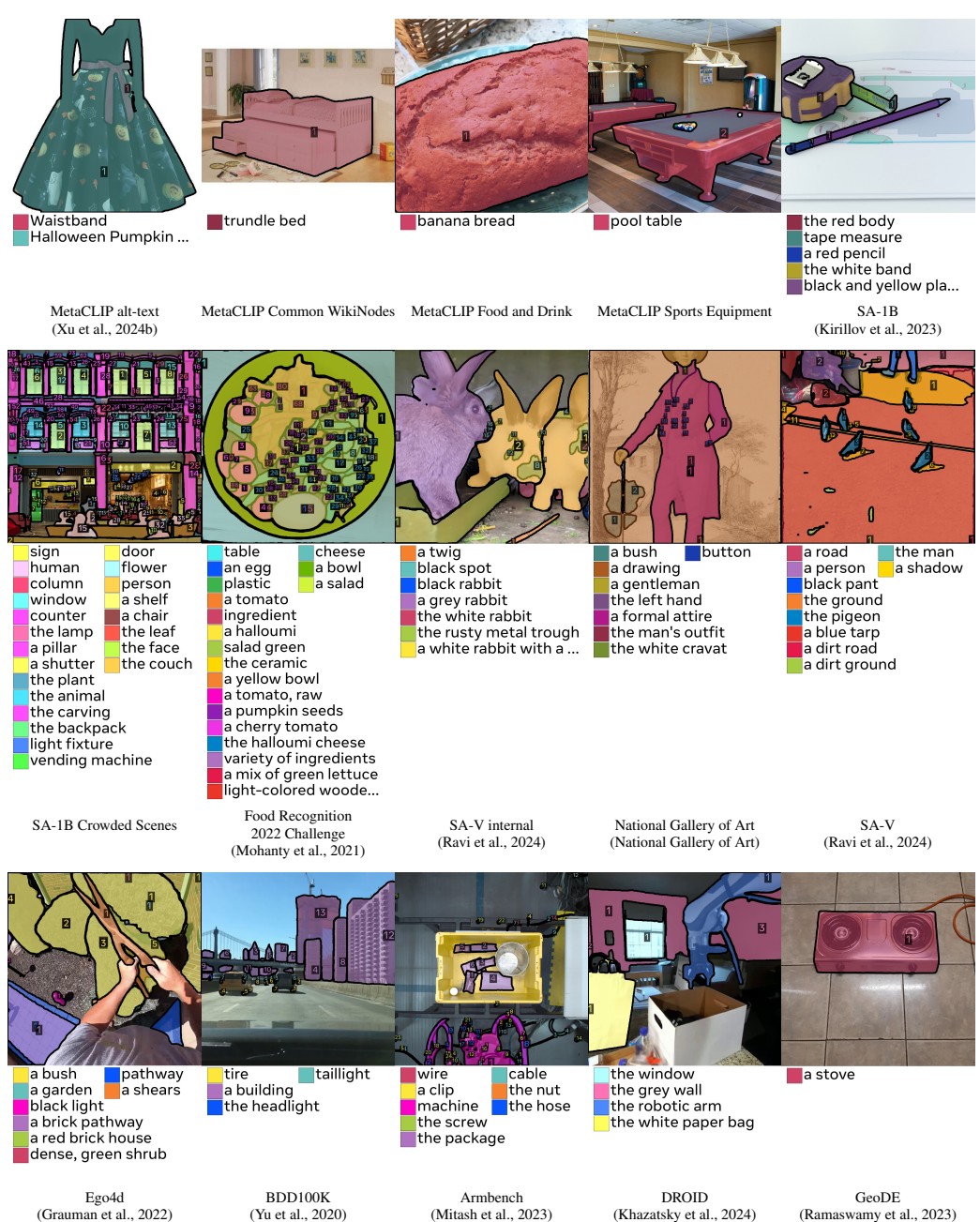

Figure 15: Per-domain examples in SA-Co/HQ. Shown with annotated phrases and masks overlaid.

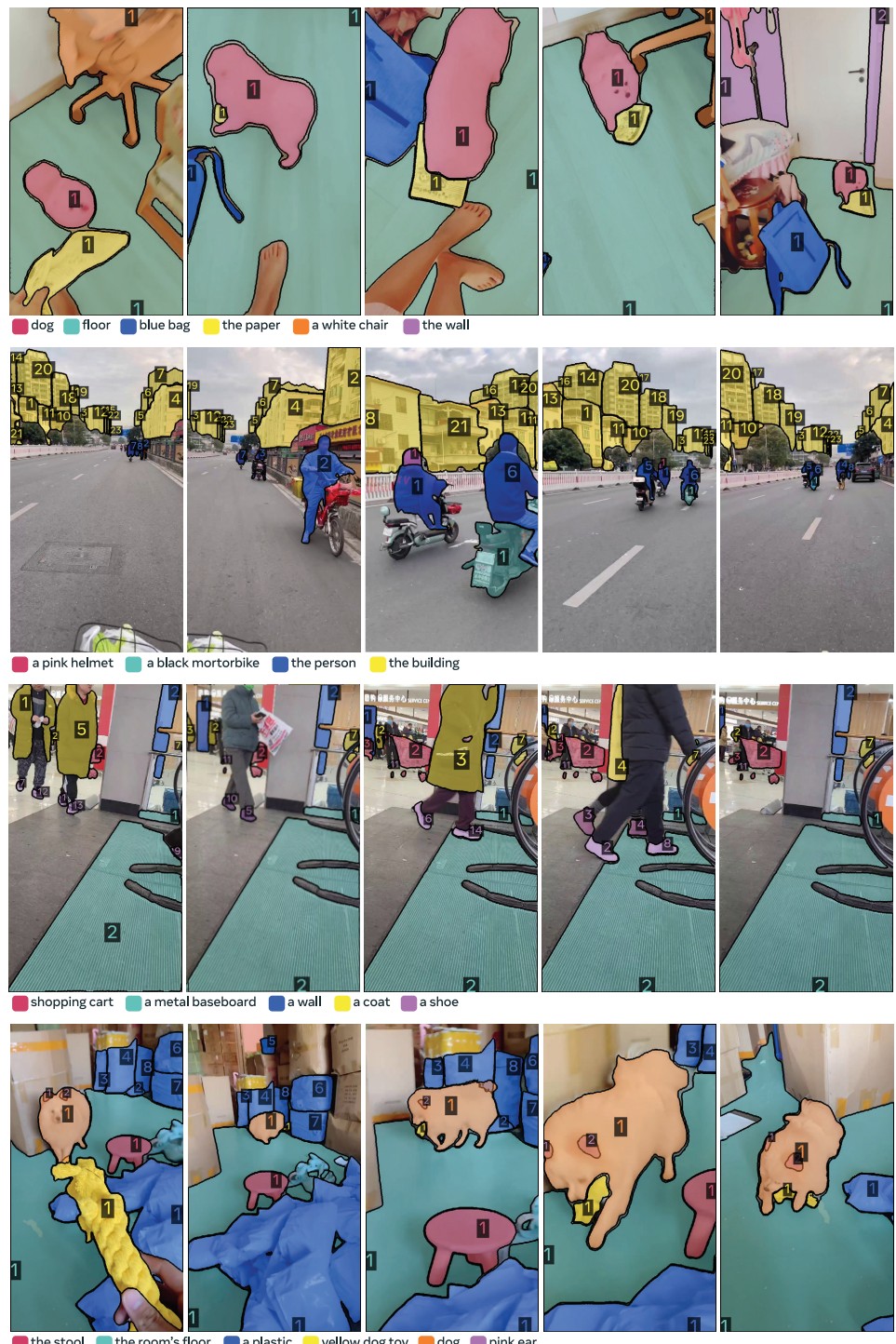

Figure 16: Example annotations from the SA-V media in the SA-Co/VIDEO and SA-Co/VEval datasets.

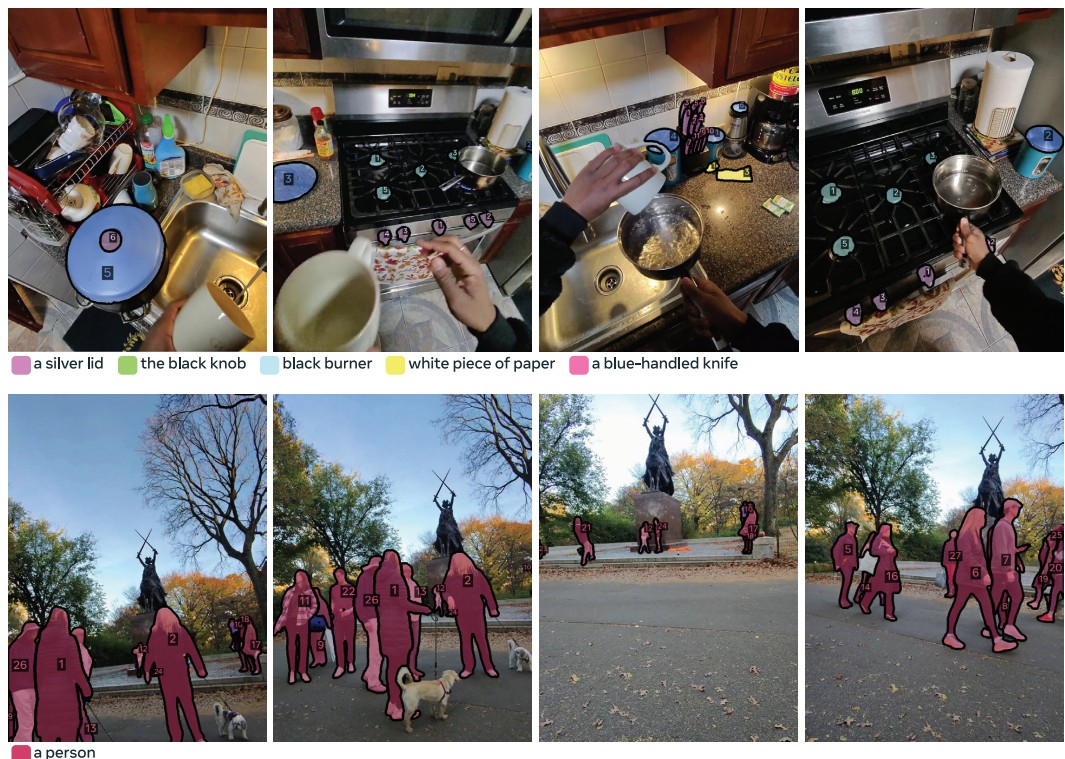

Figure 17: Example annotations from the SmartGlasses media in the SA-Co/VEval dataset.

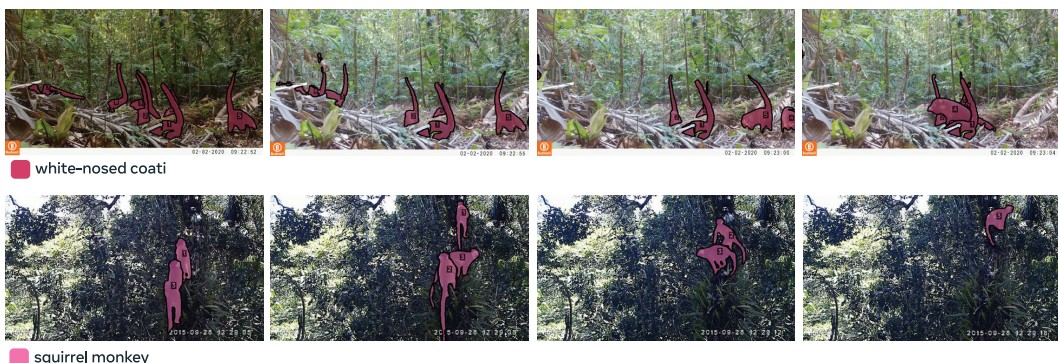

Figure 18: Example annotations from the SA-FARI dataset.

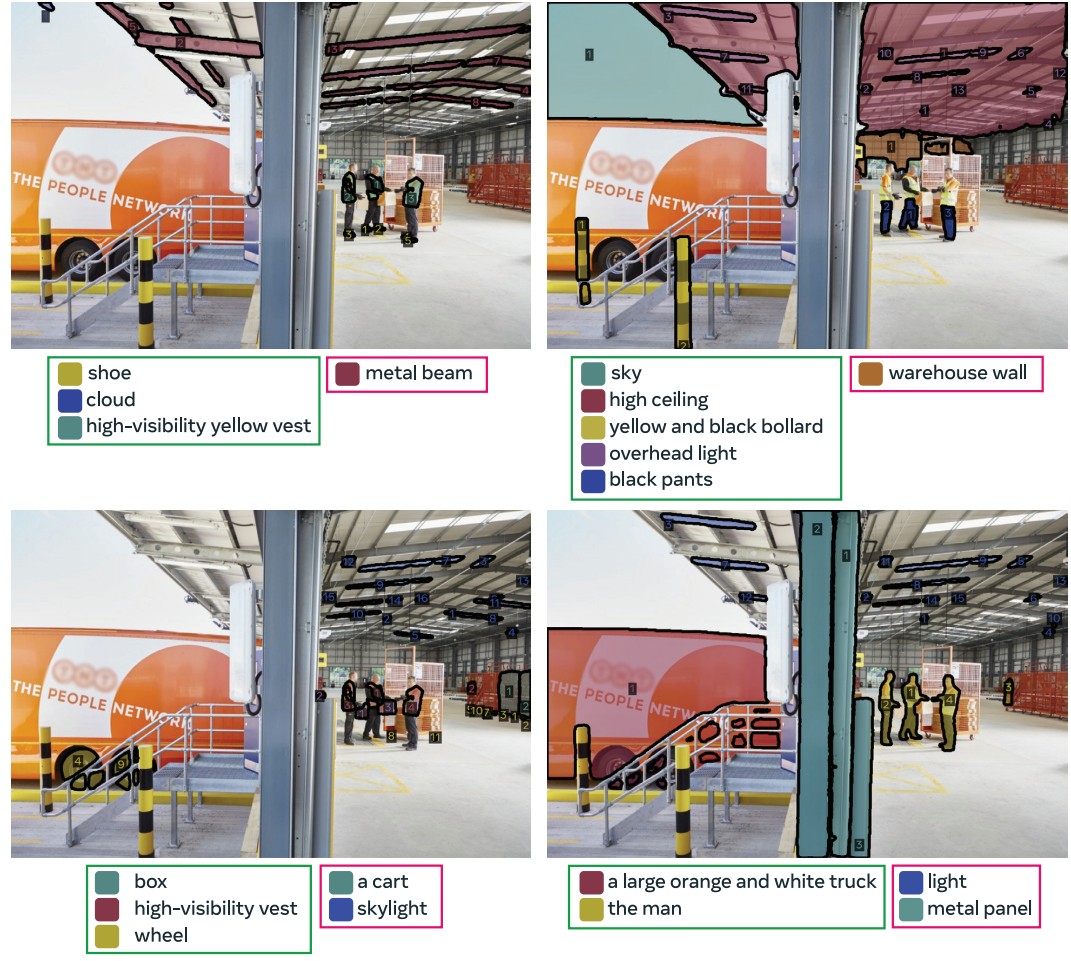

Figure 19: Example annotations on one image from the SA-Co/SYN dataset. There are 19 positive noun phrases for this image, which we visualize in 4 groups for better visualization quality. AI verifier assigned an exhaustivity label for each noun phrase: noun phrases in green box means the masks for that noun phrase is exhaustive, noun phrases in pink box means not exhaustive. There are 20 negative noun phrases that are verified and confirmed by AI verifier for this image: *a black and white post, a blue and yellow post, a large red and white truck, an orange and black post, large orange and white boat, the green and red post, the large orange and white motorcycle, the orange and white bollard, the purple and yellow post, yellow sash, the Malaysian Air Boeing 737-800, display vehicle, Zamboni ice resurfacer, boot, the small folded paper package, long yellow vehicle, the small gift-wrapped package, small trailer, bunker gear, two-seat model.*

# G  ADDITIONAL EXPERIMENTS AND DETAILS

## G.1  PCS WITH NPS ON IMAGES

This section describes the experiments in Tab. 1 in detail. We compare to OWLv2 (Minderer et al., 2024), GroundingDino (Liu et al., 2024a) and LLMDet (Fu et al., 2025). Since they produce only bounding boxes, we convert them to masks using SAM 1 to evaluate segmentation. We also compare to APE (Shen et al., 2024) and DINO-X (Ren et al., 2025), two state-of-the-art segmentation models, and finally Gemini 2.5 Flash (Comanici et al., 2025), a generalist LLM.

We report performance on LVIS (Gupta et al., 2019), COCO (Lin et al., 2014), COCO-O (Mao et al., 2023), Cityscapes (Cordts et al., 2016) (val set), ADE (Zhou et al., 2019), and Pascal Context (Mottaghi et al., 2014), reporting their official metrics. For LVIS, we report AP-fixed (Dave et al., 2022). On our new SA-Co benchmark, we report the average across every split. We report $cgF_1$, except for SA-Co/Bio where we report $pmF_1$ (this split does not have negatives, so only localization is meaningful). On SA-Co/Gold we have three ground-truth annotations per datapoint, so we report the oracle metric and estimated human performance (human performance measurement detailed in §F.4). To evaluate semantic segmentation using SAM 3, we predict instance masks for each semantic category and filtering the predictions using the presence scores, mask scores, and mask areas to create the semantic mask per image. In Tab. 34, we include additional semantic segmentation evaluation on ADE-150 (Zhou et al., 2019) and PC-459 (Mottaghi et al., 2014).

We employ the following Hugging Face model checkpoints: "google/owlv2-large-patch14" for OWLv2, "google/owlv2-large-patch14-ensemble" for OWLv2*, "IDEA-Research/grounding-dino-tiny" for gDino-T, and "iSEE-Laboratory/llmdet_large" for LLMDet-L. OWLv2* utilizes an ensemble of checkpoint weights after self-training and after fine-tuning the model on LVIS base, demonstrating improved open-world generalization compared to fine-tuning alone (Minderer et al., 2024). We provide per-domain performance of instance segmentation for all baselines, SAM 3, and human on SA-Co/Gold in Tab. 30 and on SA-Co/Silver in Tab. 31 and Tab. 32. We also include per-domain performance for the AI verifier ablation study in Tab. 9d. In Tab. 33, we compare with additional baselines using "IDEA-Research/grounding-dino-base" for gDino-B and "iSEE-Laboratory/llmdet_base" for LLMDet-B.

For OWLv2, GroundingDino, and LLMDet, we swept over the detection threshold at 0.1 intervals and determined the best threshold using the LVIS $cgF_1$ metric for the box detection task. Then, we applied this threshold to compute $cgF_1$ on the remaining datasets for the box detection and instance segmentation tasks. The detection threshold is set to 0.4 for LLMDet-L, LLMDet-B, and gDino-T; 0.3 for OWLv2*; and 0.2 for OWLv2 and gDino-B. For DINO-X, we find the detection threshold 0.5 gives the best $cgF_1$ metric. Additionally, we found that prompting multiple noun phrases at once for a given image greatly improved performance for GroundingDino and LLMDet, compared to prompting one noun phrase at a time. For example, we prompted GroundingDino and LLMDet with 30 prompts for SA-Co/Gold and 20 prompts for SA-Co/Silver, SA-Co/Bronze, and SA-Co/Bio.

For Gemini 2.5 Flash, we run inference via the Gemini API. For each (image, text query) pair, we prompt Gemini 2.5 using the same prompt template that is used in Gemini Flash 2.5 image segmentation demo (Paul Voigtlaender, Valentin Gabeur and Rohan Doshi, 2025) with the same generation settings. In addition, we prompt the model multiple times if there are any errors in generation, or parsing the result into a set of masks and bounding boxes.

| | Average | | | Metaclip | | | SA-1B | | | Crowded | | | Food&Drink | | | Sports Equip. | | | Attributes | | | Wiki-Common | | |
|---|---|---|---|---|---|---|---|---|---|---|---|---|---|---|---|---|---|---|---|---|---|---|---|---|
| | cgF₁ | IL-MCC | pmF₁ | cgF₁ | IL-MCC | pmF₁ | cgF₁ | IL-MCC | pmF₁ | cgF₁ | IL-MCC | pmF₁ | cgF₁ | IL-MCC | pmF₁ | cgF₁ | IL-MCC | pmF₁ | cgF₁ | IL-MCC | pmF₁ | cgF₁ | IL-MCC | pmF₁ |
| gDino-T | 3.3 | 0.15 | 16.2 | 2.9 | 0.21 | 13.9 | 3.1 | 0.20 | 15.4 | 0.28 | 0.08 | 3.4 | 0.96 | 0.10 | 9.8 | 1.1 | 0.10 | 11.2 | 13.8 | 0.29 | 47.3 | 0.70 | 0.06 | 12.1 |
| OWLv2* | 24.6 | 0.57 | 42.0 | 17.7 | 0.52 | 34.3 | 13.3 | 0.50 | 26.8 | 15.8 | 0.51 | 30.7 | 32.0 | 0.65 | 49.4 | 36.0 | 0.64 | 56.2 | 35.6 | 0.63 | 56.2 | 21.7 | 0.54 | 40.3 |
| OWLv2 | 17.3 | 0.46 | 36.8 | 12.2 | 0.39 | 31.3 | 9.8 | 0.45 | 21.7 | 8.9 | 0.36 | 24.8 | 24.4 | 0.51 | 47.9 | 24.4 | 0.52 | 47.0 | 25.9 | 0.54 | 48.2 | 15.4 | 0.42 | 36.6 |
| LLMDet-L | 6.5 | 0.21 | 27.3 | 4.5 | 0.23 | 19.4 | 5.3 | 0.23 | 22.8 | 2.4 | 0.18 | 13.7 | 5.5 | 0.19 | 29.1 | 4.4 | 0.17 | 25.3 | 22.2 | 0.39 | 57.1 | 1.2 | 0.05 | 23.3 |
| APE-D | 16.4 | 0.40 | 36.9 | 12.6 | 0.42 | 30.1 | 2.2 | 0.22 | 10.0 | 7.2 | 0.35 | 20.3 | 22.7 | 0.51 | 45.0 | 31.8 | 0.56 | 56.5 | 26.7 | 0.47 | 57.3 | 11.6 | 0.29 | 39.5 |
| DINO-X | 21.3 | 0.38 | 55.2 | 17.2 | 0.35 | 49.2 | 19.7 | 0.48 | 40.9 | 12.9 | 0.34 | 37.5 | 30.1 | 0.49 | 61.7 | 28.4 | 0.41 | 69.4 | 31.0 | 0.42 | 74.0 | 9.7 | 0.18 | 53.5 |
| Gemini 2.5 | 13.0 | 0.29 | 46.1 | 9.9 | 0.29 | 33.8 | 13.1 | 0.41 | 32.1 | 8.2 | 0.27 | 30.3 | 19.6 | 0.33 | 59.5 | 15.1 | 0.28 | 53.5 | 18.8 | 0.30 | 63.1 | 6.5 | 0.13 | 50.3 |
| SAM 3 | 54.1 | 0.82 | 66.1 | 47.3 | 0.81 | 58.6 | 53.7 | 0.86 | 62.6 | 61.1 | 0.9 | 67.7 | 53.4 | 0.79 | 67.3 | 65.5 | 0.89 | 73.8 | 54.9 | 0.76 | 72.0 | 42.5 | 0.70 | 60.9 |
| Human | 72.8 | 0.94 | 77.0 | 64.1 | 0.94 | 68.5 | 64.3 | 0.97 | 66.6 | 70.4 | 0.94 | 75.3 | 78.3 | 0.96 | 81.2 | 80.4 | 0.97 | 83.1 | 80.2 | 0.95 | 84.4 | 71.6 | 0.89 | 80.1 |

Table 30: Per-domain results of instance segmentation on SA-Co/Gold. ⋆: partially trained on LVIS.

| | Average | | | BDD100k | | | DROID | | | Ego4D | | | MyFoodRepo-273 | | | GeoDE | | |
|---|---|---|---|---|---|---|---|---|---|---|---|---|---|---|---|---|---|---|
| | $cgF_1$ | IL_MCC | $pmF_1$ | $cgF_1$ | IL_MCC | $pmF_1$ | $cgF_1$ | IL_MCC | $pmF_1$ | $cgF_1$ | IL_MCC | $pmF_1$ | $cgF_1$ | IL_MCC | $pmF_1$ | $cgF_1$ | IL_MCC | $pmF_1$ |
| gDino-T | 2.7 | 0.12 | 16.6 | 2.2 | 0.17 | 12.7 | 3.6 | 0.15 | 23.8 | 2.4 | 0.10 | 23.6 | 0.52 | 0.05 | 10.3 | 10.5 | 0.24 | 43.8 |
| OWLv2* | 11.5 | 0.32 | 33.0 | 15.0 | 0.46 | 32.6 | 11.1 | 0.36 | 30.7 | 7.6 | 0.23 | 33.0 | 20.0 | 0.44 | 45.4 | 27.6 | 0.50 | 55.2 |
| OWLv2 | 7.6 | 0.23 | 31.1 | 7.6 | 0.31 | 24.7 | 7.3 | 0.25 | 29.0 | 5.5 | 0.18 | 30.7 | 13.4 | 0.32 | 41.8 | 12.9 | 0.28 | 46.2 |
| LLMDet-L | 7.1 | 0.17 | 28.7 | 1.5 | 0.08 | 17.1 | 2.3 | 0.10 | 23.2 | 2.1 | 0.08 | 26.6 | 0.90 | 0.06 | 15.0 | 21.0 | 0.37 | 56.8 |
| APE-D | 7.3 | 0.24 | 24.5 | 6.7 | 0.32 | 20.9 | 8.9 | 0.30 | 29.6 | 6.3 | 0.23 | 27.9 | 7.0 | 0.26 | 26.5 | 26.9 | 0.47 | 57.5 |
| Gemini 2.5 | 8.3 | 0.19 | 38.1 | 5.1 | 0.19 | 26.9 | 4.5 | 0.14 | 31.8 | 0.32 | 0.01 | 32.4 | 8.6 | 0.24 | 35.9 | 16.4 | 0.26 | 62.9 |
| SAM 3 | 49.6 | 0.76 | 65.2 | 46.6 | 0.78 | 60.1 | 45.6 | 0.76 | 60.4 | 38.6 | 0.62 | 62.6 | 53.0 | 0.79 | 67.2 | 70.1 | 0.89 | 78.7 |

Table 31: Per-domain results of instance segmentation on SA-Co/Silver. ⋆: partially trained on LVIS.

| | iNaturalist-2017 | | | National Gallery of Art | | | SA-V | | | YT-Temporal-1B | | | Fathomnet | | |
|---|---|---|---|---|---|---|---|---|---|---|---|---|---|---|---|
| | $cgF_1$ | IL_MCC | $pmF_1$ | $cgF_1$ | IL_MCC | $pmF_1$ | $cgF_1$ | IL_MCC | $pmF_1$ | $cgF_1$ | IL_MCC | $pmF_1$ | $cgF_1$ | IL_MCC | $pmF_1$ |
| gDino-T | 0.00 | 0.00 | 3.7 | 0.88 | 0.09 | 9.8 | 4.2 | 0.19 | 22.0 | 2.5 | 0.16 | 15.6 | 0.00 | 0.00 | 0.74 |
| OWLv2* | 5.6 | 0.14 | 40.3 | 6.7 | 0.31 | 21.7 | 11.5 | 0.32 | 35.8 | 9.9 | 0.38 | 26.2 | 0.07 | 0.01 | 9.3 |
| OWLv2 | 3.3 | 0.10 | 33.2 | 5.8 | 0.25 | 23.1 | 10.8 | 0.32 | 33.6 | 9.9 | 0.35 | 28.3 | -0.20 | -0.01 | 20.7 |
| LLMDet-L | 32.7 | 0.46 | 71.2 | 1.8 | 0.13 | 13.7 | 5.0 | 0.19 | 26.3 | 3.2 | 0.16 | 20.0 | 0.65 | 0.04 | 16.9 |
| APE-D | 1.1 | 0.10 | 11.2 | 3.1 | 0.21 | 14.7 | 7.6 | 0.26 | 28.8 | 5.8 | 0.28 | 20.8 | 0.1 | 0.01 | 7.2 |
| Gemini 2.5 | 26.6 | 0.36 | 74.0 | 5.6 | 0.20 | 27.8 | 7.4 | 0.22 | 33.8 | 6.9 | 0.23 | 29.9 | 2.1 | 0.08 | 25.6 |
| SAM 3 | 65.8 | 0.82 | 80.7 | 38.1 | 0.66 | 57.6 | 44.4 | 0.67 | 66.1 | 42.1 | 0.72 | 58.4 | 51.5 | 0.86 | 60.0 |

Table 32: Per-domain results of instance segmentation on SA-Co/Silver continued. ⋆: partially trained on LVIS.

## G.2 VISUAL EXEMPLARS AND INTERACTIVITY

In Tab. 35, visual exemplar experiments, we report performance in 3 settings: (1) text prompt only, (2) visual prompt only, and (3) both text and visual prompt. We note that (2) is quite ambiguous. For example, given a visual example of a dog, one could want to detect all dogs, or only dogs of the same color or breed. As a result, SAM 3 performs worse on SA-Co/Gold in setting (2) compared to (1). Therefore, setting (3) is better suited, as the text lifts most of the ambiguity, and the additional input box gives a hint for unfamiliar concepts.

## G.3 FEW-SHOT FINE-TUNING

We evaluate SAM 3's object detection capabilities on real-world data through comprehensive zero-shot and few-shot experiments using two established benchmarks: OdinW13 Li et al. (2022a) and Roboflow-100VL Robicheaux et al. (2025). These benchmarks encompass 13 and 100 diverse object detection datasets, respectively, capturing a wide range of real-world scenarios with standardized train and test splits that enable fair comparison with existing methods.

**Few-shot training and evaluation.** For OdinW13 few-shot experiments, we train on all three official few-shot training splits and report mean performance with standard deviation on the test split. For Roboflow-100VL, we utilize the official FSOD training splits provided by the benchmark and report numbers on the test split. We treat few-shot fine-tuning runs similarly to traditional training runs, but with some differences. We train for 40 epochs a reduced learning rate that is one-tenth of the standard value on a batch size of 2. Since these benchmarks focus exclusively on object detection without mask annotations, we disable all mask-specific components and losses during training.

**OdinW13 results.** Fig. 20a presents our few-shot performance on OdinW13, comparing SAM 3 against previous state-of-the-art methods Ren et al. (2024a); Wu et al. (2024b); Xu et al. (2023); Zhang et al. (2022b). We report mean BoxAP averaged across all 13 datasets, with SAM 3 consistently achieving superior performance and establishing new state-of-the-art results. Complete dataset-specific results for each OdinW13 dataset are provided in Fig. 20b.

**Roboflow-100VL results.** Tab. 36 summarizes our comprehensive evaluation across zero-shot, few-shot, and full fine-tuning settings on Roboflow-100VL, with results averaged across all 100 datasets. While SAM 3 underperforms the current state-of-the-art Liu et al. (2023) in zero-shot evaluation, it surpasses leading methods Liu et al. (2023); Chen et al. (2024a) in both few-shot and full fine-tuning scenarios. This demonstrates SAM 3's strong visual generalization capabilities when provided with task-specific training data. We attribute the zero-shot performance gap to the use of specialized, dataset-specific prompts that may lack broad generalizability in Roboflow-100VL. However, even minimal fine-tuning closes this gap and enables substantial performance improvements.

| Model | Box Detection | | | | | | | |
| | LVIS | | COCO | | SA-Co | | | |
| | $cgF_1$ | AP | AP | $AP_o$ | Gold $cgF_1$ | Silver $cgF_1$ | Bronze $cgF_1$ | Bio $pmF_1$ |
|---|---|---|---|---|---|---|---|---|
| Human | – | – | – | – | 74.0 | – | – | – |
| OWLv2 (Minderer et al., 2024) | 19.9 | 35.2 | 38.2 | 42.4 | 16.9 | 7.1 | 4.1 | 0.95 |
| OWLv2$\star$ (Minderer et al., 2024) | 30.2 | 45.5 | 46.1 | 23.9 | 24.5 | 11.0 | 12.0 | 0.08 |
| gDino-B (Liu et al., 2024a) | 15.8 | 25.7 | 52.5 | 45.5 | 6.0 | 4.2 | 12.2 | 0.90 |
| gDino-T (Liu et al., 2024a) | 15.1 | 20.5 | 45.7 | 35.3 | 3.4 | 2.5 | 7.6 | 0.35 |
| LLMDet-L (Fu et al., 2025) | 39.3 | 42.0 | 55.6 | 49.8 | 6.8 | 6.7 | 14.0 | 0.17 |
| LLMDet-B (Fu et al., 2025) | 36.6 | 37.8 | 54.2 | 39.4 | 5.0 | 7.3 | 14.5 | 0.27 |
| APE-D (Shen et al., 2024) | – | 59.6$^\dagger$ | 58.3$^\dagger$ | – | 17.3 | 7.7 | 14.3 | 0.00 |
| DINO-X (Ren et al., 2025) | – | 52.4$^\dagger$ | 56.0$^\dagger$ | – | 22.5$^\delta$ | – | – | – |
| Gemini 2.5 (Comanici et al., 2025) | 16.1 | – | – | – | 14.4 | 9.4 | 8.2 | 12.4 |
| **SAM 3** | **41.0** | **53.7** | **56.4** | **55.7** | **55.7** | **50.0** | **47.1** | **56.3** |

Table 33: Additional evaluation on box detection. $AP_o$ corresponds to COCO-O accuracy, $\star$: partially trained on LVIS, $\dagger$: from original papers, $\delta$: from DINO-X API. Gray numbers indicate usage of respective closed set training data (LVIS/COCO). See §F.4 for details on human performance.

| Model | Semantic Segmentation | | | | |
| | ADE-847 mIoU | ADE-150 mIoU | PC-59 mIoU | PC-459 mIoU | Cityscapes mIoU |
|---|---|---|---|---|---|
| APE-D | 9.2 | 30.0 | 58.5 | **21.8** | 44.2 |
| **SAM 3** | **13.8** | **39.0** | **60.8** | 18.8 | **65.2** |

Table 34: Additional evaluation on semantic segmentation.

Roboflow-100VL also categorizes its 100 datasets into seven dataset types; we report averages per each such dataset type in Tab. 37.

### G.4 Object Counting

We evaluate an internal SAM 3 checkpoint on object counting benchmarks CountBench (Paiss et al., 2023) and PixMo-Count (Deitke et al., 2025) to compare with MLLMs (Wang et al., 2024; Deitke et al., 2025; Comanici et al., 2025) and detection expert models (Ren et al., 2025). See Tab. 38 for results. The metrics include Accuracy(%) and Mean Absolute Error (MAE). CountBench (Paiss et al., 2023) contains 540 images and their captions, with 2-10 objects in each image. By removing images with unavailable links, we test on 487 images. PixMo-Count (Deitke et al., 2025) contains 540 images and their text descriptions in the form of simple noun phrases, with 2 to 10 objects in each image. By removing images with unavailable links, we test on 529 images.

To evaluate MLLMs on CountBench, we apply the same question set as Molmo (Deitke et al., 2025), which is inherited from PaliGemma (Beyer et al., 2024). When evaluating SAM 3 on CountBench, we modify the question sentence to the simple noun phrase. To evaluate MLLMs on PixMo-Count, we construct the question as *"How many {} are there in this image"* or *"Count the {}"*, where {} is the simple noun phrase provided by PixMo-Count annotations.

We find that presence token does not help SAM 3 on counting tasks, so we do not use it. For a group of objects, we find that SAM 3 outputs predictions for both each individual and the group as a whole, which contradicts the counting task.

| Model | SA-Co/Gold | | |
| | $pmF_1$ T | $pmF_1$ I | $pmF_1$ T+I |
|---|---|---|---|
| T-Rex2 (Jiang et al., 2024) | – | 57.6 | – |
| **SAM 3** | 66.4 | 69.5 | 74.6 |

Table 35: Visual prompting on SA-Co/Gold. We report $pmF_1$ metric in different prompt types: T (text-only), I (image-only), and T+I (combined text and image).

| Model | Zero-shot | 1-Shot | 3-Shot | 5-Shot | 10-Shot | All |
|---|---|---|---|---|---|---|
| | | | AP | | | |
| GLIPv2-H | 55.5 | $61.7 \pm 0.5$ | $64.1 \pm 0.8$ | $64.4 \pm 0.6$ | $65.9 \pm 0.3$ | 70.4 |
| GLEE-Pro | 53.4 | $59.4 \pm 1.5$ | $61.7 \pm 0.5$ | $64.3 \pm 1.3$ | $65.6 \pm 0.4$ | 69.0 |
| MQ-GLIP-L | 54.1 | 62.4 | 64.2 | 65.4 | 66.6 | 71.3 |
| Grounding DINO 1.5 Pro | 58.7 | $62.4 \pm 1.1$ | $66.3 \pm 1.0$ | $66.9 \pm 0.2$ | $67.9 \pm 0.3$ | 72.4 |
| **SAM 3** | **61.0** | $\mathbf{66.1 \pm 0.8}$ | $\mathbf{69.1 \pm 0.0}$ | $\mathbf{70.2 \pm 0.3}$ | $\mathbf{71.8 \pm 0.4}$ | **75.8** |

(a) Comparison of different models under few-shot settings on ODinW13.

| | Aerialmaritimedrone(l) | Aquarium | Rabbits | Egohands(g) | NAMushrooms | Packages | PascalVOC |
|---|---|---|---|---|---|---|---|
| | | | | AP | | | |
| 0-Shot | 20.9 | 44.5 | 80.9 | 70.0 | 89.6 | 74.6 | 65.2 |
| 1-Shot | $31.5 \pm 2.6$ | $46.5 \pm 0.4$ | $82.3 \pm 1.6$ | $73.4 \pm 0.3$ | $94.8 \pm 0.6$ | $82.9 \pm 1.6$ | $69.0 \pm 0.9$ |
| 3-Shot | $35.9 \pm 1.0$ | $49.5 \pm 0.8$ | $84.8 \pm 0.2$ | $72.3 \pm 1.0$ | $97.7 \pm 1.0$ | $88.4 \pm 1.7$ | $70.1 \pm 0.3$ |
| 5-Shot | $37.7 \pm 0.9$ | $53.4 \pm 1.4$ | $83.8 \pm 0.2$ | $73.1 \pm 1.0$ | $98.1 \pm 1.4$ | $87.9 \pm 1.3$ | $70.6 \pm 0.3$ |
| 10-Shot | $39.1 \pm 0.8$ | $54.1 \pm 0.7$ | $84.6 \pm 0.7$ | $73.5 \pm 0.6$ | $99.7 \pm 0.4$ | $92.7 \pm 1.3$ | $70.7 \pm 0.1$ |
| All | 40.1 | 61.5 | 84.7 | 79.6 | 100.0 | 98.0 | 77.4 |

| | Raccoon | Shellfish | Vehicles | Pistols | Pothole | ThermalDP | 13-Average |
|---|---|---|---|---|---|---|---|
| | | | | AP | | | |
| 0-Shot | 65.9 | 63.7 | 65.0 | 62.2 | 29.3 | 61.2 | 61.0 |
| 1-Shot | $69.8 \pm 3.1$ | $64.8 \pm 1.5$ | $67.0 \pm 1.9$ | $70.5 \pm 1.0$ | $36.5 \pm 1.0$ | $70.2 \pm 2.6$ | $66.1 \pm 0.8$ |
| 3-Shot | $77.9 \pm 2.5$ | $64.0 \pm 2.1$ | $68.0 \pm 1.0$ | $71.5 \pm 0.7$ | $40.5 \pm 0.9$ | $77.9 \pm 2.6$ | $69.1 \pm 0.0$ |
| 5-Shot | $79.9 \pm 1.6$ | $64.8 \pm 2.3$ | $67.3 \pm 0.8$ | $73.1 \pm 0.9$ | $42.7 \pm 1.1$ | $79.9 \pm 3.0$ | $70.2 \pm 0.3$ |
| 10-Shot | $84.3 \pm 0.3$ | $66.1 \pm 0.8$ | $68.0 \pm 1.5$ | $73.1 \pm 0.5$ | $45.4 \pm 0.9$ | $82.2 \pm 1.9$ | $71.8 \pm 0.4$ |
| All | 86.4 | 58.8 | 72.1 | 78.8 | 58.6 | 89.7 | 75.8 |

(b) ODinW13 per-dataset results for SAM 3.

Figure 20: Zero-shot and few-shot results on ODinW13.

As a post-processing step, we perform Non-Maximal Suppression (NMS) to remove duplicate detections. Instead of the usual Intersection-over-Union (IoU) criterion, we use Interaction over Minimum (IoM), where the area of overlap is divided by the area of the smaller mask, rather than by the area of the union. By doing this, we can detect whole *vs.* part situations: if a mask is fully covered by another, the IoM will be high, even if the covering mask is much bigger (which would lead to low IoU). We set the IoM threshold to 0.5 in our NMS process. Finally, we select the predictions with confidence higher than 0.5 as the final predictions and count the number of predictions as the counting result.

## G.5 VIDEO PCS DETAILS

In this section, we provide additional details for the video PCS evaluation (in §6 and Tab. 5).

**Benchmarks.** We evaluate the video PCS capabilities of the SAM 3 model based on an input text prompt (similar to the open-vocabulary video instance segmentation task (Wang et al., 2023)) on both our collected video benchmark SA-Co/VEval and public benchmarks. For SA-Co/VEval, we evaluate separately on each subset (SA-V, YT-Temporal-1B, and SmartGlasses) based on their data sources, and report classification-gated F1 ($cgF_1$), phrase-based HOTA (pHOTA), and Track Every Thing Accuracy (TETA). The SA-Co/VEval benchmarks contain a large number of noun phrases (5.1K in SA-V and YT-Temporal-1B subsets and 4.9K in SmartGlasses), and provides each video

| | Zero-Shot | 10-Shot | All |
|---|---|---|---|
| Model | | AP | |
| Grounding Dino | **15.7** | 33.7 | — |
| LW-DETRm | — | — | 59.8 |
| **SAM 3** | 15.2 | **36.5** | **61.6** |

Table 36: Comparison on Roboflow100-VL.

| | Aerial | Document | Flora-Fauna | Industrial | Medical | Other | Sports | Average |
|---|---|---|---|---|---|---|---|---|
| | | | | AP | | | | |
| 0-Shot | 24.0 | 12.9 | 23.9 | 9.0 | 2.6 | 17.0 | 15.6 | 15.2 |
| 10-Shot | 35.4 | 35.8 | 39.6 | 40.4 | 25.7 | 35.0 | 40.3 | 36.5 |
| All | 56.9 | 64.2 | 60.1 | 68.4 | 52.5 | 63.8 | 61.5 | 61.6 |

Table 37: SAM 3 Roboflow100-VL results by dataset type.

| | | | CountBench | | PixMo-Count | |
|---|---|---|---|---|---|---|
| Model | Presence | IoM | MAE ↓ | Acc (%) ↑ | MAE ↓ | Acc (%) ↑ |
| SAM 3 | × | × | 0.34 | 84.8 | 0.34 | 76.9 |
| SAM 3 | ✓ | × | 0.50 | 83.7 | 0.41 | 75.8 |
| SAM 3 | × | ✓ | **0.11** | **94.0** | **0.21** | **86.3** |
| SAM 3 | ✓ | ✓ | 0.38 | 89.3 | 0.29 | 85.2 |

Table 38: Ablation on counting results on an internal SAM 3 checkpoint.

with a list of noun phrases as text prompts. During evaluation, for each evaluation video, we prompt SAM 3 with the list of noun phrases provided for that video, as shown in Tab. 39 (a, b, c).

For public benchmarks, we evaluate on LVVIS (Wang et al., 2023), BURST (Athar et al., 2023), YTVIS (Ke et al., 2022), OVIS (Qi et al., 2022), BDD100K (Yu et al., 2020), GMOT40 (Bai et al., 2021), and DeepSeaMOT (Barnard et al., 2025), and report the official metrics on each dataset (for DeepSeaMOT, we report the average performance over its 4 subsets). These public benchmarks are often based on a set of categories, with a relatively large vocabulary size in LVVIS and BURST (1196 categories in LVVIS and 482 categories in BURST) and much smaller numbers of categories in other datasets. We use the category name as the text prompt, and prompt SAM 3 with all category names in the dataset on every evaluation video, as shown in Tab. 39 (d).

**Video PCS Metrics.** Similar to its definition in the image domain in §F.3, we define the classification-gated F1 (**cgF$_1$**) metric on videos as the multiplication between the video-level Matthews correlation coefficient (**VL_MCC**) on whether the noun phrase exists in the video and the localization positive macro F1 (**pmF$_1$**) on positive noun phrases. To decide whether a predicted masklet matches a ground-truth masklet, we measure their volume intersection-over-union (IoU), defined by their total intersection volume divided by their total union volume over the video. When computing pmF$_1$, we averaged the results over multiple volume IoU thresholds from 0.5 to 0.95 with increments of 0.05, similar to how it is computed on images.

We also evaluate the phrase-based HOTA (**pHOTA**) metric, where we compute the Higher Order Tracking Accuracy (HOTA) metric (Luiten et al., 2021) over all video-NP pairs along with their breakdown into phrase-based detection accuracy (**pDetA**) and phrase-based association accuracy (**pAssA**). As the HOTA metric was originally designed for category-based evaluation, to get its phrase-based variant pHOTA for open-vocabulary prompts, we remap each video-NP pair in the evaluation benchmark as a new unique video ID and then set all ground-truth annotations and predictions to have the same category ID (i.e., the total number of video IDs after remapping equals the total number of video-NP pairs in the evaluation benchmark). That is, each video-NP pair in the benchmark is treated as an isolated sample for prediction and evaluation, and the results are aggregated over all video-NP pairs in a class-agnostic manner. More specifically, we save the remapped ground-truth annotations and the predictions as JSON files under the YTVIS format, and use the TrackEval package (Jonathon Luiten, 2020) to obtain the mask HOTA statistics on this remapped dataset (using the YTVIS dataset wrapper in TrackEval along with its default parameters), and report their results as pHOTA, pDetA, and pAssA. Similarly, we also evaluate the Track Every Thing Accuracy (**TETA**) metric (Li et al., 2022c) over the masklet predictions on these datasets.

**Baselines.** We compare the SAM 3 model with several baselines, including GLEE (Wu et al., 2024a) (a previous work on open-vocabulary image/video segmentation), "LLMDet as detector + SAM 3 Tracker", by replacing the Detector component in SAM 3 with a recent open-vocabulary detector LLMDet (Fu et al., 2025), and "SAM 3 Detector + T-by-D as tracker", by replacing the Tracker in SAM 3 with an association module similar as in tracking-by-detection approaches (Wojke et al., 2017; Zhang et al., 2022c).

For GLEE (Wu et al., 2024a), we follow its official implementation. Since GLEE supports taking as inputs multiple text prompt simultaneously, we evaluate it in two ways: a) prompting it with all the noun phrases from an evaluation video at once, denoted as "GLEE (prompted w/ all NPs at once)" in Tab. 39, and b) looping over each noun phrase in the evaluation video and prompting GLEE with one noun phrase at a time, denoted as "GLEE (prompted w/ one NP at a time)". We find that for open-vocabulary segmentation on videos, it is usually better to prompt GLEE with one noun phrase at a time instead of prompting it with all noun phrases at once.

For "LLMDet as Detector + SAM 3 Tracker", we replace the detection outputs from the SAM 3 detector with LLMDet (Fu et al., 2025) bounding box outputs, and obtain the mask output by prompting it with the SAM 3 component. Then we apply the SAM 3 Tracker similar to how it is applied over the SAM 3 Detector output. We also note that GLEE and LLMDet have not been trained on the noun phrases in the SA-Co dataset, so their results should be seen as zero-shot on the SA-Co/VEval benchmark.

For "SAM 3 Detector + T-by-D as tracker", we replace the SAM 3 Tracker with a detection-to-masklet association module as commonly used in the tracking-by-detection paradigm, e.g. Wojke et al. (2017); Zhang et al. (2022c). The detection-to-masklet association module tries to match the masklets already tracked in previous frames with detected objects in the current frame, based on a dot product between the visual features of each detected object and the visual features of the past 16 frames of a masklet. If a high-confidence detection isn't matched to any existing masklet, we add it as a new object and start a new masklet for it. The association module is trained on the SA-Co dataset.

**Results.** As shown in Tab. 39, SAM 3 largely outperforms these baselines across the benchmarks. On SA-Co/VEval with a very large number of noun phrases, SAM 3 excels in both frame-level detection (pDetA) and cross-frame association (pAssA). Comparisons with "LLMDet as Detector + SAM 3 Tracker" and "SAM 3 Detector + T-by-D as tracker" demonstrate that both the Detector module and the Tracker module in SAM 3 play a critical role in the final video performance. In public benchmarks, SAM 3 also achieves a strong performance, including new state-of-the-art results on LVVIS and OVIS. We also note that GLEE and LLMDet have not been trained on the SA-Co dataset, so their results should be seen as zero-shot on SA-Co/VEval. In addition, the SmartGlasses subset in SA-Co/VEval contains many egocentric videos, which might be out of the training distribution for GLEE and LLMDet.

**Strategies on Temporal Disambiguation.** As described in §D.3, SAM 3 adopts several strategies to address ambiguities in videos. In Tab. 39, we also report two other settings where we turn off all these temporal disambiguation strategies ("**SAM 3** w/o any temporal disambiguation"). The results show that the disambiguation strategies boost the video PCS performance (especially under the pHOTA metric). We also find that the disambiguation strategies notably improve the qualitative outputs on videos.

### G.6 PVS DETAILS

We evaluate SAM 3 on a range of Promptable Video Segmentation (PVS) tasks as in Ravi et al. (2024).

**Video Object Segmentation (VOS).** The VOS task requires tracking an object throughout a video given an input segmentation mask. As shown in Tab. 6, we compare SAM 3 with recent state-of-the-art models on the VOS task, including SAMURAI (Yang et al., 2024), SAM2Long (Ding et al., 2024), and SeC (Zhang et al., 2025). SAM 3 brings gains on all datasets, including the challenging MOSEv2 benchmark (Ding et al., 2025) and datasets with long videos such as LVOSv2 (Hong et al., 2024).

**Interactive Image Segmentation.** We evaluate SAM 3 on the 37-dataset benchmark introduced in Ravi et al. (2024) for the interactive image segmentation task. As shown in Tab. 7, SAM 3 outperforms SAM 1 and SAM 2 on average $\mathrm{mIoU}$, producing more accurate segmentation masks when prompted with 1 or 5 clicks.

**Interactive Video Segmentation.** We follow interactive offline and online evaluation protocol Ravi et al. (2024) and compare SAM 3 with baseline methods, including SAM 2, SAM + XMem++, and SAM + Cutie. The interactive offline evaluation involves multiple passes over the entire video, while the interactive online evaluation involves only one pass over the entire video. We use the same 9

| | **SA-Co/VEval SA-V val** (1.5K NPs) | | | | | | | **SA-Co/VEval SA-V test** (1.5K NPs) | | | | | | |
|---|---|---|---|---|---|---|---|---|---|---|---|---|---|---|
| Model | cgF$_1$ | VL_MCC | pmF$_1$ | pHOTA | pDetA | pAssA | TETA | cgF$_1$ | VL_MCC | pmF$_1$ | pHOTA | pDetA | pAssA | TETA |
| Human | 53.9 | 0.86 | 62.4 | 71.7 | 56.9 | 90.8 | 68.9 | 53.1 | 0.87 | 60.8 | 70.5 | 55.5 | 90.2 | 69.1 |
| GLEE[†] (prompted w/ all NPs at once) | 0.1 | 0.17 | 0.4 | 7.7 | 2.6 | 28.5 | 6.5 | 0.1 | 0.22 | 0.6 | 8.7 | 2.8 | 30.9 | 7.1 |
| GLEE[†] (prompted w/ one NP at a time) | 0.2 | 0.06 | 2.5 | 12.4 | 3.9 | 41.7 | 15.7 | 0.1 | 0.05 | 2.2 | 11.8 | 3.4 | 43.1 | 14.9 |
| LLMDet as detector + **SAM 3** Tracker | 1.6 | 0.11 | 14.1 | 30.5 | 13.0 | 72.1 | 29.3 | 2.3 | 0.15 | 14.7 | 30.1 | 12.0 | 76.0 | 28.5 |
| **SAM 3** Detector + T-by-D as tracker | 23.7 | **0.68** | 34.7 | 57.6 | 44.5 | 75.3 | 53.4 | 25.7 | **0.72** | 35.8 | 55.7 | 40.2 | 78.3 | 53.2 |
| **SAM 3** w/o any temporal disambiguation | 24.4 | 0.56 | 43.4 | 57.4 | 40.1 | 83.1 | **57.4** | 27.1 | 0.63 | 42.9 | 55.9 | 37.9 | 83.4 | **57.1** |
| **SAM 3** | 29.3 | 0.66 | **44.5** | **60.7** | **44.7** | 83.2 | 57.3 | 30.3 | 0.69 | **43.7** | **58.0** | **40.9** | 83.4 | 56.8 |

(a) Results on SA-Co/VEval SA-V val and test

| | **SA-Co/VEval YT-Temporal-1B val** (1.4K NPs) | | | | | | | **SA-Co/VEval YT-Temporal-1B test** (1.5K NPs) | | | | | | |
|---|---|---|---|---|---|---|---|---|---|---|---|---|---|---|
| Model | cgF$_1$ | VL_MCC | pmF$_1$ | pHOTA | pDetA | pAssA | TETA | cgF$_1$ | VL_MCC | pmF$_1$ | pHOTA | pDetA | pAssA | TETA |
| Human | 71.3 | 0.98 | 73.1 | 78.3 | 68.6 | 89.6 | 83.4 | 71.2 | 0.97 | 73.2 | 78.4 | 68.8 | 89.7 | 84.3 |
| GLEE[†] (prompted w/ all NPs at once) | 1.8 | 0.26 | 7.0 | 17.1 | 7.3 | 42.0 | 16.9 | 1.6 | 0.24 | 6.7 | 16.7 | 6.8 | 42.6 | 16.5 |
| GLEE[†] (prompted w/ one NP at a time) | 2.3 | 0.23 | 10.3 | 19.5 | 9.2 | 42.0 | 23.2 | 2.2 | 0.22 | 9.9 | 18.9 | 8.5 | 42.9 | 22.5 |
| LLMDet as detector + **SAM 3** Tracker | 10.5 | 0.39 | 26.8 | 39.6 | 20.1 | 78.3 | 37.0 | 8.0 | 0.33 | 24.1 | 37.9 | 18.7 | 77.0 | 33.4 |
| **SAM 3** Detector + T-by-D as tracker | 46.0 | **0.90** | 51.3 | 68.5 | **61.0** | 77.6 | 70.4 | 47.6 | **0.93** | 51.3 | 68.2 | **60.8** | 77.0 | 70.4 |
| **SAM 3** w/o any temporal disambiguation | 47.0 | 0.84 | 55.8 | 68.6 | 57.8 | 82.0 | 70.1 | 47.3 | 0.84 | 56.3 | 67.8 | 57.1 | 81.2 | 69.8 |
| **SAM 3** | 50.2 | 0.88 | **57.2** | **70.5** | 60.5 | **82.7** | **70.8** | 50.8 | 0.89 | **57.2** | **69.9** | 60.2 | **81.7** | **70.5** |

(b) Results on SA-Co/VEval YT-Temporal-1B val and test

| | **SA-Co/VEval SmartGlasses val** (2.2K NPs) | | | | | | | **SA-Co/VEval SmartGlasses test** (2.2K NPs) | | | | | | |
|---|---|---|---|---|---|---|---|---|---|---|---|---|---|---|
| Model | cgF$_1$ | VL_MCC | pmF$_1$ | pHOTA | pDetA | pAssA | TETA | cgF$_1$ | VL_MCC | pmF$_1$ | pHOTA | pDetA | pAssA | TETA |
| Human | 53.9 | 0.91 | 59.6 | 68.1 | 53.0 | 87.9 | 71.0 | 58.5 | 0.92 | 63.5 | 72.3 | 58.8 | 89.4 | 77.1 |
| GLEE[†] (prompted w/ all NPs at once) | 0.0 | 0.15 | 0.2 | 4.1 | 1.4 | 13.1 | 8.4 | 0.0 | 0.16 | 0.3 | 4.7 | 1.6 | 15.3 | 9.3 |
| GLEE[†] (prompted w/ one NP at a time) | 0.1 | 0.14 | 0.5 | 4.7 | 1.4 | 16.0 | 13.3 | 0.1 | 0.14 | 0.4 | 5.6 | 1.8 | 18.2 | 14.7 |
| LLMDet as detector + **SAM 3** Tracker | 0.3 | 0.02 | 14.1 | 16.4 | 3.8 | 71.4 | 36.9 | 0.3 | 0.02 | 16.8 | 18.6 | 4.7 | 74.1 | 39.0 |
| **SAM 3** Detector + T-by-D as tracker | 27.2 | **0.84** | 32.4 | 56.2 | **46.6** | 68.4 | 60.6 | 29.7 | **0.85** | 35.1 | 60.0 | **50.6** | 71.7 | 61.2 |
| **SAM 3** w/o any temporal disambiguation | 30.9 | 0.71 | 43.5 | 58.1 | 43.2 | 78.7 | **65.1** | 34.4 | 0.75 | 45.8 | 61.6 | 47.0 | 81.2 | **66.0** |
| **SAM 3** | 33.5 | 0.76 | **44.4** | **60.2** | 46.2 | **79.3** | 65.0 | 36.4 | 0.78 | **46.6** | **63.6** | 50.0 | **81.5** | 65.9 |

(c) Results on SA-Co/VEval SmartGlasses val and test

| | **LVVIS test** (1.2K NPs) | | | **BURST test** (482 NPs) | | | **YTVIS21 val** (40 NPs) | **OVIS val** (25 NPs) | **BDD100K val** (8 NPs) | **GMOT40** (10 NPs) | **DeepSeaMOT** (29 NPs) |
|---|---|---|---|---|---|---|---|---|---|---|---|
| Model | mAP | mAP$_{base}$ | mAP$_{novel}$ | HOTA | HOTA$_{common}$ | HOTA$_{uncommon}$ | mAP | mAP | TETA | HOTA | HOTA |
| GLEE (prompted w/ all NPs at once) | 20.8 | 24.0 | 18.4 | 28.4 | 46.4 | 24.8 | **62.2** | 38.7 | 18.0 | 36.8 | 6.8 |
| GLEE (prompted w/ one NP at a time) | 9.3 | 13.9 | 5.9 | 20.2 | 42.9 | 15.7 | 56.5 | 32.4 | 14.9 | 29.9 | 22.9 |
| LLMDet as detector + **SAM 3** Tracker | 15.2 | 15.1 | 15.3 | 33.3 | 45.8 | 30.8 | 31.3 | 20.4 | 28.9 | 24.9 | 36.4 |
| **SAM 3** Detector + T-by-D as tracker | 35.9 | 32.9 | 38.2 | 39.7 | 59.7 | 35.8 | 56.5 | 55.1 | **51.0** | 60.5 | 35.5 |
| **SAM 3** | **36.3** | **33.3** | **38.5** | **44.5** | **63.5** | **40.7** | 57.4 | **60.5** | 47.2 | 60.3 | **39.9** |

(d) Results on public benchmarks

Table 39: More details on video PCS from a text prompt (open-vocabulary video instance segmentation) on SA-Co/VEval and public benchmarks (# NPs of each benchmark in parentheses). SAM 3 excels in both frame-level detection (pDetA) and cross-frame association (pAssA) and largely outperforms the baselines, especially on benchmarks with a very large number of noun phrases. †: GLEE and LLMDet have not been trained on the SA-Co dataset, so their results should be seen as zero-shot on SA-Co/VEval.

zero-shot datasets and 3 clicks per interacted frame as in Ravi et al. (2024) (see Sec. F.1.2 of Ravi et al. (2024) for details). The results are in Fig. 21, where SAM 3 achieves better overall performance in both interactive offline and online evaluation.

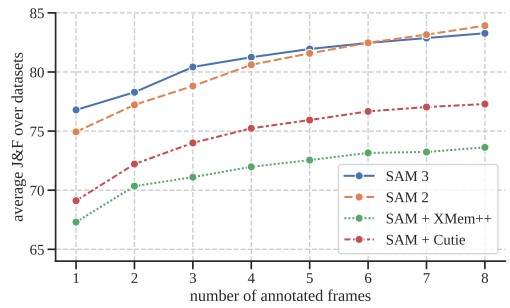

(a) *offline* $\mathcal{J}\&\mathcal{F}$ averaged across 9 datasets

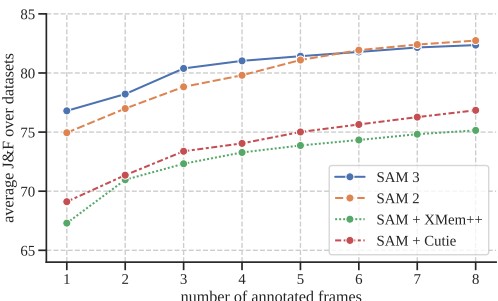

(b) *online* $\mathcal{J}\&\mathcal{F}$ averaged across 9 datasets

| Method | EndoVis 2018 | ESD | LVOSv2 | LV-VIS | PUMaVOS | UVO | VIPSeg | Virtual KITTI 2 | VOST | (average) |
|---|---|---|---|---|---|---|---|---|---|---|
| SAM + XMem++ | 68.9 | 88.2 | 72.1 | 86.4 | 60.2 | 74.5 | 84.2 | 63.8 | 46.6 | 71.7 |
| SAM + Cutie | 71.8 | 87.6 | 82.1 | 87.1 | 59.4 | 75.2 | 84.4 | 70.3 | 54.3 | 74.7 |
| SAM 2 | 77.0 | 90.2 | 87.9 | **90.3** | 68.5 | **79.2** | **88.3** | 74.1 | 67.5 | 80.3 |
| **SAM 3** | **79.1** | **91.0** | **89.7** | 88.8 | **68.9** | 77.6 | 85.9 | **75.6** | **71.6** | **80.9** |

(c) $\mathcal{J}\&\mathcal{F}$ on each dataset averaged over 1~8 interacted frames under interactive *offline* evaluation

| Method | EndoVis 2018 | ESD | LVOSv2 | LV-VIS | PUMaVOS | UVO | VIPSeg | Virtual KITTI 2 | VOST | (average) |
|---|---|---|---|---|---|---|---|---|---|---|
| SAM + XMem++ | 71.4 | 87.8 | 72.9 | 85.2 | 63.7 | 74.7 | 82.5 | 63.9 | 52.7 | 72.8 |
| SAM + Cutie | 70.5 | 87.3 | 80.6 | 86.0 | 58.9 | 75.2 | 82.1 | 70.4 | 54.6 | 74.0 |
| SAM 2 | 77.5 | 88.9 | 87.8 | **88.7** | 72.7 | **78.6** | **85.5** | 74.0 | 65.0 | 79.8 |
| **SAM 3** | **79.2** | **89.6** | **89.7** | 87.9 | **73.1** | 77.5 | 84.2 | **75.6** | **67.9** | **80.5** |

(d) $\mathcal{J}\&\mathcal{F}$ on each dataset averaged over 1~8 interacted frames under interactive *online* evaluation

Figure 21: Interactive video segmentation of SAM 3 vs baselines under offline and online evaluation, following the setup in Ravi et al. (2024) with the same 9 zero-shot datasets and 3 clicks per interacted frame. The $\mathcal{J}\&\mathcal{F}$ under different numbers of interactive frames are shown in (a) and (b), while the $\mathcal{J}\&\mathcal{F}$ on each dataset is shown in (c) and (d).

### G.6.1 Additional Model Outputs for Different Tasks

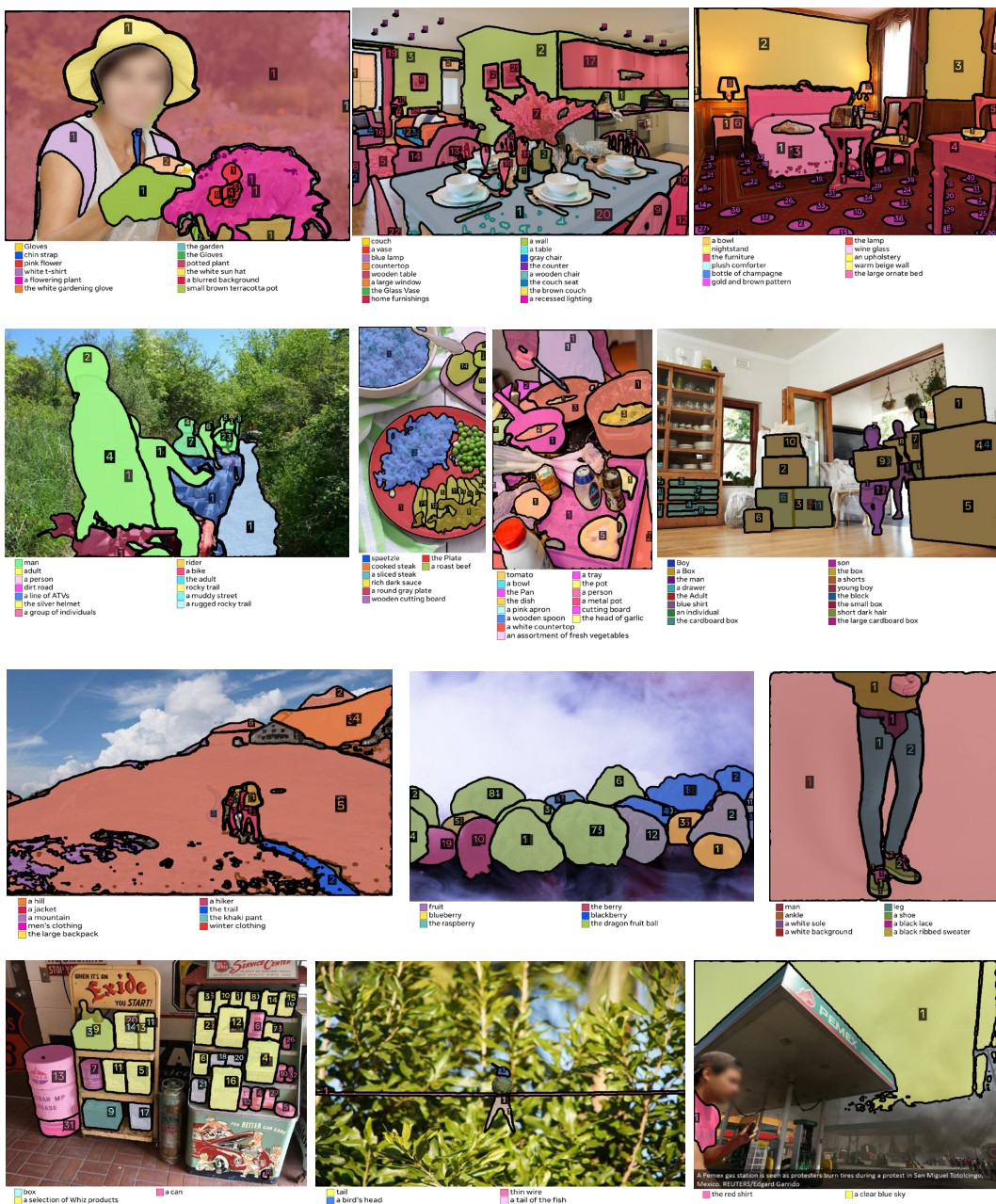

Figure 22: Example predictions on the SA-Co/Gold dataset. As these are model outputs, occasional errors may be present.

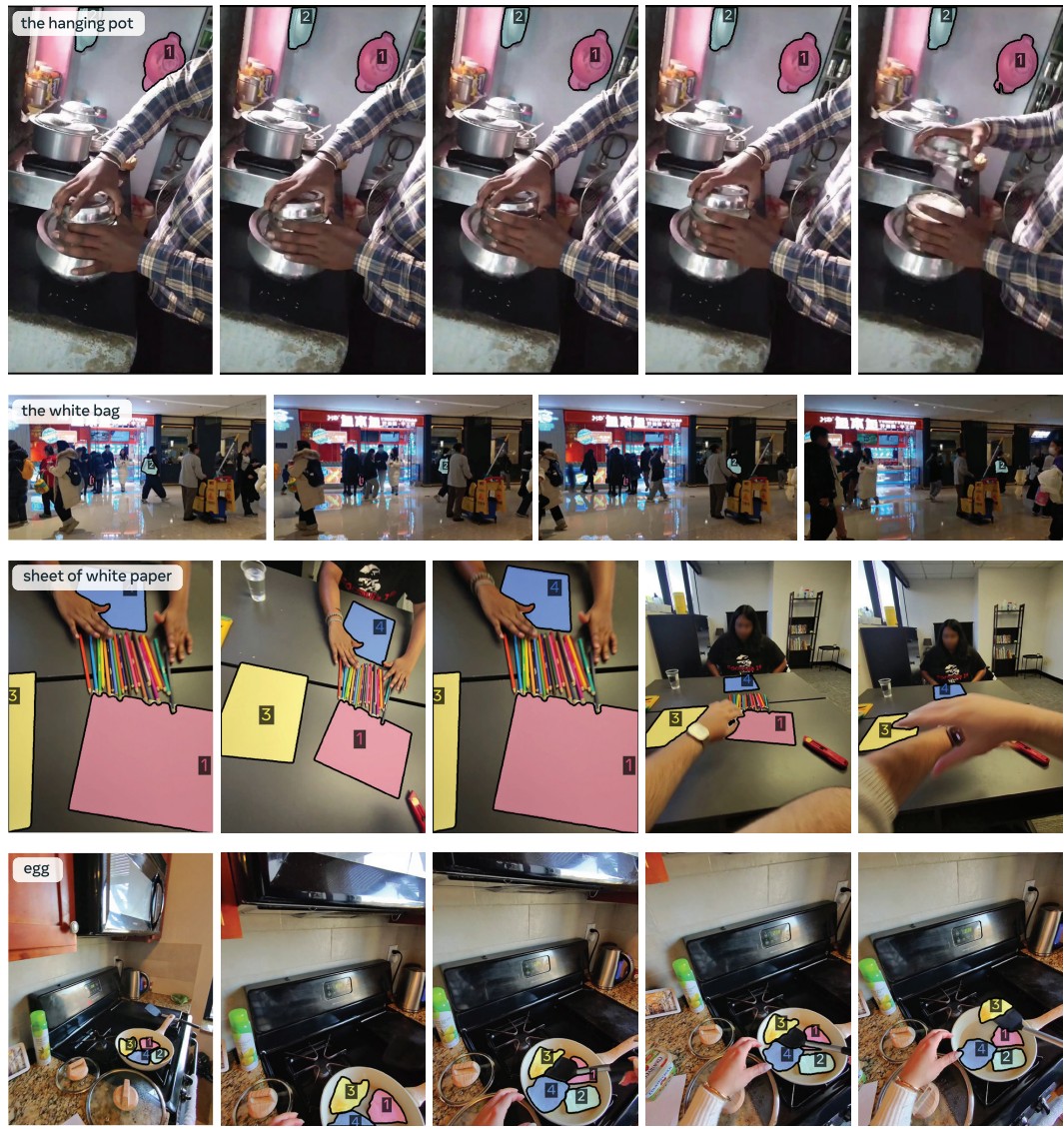

Figure 23: Example video concept segmentation predictions on the SA-Co/VEval SA-V test set (top two rows) and the SA-Co/VEval SmartGlasses test set (bottom two rows).

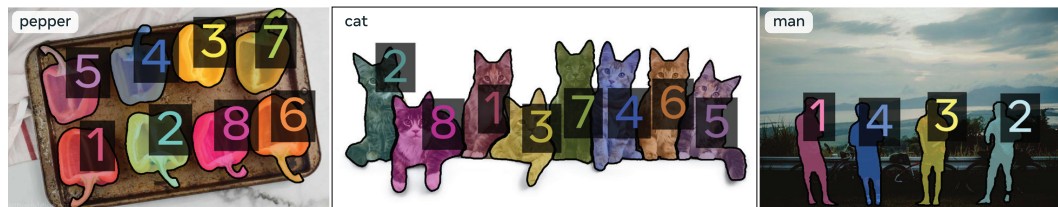

Figure 24: Example predictions on the countbench dataset.

# H  SAM 3 AGENT

## H.1  AGENT DESIGN

In this section, we introduce SAM 3 Agent, a visual agentic system that turns natural-language segmentation requests into precise masks through dynamically querying a multimodal LLM (MLLM) and SAM 3. Given an input image and a user request, an MLLM acts as a planner/controller: it analyzes the image, devises a step-by-step plan, invokes SAM 3 to generate masks, inspects the results, and finalizes candidate masks. After each action, the agent receives visual and textual feedback describing the updated environment state, enabling it to revise the plan and decide the next action. This perception-action loop continues until the agent is confident it has satisfied the goal (or determines that no valid mask exists), at which point it returns a final set of masks. The resulting pipeline handles queries far more complex than simple noun phrases which require understanding relationships between scene elements and visual common sense.

Each action consists of calling one of several "tools". We define the following four basic tools for the MLLM to call: *segment_phrase*, *examine_each_mask*, *select_masks_and_return*, and *report_no_mask*. Among these four tools, select_masks_and_return and report_no_mask are return tools, which will trigger a return function and end the current task. The other two functions: segment_phrase and examine_each_mask, are intermediate tools, which will either call the SAM 3 model on a noun phrase or trigger an iterative process for the MLLM to examine each generated mask.

---

**Tool #1: Segment Phrase**

(1) **Definition**: Use the Segment Anything 3 model to ground all instances of a simple noun phrase by generating segmentation mask(s) that cover those instances on the raw input image. At the same time, all previously generated mask(s) will be deleted and cannot be referred to in future messages.

(2) **Use cases**: Given a simple, direct, and singular noun phrase (not a referring expression that requires additional understanding/reasoning), segment_phrase will try to locate all object instance(s) on the raw input image that match the simple noun phrase you provided. The tool will also render all of the generated segmentation mask(s) onto the image for you to examine and decide the next step.

(3) **Parameters**: {"type": "object", "properties": {"text_prompt": {"type": "string", "description": "A short and simple noun phrase, e.g., rope, bird beak, speed monitor, brown handbag, person torso"}}, "required": ["text_prompt"]}

(4) **Return type**: A new image with differently colored segmentation mask(s) rendered on it, and a text message indicating the number of mask(s) generated by the Segment Anything 3 model for this "text_prompt" only.

---

**Tool #2: Examine Each Mask**

(1) **Definition**: Use this tool when the segment_phrase tool generates multiple small or overlapping mask(s), making it difficult to distinguish the correct mask(s). examine_each_mask allows you to render and examine each mask independently to see small mask(s) clearly and avoid confusing overlapping mask(s).

(2) **Use cases**: Sometimes there are multiple small mask(s) or overlapping mask(s) rendered on an image, making it difficult to distinguish each mask from others. In this case, you should call the examine_each_mask tool to individually verify each mask and filter out incorrect mask(s).

(3) **Parameters**: None

(4) **Return type**: A new image with colored segmentation mask(s) accepted by the examine_each_mask tool, and a text message indicating how many masks were accepted.

---

---

**Tool #3: Select Masks And Return**

(1) **Definition**: Call this tool to select a subset of or all of the mask(s) rendered on the most recent image as your final output. When calling select_masks_and_return, you cannot select any mask(s) generated by previous rounds other than the most recent round in your "final_answer_masks". You can only use mask(s) from the most recent image in your message history.

(2) **Use cases**: Given an image with one or more segmentation mask(s) already rendered on it, select_masks_and_return returns the set of mask(s) you select as the final output.

(3) **Parameters**: {"type": "object", "properties": {"final_answer_masks": {"type": "array", "description": "An array of integers representing the selected mask(s) you want to choose as your final output, e.g., [1, 4, 5]"}}, "required": ["final_answer_masks"]}

(4) **Return type**: None (End of Conversation)

---

**Tool #4: Report No Mask**

(1) **Definition**: Call this tool when you are absolutely sure that there are no object(s) in the image that match or answer the initial user input query.

(2) **Use cases**: Reporting that the given image does not contain any target object(s) that match or answer the initial user input query.

(3) **Parameters**: None

(4) **Return type**: None (End of Conversation)

---

After each intermediate tool call has been executed, the system will provide the MLLM with the following two pieces of information:

- The user input image with all generated and currently available segmentation masks rendered on it in a Set-of-Marks (Yang et al., 2023) manner. The masks are randomly colored and numbered from 1 to N in decreasing order of SAM 3 confidence scores received at the time of mask generation. The set of currently available masks, combined with the original user input image, defines the environment state of the SAM 3 Agent at the current time step.

- An automatically generated text message stating all changes from the previous environment state (e.g. how many masks have been generated by the segment_phrase tool, or how many masks were removed by the examine_each_mask tool).

After analyzing the updated image with currently available masks rendered on it (current environment state) in the context of the initial user query (task goal), the MLLM must update its tool-calling plan and generate the next tool call (current action). We allow the MLLM to call each intermediate tool as many times as it needs, before arriving at a final set of segmentation masks on the input image (terminal state) that it is satisfied with.

Empirically, we observe that for especially challenging queries, SAM 3 Agent may produce as many as 60 steps of trial and error before being satisfied with its grounding outcome and calling a return tool. This results in an extremely long environment-state context history with each step containing a new image, pushing both the context limit and multi-image reasoning capability of even current state-of-the-art MLLMs.

To resolve this issue, we propose an aggressive context engineering mechanism that prunes all intermediate trial-and-error states between the initial user text query and the most recent agent call to the segment_phrase tool. We also discard all previously generated masks after each tool call to the segment_phrase tool, which avoids cluttering the rendered Set-of-Marks image with redundant masks. To avoid losing important failure experience from pruned steps, we provide a continuously updated list of all previously used (and discarded) SAM 3 noun phrase prompts for the model to note.

## H.2 QUALITATIVE ANALYSIS

In this section, we provide success 25 and failure 26 examples of SAM 3 Agent on the ReasonSeg (Lai et al., 2024) and RefCOCOg (Kazemzadeh et al., 2014) datasets, as they are currently the most

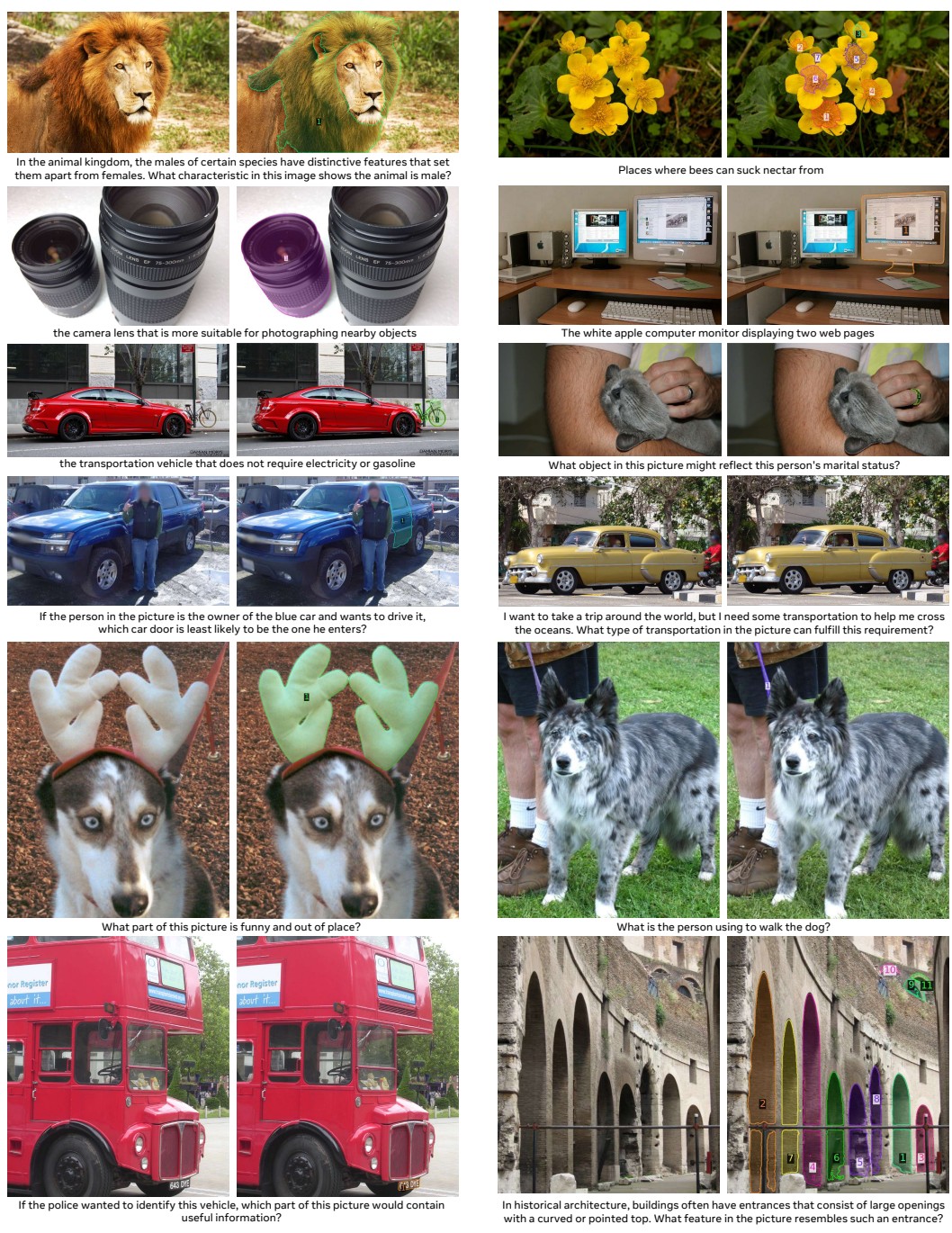

Figure 25: Successful examples of SAM 3 Agent (Qwen2.5-VL 72B) on the ReasonSeg (Lai et al., 2024) dataset for Reasoning Segmentation and the RefCOCOg (Kazemzadeh et al., 2014) dataset for Referring Expression Segmentation. For each example, see the original input image (left), textual user query (bottom), and final segmentation output (if applicable) from SAM 3 Agent (right).

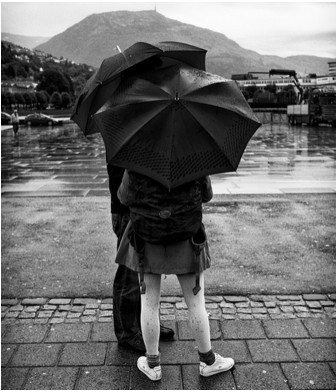 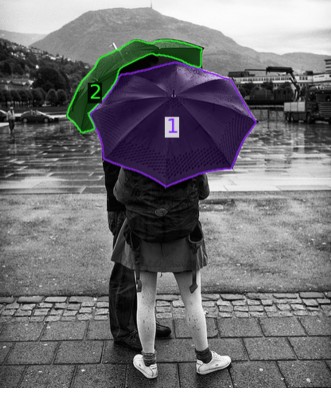 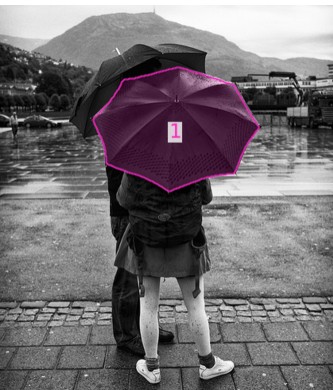

(a) Original Input Image            (b) SAM3 Intermediate Masks            (c) Final Selected Mask

Figure 26: Error example of SAM 3 Agent (Qwen2.5-VL 72B) including the original input image and SAM 3 intermediate output masks. The textual user query is "A black object that protects you from the rain, being held by a person in jeans". The agent was able to reason and find a suitable grounding target for calling SAM 3 ("black umbrella"). However, it failed to select the correct mask during final mask selection due to a visual reasoning error by the MLLM.

challenging and widely used reasoning segmentation and referring expression segmentation datasets. We also provide a complete reasoning trace example of SAM 3 Agent, demonstrating how SAM 3 Agent solves complex reasoning segmentation queries by leveraging SAM 3 for precise grounding and MLLM for visual reasoning.

Empirically, we observe that SAM 3 Agent is able to handle free-form textual queries of varying spatial and logical reasoning complexity. It is able to reject queries that do not point to any object in the given image and to generate multiple output masks for queries that match multiple objects in the given image. The MLLM backbone allows SAM 3 Agent to robustly handle queries of varying length and format. It also enables SAM 3 Agent to accurately segment fine-grained concepts that SAM 3 alone struggle with, such as specific aircraft types and fine-grained food categories.

---

**SAM3 Agent Complete Reasoning Trace Example**

```
[
    {
        "role": "system",
        "content": {System prompt including tool definitions, agent instructions, and
    response format.}
    },
    {
        "role": "user",
        "content": [
            {
                "type": "image",
                "image": Figure {\ref{fig:agent_mario_1}}
            },
            {
                "type": "text",
                "text": "the stronger Mario"
            }
        ]
    },
    {
        "role": "assistant",
        "content": [
            {
                "type": "text",
                "text": "<think> There is only one image in the context. The image shows a
    banner with pixel art characters resembling Mario. The user wants to ground \"the
    stronger Mario,\" which appears to be the character on the right side of the equation
    depicted on the banner. I will use the `segment_phrase` function with the text
    prompt \"stronger Mario\" to attempt to ground the target object.\n\nBefore
    proceeding, I remind myself that each call to `segment_phrase` will delete all
```

```
                        previously generated masks, so I need to ensure my plan is effective in one go. I
                        will start by attempting to ground \"stronger Mario.\"\n\n</think>\n<tool> {\"name\":
                         \"segment_phrase\", \"parameters\": {\"text_prompt\": \"stronger Mario\"}} </tool>\n
                        \n"
                    }
                ]
            },
            {
                "role": "user",
                "content": [
                    {
                        "type": "text",
                        "text": "SAM3 did not generate any masks for the prompt 'stronger Mario',
                        please call the segment_phrase tool again with a different, perhaps more general and
                        simple noun phrase prompt."
                    }
                ]
            },
            {
                "role": "assistant",
                "content": [
                    {
                        "type": "text",
                        "text": "<think> There is still only one image in the context. The
                        previous attempt to ground \"stronger Mario\" did not yield any masks. I need to try
                        a simpler and more general noun phrase that still captures the essence of the target.
                         Given the pixel art style, I'll try grounding \"Mario\" to see if it captures the
                        character on the right side of the equation.\n\nI remind myself that each call to `
                        segment_phrase` will delete all previously generated masks, so I need to ensure my
                        plan is effective in one go. I will now attempt to ground \"Mario.\"\n\n</think>\n<
                        tool> {\"name\": \"segment_phrase\", \"parameters\": {\"text_prompt\": \"Mario\"}} </
                        tool>\n\n"
                    }
                ]
            },
            {
                "role": "user",
                "content": [
                    {
                        "type": "text",
                        "text": "SAM3 did not generate any masks for the prompt 'Mario', please
                        call the segment_phrase tool again with a different, perhaps more general and simple
                        noun phrase prompt."
                    }
                ]
            },
            {
                "role": "assistant",
                "content": [
                    {
                        "type": "text",
                        "text": "<think> There is still only one image in the context. The
                        previous attempts to ground \"stronger Mario\" and \"Mario\" did not yield any masks.
                         Given the pixel art style and the context of the image, I will try a more general
                        and simple noun phrase related to the character's appearance. I will attempt to
                        ground \"pixel character\" to see if it captures the target.\n\nI remind myself that
                        each call to `segment_phrase` will delete all previously generated masks, so I need
                        to ensure my plan is effective in one go. I will now attempt to ground \"pixel
                        character.\"\n\n</think>\n<tool> {\"name\": \"segment_phrase\", \"parameters\": {\"
                        text_prompt\": \"pixel character\"}} </tool>\n\n"
                    }
                ]
            },
            {
                "role": "user",
                "content": [
                    {
                        "type": "text",
                        "text": "The original user query was: 'the stronger Mario', and the SAM3
                        prompt you used was: 'pixel character'. SAM3 generated 4 masks for the prompt 'pixel
                        character', which are rendered in the latest image, now please analyze this new image
                         and decide your next action."
                    },
                    {
                        "type": "image",
                        "image": Figure {\ref{fig:agent_mario_2}}
                    }
                ]
            },
```

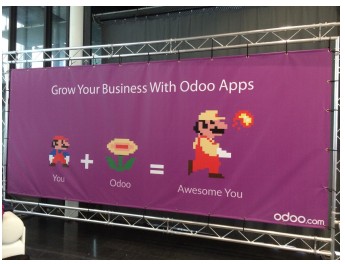 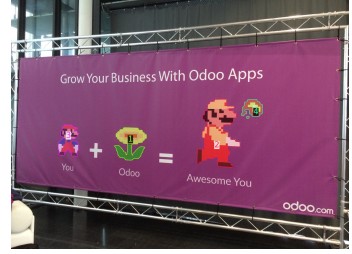 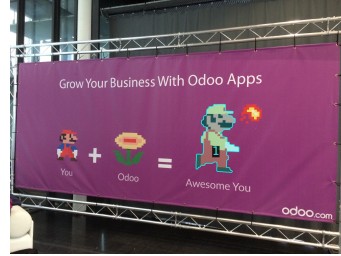

(a) Original Input Image      (b) SAM3 Intermediate Masks      (c) Final Output Mask

Figure 27: Images from the Complete Reasoning Trace Example of SAM 3 Agent (Qwen2.5-VL 72B), including images (a) and (b) from the agent context and image (c) containing the final agent output mask. The textual user query is "the stronger Mario". The SAM 3 Intermediate Masks are generated by SAM 3 using the text prompt "pixel character".

```
{
    "role": "assistant",
    "content": [
        {
            "type": "text",
            "text": "<think> There are two images in the context now: the raw input
image and the current output image with four masks rendered. Mask 2 corresponds to
the \"stronger Mario\" character, which is the target of the user's query. Masks 1,
3, and 4 are not relevant to the query. \n\nSince I have identified the correct mask,
 I will proceed to select it as the final answer.\n\n</think>\n<tool> {\"name\": \"
select_masks_and_return\", \"parameters\": {\"final_answer_masks\": [2]}} </tool>"
        }
    ]
},
{
    "role": "user",
    "content": [
        {
            "type": "text",
            "text": "Final Answer:"
        },
        {
            "type": "image",
            "image": Figure {\ref{fig:agent_mario_3}}
        }
    ]
},
]
```

## H.3 FULL QUANTITATIVE RESULTS

We report the full quantitative experiment results of SAM 3 Agent on ReasonSeg in Tab. 40, OmniLabel in Tab. 41, and RefCOCO-Seg in Tab. 42. SAM 3 Agent achieves the best results on both ReasonSeg and OmniLabel in a zero-shot manner, without training on any referring expression segmentation or reasoning segmentation data. SAM 3 Agent also surpasses previous zero-shot state-of-the-art results on RefCOCO+ and RefCOCOg, and is close to best methods that leverage the training datasets. We hypothesize that on RefCOCO, where all masks come from the MSCOCO dataset and each query points to exactly one ground-truth object mask, training-based methods learn the specific dataset annotation biases. We show examples of such annotation biases in the RefCOCO-Seg datasets in Fig. 28. SAM 3 Agent, being a zero-shot method, is unable to exploit these (generally undesirable) biases.

| Model | | Training | | Val Set | | Test Set | | Test (Short) | | Test (Long) | |
|---|---|---|---|---|---|---|---|---|---|---|---|
| Name | Version | RES | ReasonSeg | gIoU | cIoU | gIoU | cIoU | gIoU | cIoU | gIoU | cIoU |
| SEEM | – | ✗ | ✗ | 25.5 | 21.2 | 24.3 | 18.7 | 20.1 | 11.5 | 25.6 | 20.8 |
| Grounded SAM | – | ✗ | ✗ | 26.0 | 14.5 | 21.3 | 16.4 | 17.8 | 10.8 | 22.4 | 18.6 |
| OVSeg | – | ✗ | ✗ | 28.5 | 18.6 | 26.1 | 20.8 | 18.0 | 15.5 | 28.7 | 22.5 |
| GLaMM | Vicuna 7B | ✓ | ✗ | 47.4 | 47.2 | – | – | – | – | – | – |
| SAM4MLLM | Qwen-VL 7B | ✓ | ✗ | 46.7 | 48.1 | – | – | – | – | – | – |
| SAM4MLLM | LLaVA1.6 8B | ✓ | ✗ | 58.4 | 60.4 | – | – | – | – | – | – |
| Seg-Zero | Qwen2.5-VL 3B | ✓ | ✗ | 58.2 | 53.1 | 56.1 | 48.6 | – | – | – | – |
| Seg-Zero | Qwen2.5-VL 7B | ✓ | ✗ | 62.6 | 62.0 | 57.5 | 52.0 | – | – | – | – |
| X-SAM | Phi3 3.8B | ✓ | ✓ | 56.6 | 32.9 | 57.8 | 41.0 | 47.7 | 48.1 | 56.0 | 40.8 |
| HyperSeg | Phi2 3B | ✓ | ✓ | 59.2 | 56.7 | – | – | – | – | – | – |
| Kang et al. | LLaVA1.5 7B | ✗ | ✗ | – | 52.4 | – | 48.7 | – | 48.0 | – | 49.1 |
| Kang et al. | LLaVA1.5 13B | ✗ | ✗ | – | 60.5 | – | 49.9 | – | 48.7 | – | 51.0 |
| LISA | LLaVA 7B | ✓ | ✗ | 44.4 | 46.0 | 36.8 | 34.1 | 37.6 | 34.4 | 36.6 | 34.7 |
| LISA | LLaVA 7B | ✓ | ✓ | 52.9 | 54.0 | 47.3 | 34.1 | 40.6 | 40.6 | 49.4 | 51.0 |
| LISA | LLaVA 13B | ✓ | ✗ | 48.9 | 46.9 | 44.8 | 45.8 | 39.9 | 43.3 | 46.4 | 46.5 |
| LISA | LLaVA 13B | ✓ | ✓ | 56.2 | 62.9 | 51.7 | 51.1 | 44.3 | 42.0 | 54.0 | 54.3 |
| LISA | Llama2 13B | ✓ | ✓ | 60.0 | 67.8 | 51.5 | 51.3 | 43.9 | 45.8 | 54.0 | 53.8 |
| LISA | LLaVA1.5 7B | ✓ | ✗ | 53.6 | 52.3 | 48.8 | 47.1 | 48.3 | 48.8 | 49.2 | 48.9 |
| LISA | LLaVA1.5 7B | ✓ | ✓ | 61.3 | 62.9 | 55.6 | 56.9 | 48.3 | 46.3 | 57.9 | 59.7 |
| LISA | LLaVA1.5 13B | ✗ | ✗ | 57.7 | 60.3 | 53.8 | 50.8 | 50.8 | 50.0 | 54.7 | 50.9 |
| LISA | LLaVA1.5 13B | ✓ | ✓ | 65.0 | **72.9** | 61.3 | 62.2 | 55.4 | 50.6 | 63.2 | 65.3 |
| RSVP | LLaVA1.6 7B | ✗ | ✗ | 59.2 | 56.7 | 56.9 | 50.7 | 47.9 | 42.0 | 58.4 | 53.0 |
| RSVP | Qwen2-VL 7B | ✗ | ✗ | 58.6 | 48.5 | 56.1 | 51.6 | 48.5 | 44.3 | 57.1 | 53.0 |
| RSVP | Gemini1.5-Flash | ✗ | ✗ | 56.9 | 49.2 | 57.1 | 59.2 | 47.3 | 40.2 | 60.2 | 65.6 |
| RSVP | GPT-4o | ✗ | ✗ | 64.7 | 63.1 | 60.3 | 60.0 | 55.4 | 50.4 | 61.9 | 62.5 |
| Gemini Seg | Gemini2.5 Flash | ? | ? | 28.3 | 13.3 | 30.6 | 9.2 | 16.5 | 8.0 | 35.0 | 9.5 |
| **SAM 3 Agent** | Qwen2.5-VL 7B | ✗ | ✗ | 62.2 | 49.1 | 63.0 | 53.5 | 59.4 | 43.5 | 64.1 | 56.2 |
| **SAM 3 Agent** | Qwen2.5-VL 72B | ✗ | ✗ | 74.6 | 65.1 | 70.8 | 64.0 | 70.3 | 55.7 | 71.0 | 66.3 |
| **SAM 3 Agent** | Llama4 Maverick | ✗ | ✗ | 68.5 | 61.5 | 67.1 | 60.9 | 66.8 | 59.4 | 67.2 | 61.3 |
| **SAM 3 Agent** | Qwen3-VL 8B Thinking | ✗ | ✗ | 68.5 | 57.4 | 70.2 | 67.3 | 70.5 | 62.5 | 70.2 | 68.6 |
| **SAM 3 Agent** | Qwen3-VL 235B Thinking | ✗ | ✗ | 76.6 | 67.0 | 73.7 | 71.8 | 75.1 | 67.2 | 73.3 | 73.0 |
| **SAM 3 Agent** | Gemini2.5 Pro | ✗ | ✗ | **77.0** | 66.0 | **74.0** | **72.1** | **75.8** | 67.5 | **73.4** | **73.2** |

Table 40: SAM 3 Agent experiments on the ReasonSeg dataset (Lai et al., 2024) for Reasoning Segmentation. Training-RES indicates whether the model has been fine-tuned on the Referring Expression Segmentation task. Training-ReasonSeg indicates whether the model has been fine-tuned on the ReasonSeg training set. The overall best performances are shown in **bold** and the best zero-shot performances for models not trained on the ReasonSeg training set are underlined. Notable baselines include: SEEM (Zou et al., 2023), Grounded SAM (Ren et al., 2024b), OVSeg (Liang et al., 2023), GLaMM (Rasheed et al., 2024), SAM4MLLM (Chen et al., 2024b), Seg-Zero (Liu et al., 2025), X-SAM (Wang et al., 2025a), HyperSeg (Wei et al., 2024), (Kang et al., 2025), LISA (Lai et al., 2024), RSVP (Lu et al., 2025) and Gemini-seg (Paul Voigtlaender, Valentin Gabeur and Rohan Doshi, 2025)

.

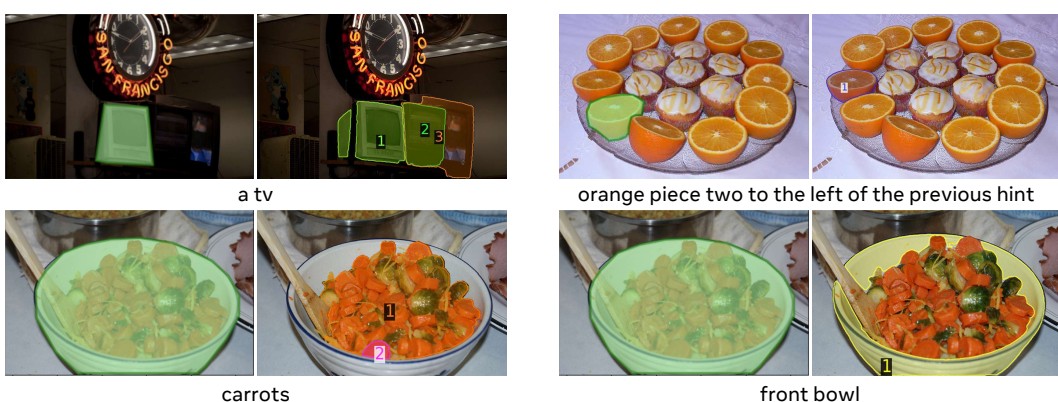

a tv

orange piece two to the left of the previous hint

carrots

front bowl

Figure 28: Examples of annotation bias and ground truth errors from the RefCOCO-Seg datasets (Kazemzadeh et al., 2014; Mao et al., 2016). For each example, see the original dataset ground truth annotation (left image), the textual user query (bottom text), and the SAM 3 Agent (Qwen2.5-VL 72B) final segmentation output (right image). Our error analysis reveals such annotation bias and ground truth errors account for the majority of low-IoU predictions by SAM 3 Agent on the RefCOCO-Seg datasets.

| Model | | 2023 Val Set (AP @ IoU=0.50:0.95) | | | | | | |
|---|---|---|---|---|---|---|---|---|
| Name | Version | AP | AP-categ | AP-descr | AP-descr-pos | AP-descr-S | AP-descr-M | AP-descr-L |
| FIBER | FIBER-B | 25.7 | 30.3 | 22.3 | 34.8 | 38.6 | 19.5 | 12.4 |
| GLIP | GLIP-T | 19.3 | 23.6 | 16.4 | 25.8 | 29.4 | 14.8 | 8.2 |
| GLIP | GLIP-L | 25.8 | 32.9 | 21.2 | 33.2 | 37.7 | 18.9 | 10.8 |
| Zhao et al. | GLIP-T | 22.2 | 27.2 | 18.8 | 29.0 | – | – | – |
| Zhao et al. | FIBER-B | 28.1 | 32.1 | 25.1 | 36.5 | – | – | – |
| DesCo | GLIP-T | 23.8 | 27.4 | 21.0 | 30.4 | – | – | – |
| DesCo | FIBER-B | 29.3 | 31.6 | 27.3 | 37.7 | – | – | – |
| GLEE | Lite | 20.3 | 37.5 | 14.0 | 19.1 | 23.0 | 12.7 | 10.0 |
| GLEE | Lite-Scale | 22.7 | 35.5 | 16.7 | 22.3 | 33.7 | 14.3 | 10.2 |
| GLEE | Plus | 25.4 | 46.7 | 17.5 | 23.9 | 28.4 | 16.3 | 12.5 |
| GLEE | Plus-Scale | 27.0 | 44.5 | 19.4 | 25.9 | 36.0 | 17.2 | 12.4 |
| Real | Swin-B | – | – | 36.5 | 52.1 | 54.4 | 33.2 | 25.5 |
| LED | Qwen2 | 27.9 | 33.9 | 23.7 | 36.2 | – | – | – |
| LED | InternLM2-2B | 27.9 | 33.4 | 23.9 | 36.1 | – | – | – |
| LED | InternLM2-6B | 26.3 | 32.0 | 22.4 | 34.3 | – | – | – |
| ROD-MLLM | Vicuna 7B | – | – | 25.3 | 30.9 | 31.8 | 24.5 | 21.0 |
| WSCL | GLIP-T | 24.3 | 23.9 | 24.7 | 34.4 | 39.3 | 21.6 | 16.4 |
| WSCL | FIBER-B | 30.5 | 31.6 | 29.5 | 40.3 | 43.7 | 26.3 | 21.3 |
| WSCL | Desco-GLIP | 26.5 | 27.1 | 25.9 | 35.6 | 38.1 | 23.2 | 18.7 |
| WSCL | Desco-FIBER | 32.0 | 33.1 | 30.9 | 40.4 | 45.2 | 27.7 | 22.9 |
| **SAM 3 Agent** | Qwen2.5-VL 7B | 38.8 | 41.1* | 36.7 | 42.8 | 52.6 | 34.3 | 26.6 |
| **SAM 3 Agent** | Qwen2.5-VL 72B | 41.6 | 41.1* | 42.0 | 52.6 | 56.0 | 40.4 | 33.2 |
| **SAM 3 Agent** | Llama4 Maverick | 36.5 | 41.1* | 32.8 | 43.0 | 43.7 | 30.9 | 27.5 |
| **SAM 3 Agent** | Qwen3-VL 8B Thinking | 40.9 | 41.1* | 40.7 | 52.9 | 53.3 | 39.5 | 32.4 |
| **SAM 3 Agent** | Qwen3-VL 235 Thinking | **44.4** | 41.1* | **48.4** | **59.4** | **58.0** | **47.8** | **41.5** |
| **SAM 3 Agent** | Gemini2.5 Pro | 43.1 | 41.1* | 45.3 | 63.8 | 53.8 | 45.1 | 37.7 |

Table 41: SAM 3 Agent experiments on the OmniLabel dataset (Schulter et al., 2023) (val 2023) for generalized referring expression comprehension (box prediction). * indicates predictions generated by the base SAM 3 model without MLLM. The overall best performances are shown in **bold**. Notable baselines include: FIBER (Dou et al., 2022), GLIP (Li et al., 2022b), (Zhao et al., 2024), DesCo (Li et al., 2023c), GLEE (Wu et al., 2024a), Real (Chen et al., 2025), LED (Zhou et al., 2025), ROD-MLLM (Yin et al., 2025), and WSCL (Park et al., 2024).

| Model | | Training | RefCOCO | | | RefCOCO+ | | | RefCOCOg | | |
|---|---|---|---|---|---|---|---|---|---|---|---|
| Name | Version | RES | val | testA | testB | val | testA | testB | val (U) | test (U) | val (G) |
| LISA | LLaVA 7B | ✓ | 74.9 | 79.1 | 72.3 | 65.1 | 70.8 | 58.1 | 67.9 | 70.6 | – |
| GSVA | 13B | ✓ | 79.2 | 81.7 | 77.1 | 70.3 | 73.8 | 63.6 | 75.7 | 77.0 | – |
| GLaMM | Vicuna 7B | ✓ | 79.5 | 83.2 | 76.9 | 72.6 | 78.7 | 64.6 | 74.2 | 74.9 | – |
| SAM4MLLM | LLaVA1.6 7B | ✓ | 79.6 | 82.8 | 76.1 | 73.5 | 77.8 | 65.8 | 74.5 | 75.6 | – |
| SAM4MLLM | LLaVA1.6 8B | ✓ | 79.8 | 82.7 | 74.7 | 74.6 | 80.0 | 67.2 | 75.5 | 76.4 | – |
| GLEE | Plus | ✓ | 79.5 | – | – | 68.3 | – | – | 70.6 | – | – |
| GLEE | Pro | ✓ | 80.0 | – | – | 69.6 | – | – | 72.9 | – | – |
| DETRIS | DETRIS-L | ✓ | 81.0 | 81.9 | 79.0 | 75.2 | 78.6 | 70.2 | 74.6 | 75.3 | – |
| UniLSeg | UniLSeg-20 | ✓ | 80.5 | 81.8 | 78.4 | 72.7 | 77.0 | 67.0 | 78.4 | 79.5 | – |
| UniLSeg | UniLSeg-100 | ✓ | 81.7 | 83.2 | 79.9 | 73.2 | 78.3 | 68.2 | 79.3 | 80.5 | – |
| PSALM | Phi1.5 1.3B | ✓ | 83.6 | 84.7 | 81.6 | 72.9 | 75.5 | 70.1 | 73.8 | 74.4 | – |
| EVF-SAM | RC | ✓ | 82.1 | 83.7 | 80.0 | 75.2 | 78.3 | 70.1 | 76.8 | 77.4 | – |
| EVF-SAM | Extra Data | ✓ | 82.4 | 84.2 | 80.2 | 76.5 | 80.0 | 71.9 | 78.2 | 78.3 | – |
| RICE | Qwen2.5-7B | ✓ | 83.5 | 85.3 | 81.7 | **79.4** | 82.8 | 75.4 | 79.8 | 80.4 | – |
| MLCD-seg | Qwen2.5-7B | ✓ | 83.6 | 85.3 | 81.5 | **79.4** | 82.9 | **75.6** | 79.7 | 80.5 | – |
| HyperSeg | Phi2 2.7B | ✓ | 84.8 | 85.7 | **83.4** | 79.0 | **83.5** | 75.2 | 79.4 | 78.9 | – |
| X-SAM | Phi3 3.8B | ✓ | **85.1** | **87.1** | **83.4** | 78.0 | 81.0 | 74.4 | **83.8** | **83.9** | – |
| GL-CLIP | ResNet-50 | ✗ | 32.7 | 35.3 | 30.1 | 37.7 | 40.7 | 34.9 | 41.6 | 42.9 | 44.0 |
| GL-CLIP | ViT-B/32 | ✗ | 32.9 | 34.9 | 30.1 | 38.4 | 42.1 | 32.7 | 42.0 | 42.0 | 42.7 |
| CaR | ViT-B/16 | ✗ | 33.6 | 35.4 | 30.5 | 34.2 | 36.0 | 31.0 | 36.7 | 36.6 | 36.6 |
| Ref-Diff | VAE | ✗ | 37.2 | 38.4 | 37.2 | 37.3 | 40.5 | 33.0 | 44.0 | 44.5 | 44.3 |
| TAS | ResNet-50 | ✗ | 39.9 | 42.9 | 35.9 | 44.0 | 50.6 | 36.4 | 47.7 | 47.4 | 48.7 |
| TAS | ViT-B/32 | ✗ | 39.8 | 41.1 | 36.2 | 43.6 | 49.1 | 36.5 | 46.6 | 46.8 | 48.1 |
| IteRPrimeE | – | ✗ | 40.2 | 46.5 | 33.9 | 44.2 | 51.6 | 35.3 | 46.0 | 45.1 | 45.8 |
| Pseudo-RIS | CRIS | ✗ | 39.8 | 44.8 | 33.0 | 42.2 | 46.3 | 34.5 | 43.7 | 43.4 | 43.8 |
| Pseudo-RIS | ETRIS | ✗ | 41.1 | 48.2 | 33.5 | 44.3 | 51.4 | 35.1 | 46.0 | 46.0 | 46.8 |
| LGD+DINO | ViT-B/32 | ✗ | 49.5 | 54.7 | 41.0 | 49.6 | 58.4 | 38.6 | 50.3 | 51.1 | 52.5 |
| VLM-VG | ResNet-50 | ✗ | 47.7 | 51.8 | 44.7 | 41.2 | 45.9 | 34.7 | 46.6 | 47.1 | – |
| VLM-VG | ResNet-101 | ✗ | 49.9 | 53.1 | 46.7 | 42.7 | 47.3 | 36.2 | 48.0 | 48.5 | – |
| HybridGL | ViT-B/32 | ✗ | 49.5 | 53.4 | 45.2 | 43.4 | 49.1 | 37.2 | 51.3 | 51.6 | – |
| Kang et al. | LLaVA-1.5 7B | ✗ | 74.2 | 76.5 | 70.4 | 62.5 | 65.2 | 56.0 | 64.2 | 68.1 | – |
| Kang et al. | LLaVA-1.5 13B | ✗ | 76.1 | 78.9 | 72.8 | 64.1 | 67.1 | 57.3 | 67.7 | 69.0 | – |
| **SAM 3 Agent** | Qwen2.5-VL 7B | ✗ | 59.4 | 64.3 | 55.0 | 51.4 | 57.0 | 44.9 | 57.2 | 58.8 | 59.7 |
| **SAM 3 Agent** | Qwen2.5-VL 72B | ✗ | 71.6 | 74.9 | 66.0 | 64.9 | 70.8 | 57.9 | 68.8 | 70.2 | 70.4 |
| **SAM 3 Agent** | Llama4 Maverick | ✗ | 71.7 | 76.5 | 66.5 | 65.1 | 71.2 | 57.6 | 67.7 | 68.7 | 68.4 |
| **SAM 3 Agent** | Qwen3-VL 8B Thinking | ✗ | 68.6 | 72.3 | 63.9 | 59.3 | 64.5 | 55.6 | 66.4 | 66.9 | 67.8 |
| **SAM 3 Agent** | Qwen3-VL 235 Thinking | ✗ | 73.9 | 77.0 | 69.7 | 66.9 | 70.9 | 62.2 | 72.1 | 72.9 | 72.5 |
| **SAM 3 Agent** | Gemini2.5 Pro | ✗ | 75.5 | 77.6 | 71.0 | 67.3 | 71.1 | 63.4 | 73.4 | 74.0 | 74.6 |

Table 42: SAM 3 Agent experiments on the RefCOCO / RefCOCO+ / RefCOCOg datasets (Kazemzadeh et al., 2014; Mao et al., 2016) for Referring Expression Segmentation (RES), metric is cIoU. Training-RES indicates whether the model has been fine-tuned on the RefCOCO/RefCOCO+/RefCOCOg segmentation datasets (notice that nearly all MLLMs were trained on RefCOCO/RefCOCO+/RefCOCOg bbox datasets). The overall best performances are shown in **bold** and the best zero-shot performances for models *not* trained on the RES task are underlined. SAM 3 Agent achieves state-of-the-art performance on RefCOCO+/RefCOCOg in the zero-shot setting, and is close to best fine-tuned models. Notable baselines include: LISA (Lai et al., 2024), GSVA (Xia et al., 2024), GLaMM (Rasheed et al., 2024), SAM4MLLM (Chen et al., 2024b), GLEE (Wu et al., 2024a), DETRIS (Huang et al., 2025), UniLSeg (Liu et al., 2024b), PSALM (Zhang et al., 2024c), EVF-SAM (Zhang et al., 2024b), RICE (Xie et al., 2025), MLCD-seg (An et al., 2024), HyperSeg (Wei et al., 2024), X-SAM (Wang et al., 2025a), GL-CLIP (Yu et al., 2023b), CaR (Sun et al., 2024), Ref-Diff (Ni et al., 2023), TAS (Suo et al., 2023), IteRPrimeE (Wang et al., 2025c), Pseudo-RIS (Yu et al., 2024), LGD+DINO (Li et al., 2025), VLM-VG (Wang et al., 2025b), HybridGL (Liu & Li, 2025), and (Kang et al., 2025).

# I  MODEL AND ANNOTATION CARDS

## I.1  DATA ANNOTATION CARD

**Task Formulation.**

1. *At a high level, what are the subjective aspects of your task?* There is ambiguity in the task. Annotators may have multiple valid interpretations of what should be masked for a given phrase. *E.g. If a person is wearing a backpack should a mask for the phrase 'person' include the backpack? If the person is standing next to a painting that contains a person, should that person be masked too?* We accept this ambiguity in the task, and in the SA-Co/Gold benchmark we use reviews from three different annotators to help capture multiple interpretations.

2. *What assumptions do you make about annotators?* Annotators worked full time on the annotation task, which allowed for frequently sharing feedback that led to improved annotations. Annotators were proficient in English and completed adequate research to understand concepts that they were not familiar with. This research allowed us to annotate more fine-grained or specific concepts, like car brands. Annotators were detail-oriented looking for all possible instances of the phrase in the image. We focused more on annotation quality over annotation speed to allow annotators to carefully look for all instances.

3. *How did you choose the specific wording of your task instructions? What steps, if any, were taken to verify the clarity of task instructions and wording for annotators?* We provided detailed guidelines that included numerous examples of correct and incorrect annotations. We broke down the task into different scenarios, and the expected outcome for each scenario. We made frequent guideline updates to handle ambiguities and address new corner cases surfaced by the vendor. The vendor trained the raters on the updated guidelines, and QA'ed the annotators to ensure adoption. We maintained a log of vendor-posed questions and answers around guideline clarifications. We met with the vendor weekly to provide feedback on annotation quality and surface common mistake patterns. This decreased repeat errors, and increased the quality of vendor QA's.

4. *What, if any, risks did your task pose for annotators and were they informed of the risks prior to engagement with the task?* Annotators were instructed to reject objectionable content and flag phrases that were harmful or offensive to ground.

5. *What are the precise instructions that were provided to annotators?* The instructions varied for each of the annotation tasks. For images, we had annotators work on three separate tasks. 1) Verify the quality of masks for a given phrase in an image 2) Check if masks were exhaustively annotated in an image for a given phrase and 3) Add any missing masks and correct mask annotations such that all instances of the phrase were masked in the image. For video, there were two separate tasks. 1) Exhaustively annotate all instances of the phrase in the video and 2) Verify whether all instances are annotated with high-quality masklets in the video.

**Selecting Annotations.**

1. *Are there certain perspectives that should be privileged? If so, how did you seek these perspectives out?* All annotators had a minimum of B-2 English proficiency. Annotators had previous segmentation experience. Annotators researched fine-grained concepts that they were unfamiliar with.

2. *Are there certain perspectives that would be harmful to include? If so, how did you screen these perspectives out?* No.

3. *Were sociodemographic characteristics used to select annotators for your task? If so, please detail the process.* No.

4. *If you have any aggregated socio-demographic statistics about your annotator pool, please describe. Do you have reason to believe that sociodemographic characteristics of annotators may have impacted how they annotated the data? Why or why not?* We worked with annotators based in APAC and EMEA. The sociodemographic characteristics of annotators may have some impact on the annotated data. Across different regions, words can differ in their meanings and the same concept may look different across regions.

5. *Consider the intended context of use of the dataset and the individuals and communities that may be impacted by a model trained on this dataset. Are these communities represented in your annotator pool?* Our annotator pool does not represent all communities that will use the SAM 3. Annotators researched concepts they were unfamiliar with. When annotators were unsure, they researched concepts to better understand different visual representations of the concept. If annotators were still unsure, they rejected the job as unsure in order to make sure that our annotations only contained confident responses. Annotators flagged concepts that were harmful in context of the image or the video.

**Platform and Infrastructure Choices.**

1. *What annotation platform did you utilize? At a high level, what considerations informed your decision to choose this platform? Did the chosen platform sufficiently meet the requirements you outlined for annotator pools? Are any aspects not covered?* We used an internal annotation platform.

2. *What, if any, communication channels did your chosen platform offer to facilitate communication with annotators? How did this channel of communication influence the annotation process and/or resulting annotations?*

   The research team QA'ed the vendors' quality team and annotators, shared feedback and met weekly with the vendor to align on the guidelines, clarify ambiguities and surface common mistake patterns. The research team maintained a spreadsheet where they answered the vendor's questions requiring the desired annotations for specific jobs that were corner cases or ambiguous. The guidelines were frequently updated to include new corner cases surfaced. These processes helped align the vendor to our desired output, which allowed the vendor to more effectively QA the annotators and provide per-annotator feedback which decreased repeat errors. A chat thread was also maintained between the research team and vendor.

3. *How much were annotators compensated? Did you consider any particular pay standards, when determining their compensation? If so, please describe.* The annotators were compensated with an hourly wage set by the vendor.

**Dataset Analysis and Evaluation.**

1. *How do you define the quality of annotations in your context, and how did you assess the quality in the dataset you constructed?*

   Annotation quality was based more on ensuring completeness and validity of the annotator's interpretation. We defined quality across three axes: 1) mask quality (e.g. masks should not have holes or missing pieces) 2) mask concept correctness (e.g there should not be a mask around a dog when the concept is "cat") 3) mask exhaustivity (e.g. all instances including small background instances should be masked). We set a high bar for quality, and aligned with the vendor on the requirements of a correct annotation.

For each task, annotators underwent a 2-day training session led by the vendor, followed by annotating jobs from a training queue. They were only eligible to move into the production annotation queues after the vendor QA or research team reviewed their annotations and approved of the qualityThe vendor QA team continuously reviewed production annotations, covering covering 10% avg of all annotations, ranging from 5% - 20% dependent on task complexity across the duration of the program. The research team manually reviewed small subsets of the production annotations and shared feedback weekly.

We ensure high data quality by letting annotators reject low confidence or vague annotations. Annotators were asked to research concepts they were unfamiliar with or unsure if they were present in the media. If, after researching, annotators were still unsure or the concept was considered vague (e.g. "sunlight") the annotators were instructed to reject the job as unsure.

In addition, all video annotations are manually reviewed and only those that meet the criteria across all three axes are accepted for use in training and evaluation, ensuring that the dataset contains only high-quality, validated data.

2. *Have you conducted any analysis on disagreement patterns? If so, what analyses did you use and what were the major findings? Did you analyze potential sources of disagreement?* The PCS task is inherently ambiguous and the sources of annotation disagreement are explained in §2. For SA-Co/Gold, we recorded 3 independent annotator responses in order to capture the ambiguity in the task and measured their agreement. See the human metrics discussion in §F.4 for more details.

3. *How do the individual annotator responses relate to the final labels released in the dataset?* For images, each image, phrase pair underwent three annotation tasks: 1) mask quality verification of masks pseudo-labeled by SAM-3 PL masks 2) exhaustivity verification of accepted masks for the phrase and 3) manually adding missing masks until all instances of the concept were masked. For video annotations, individual annotator responses are validated and only annotations that meet the required quality criterion are accepted. The final labels released in the video subset consist exclusively of these accepted annotation.

*For SA-Co/Gold*: Gold subsets were 3x multi-reviewed. All three annotators had to accept a mask as high quality for it to be included in the dataset. Image and phrase pairs that at least one of the raters marked as non-exhaustively annotated in step two were sent to manual annotation. All three of the annotators' responses were saved as separate versions of the annotations for the give image and phrase pair. This helped capture the natural ambiguity in the task and alternative but valid interpretations of a concept.

*For SA-Co/Silver*: Silver subsets were 1x reviewed. The first annotator's accepted masks were given to the exhaustivity annotator. If the exhaustivity annotator marked the phrase and its associated set of masks as exhaustive, this was the

**Dataset Release and Maintenance.**

1. *Do you have reason to believe the annotations in this dataset may change over time? Do you plan to update your dataset?* The dataset contains annotations for public images or external public datasets. Some images or datasets may become unavailable over time.

2. *Are there any conditions or definitions that, if changed, could impact the utility of your dataset?* Concepts may change visually - for example masks annotated with *'smartphone'* may no longer represent a modern day smartphone. New phrases or types of items will not be represented.

3. *Will you attempt to track, impose limitations on, or otherwise influence how your dataset is used? If so, how?* Our benchmark is for model evaluation, and should not be used for training.

4. *Were annotators informed about how the data is externalized? If changes to the dataset are made, will they be informed?* No.

5. *Is there a process by which annotators can later choose to withdraw their data from the dataset? If so, please detail.* No.

## I.2 MODEL CARD

**Model Overview**

| | |
|---|---|
| Name | SAM 3 (Segment Anything Model 3) |
| Version | 1.0 |
| Date | 2025 |
| Organization | Meta SAM Team |
| Mode type | Promptable segmentation model |
| Architecture | See Section 3 |
| Repository | `https://github.com/facebookresearch/sam3` |
| License | SAM license (detailed in the SAM 3 repository) |

**Intended Use**

| | |
|---|---|
| Primary intended users | SAM 3 was designed as a model for promtable concept segmentation (PCS) and promptable visual segmentation in images and videos. The model was primarily developed for research use cases. SAM 3 is released under the license detailed in the SAM 3 repository. |
| Out-of-scope use cases | SAM 3 does not support complex text queries or queries including referring expressions. See license for restrictions. |
| Caveats and recommendations | SAM 3 generally performs well in zero-shot settings, but will benefit from fine-tuning for niche domains. In video, the cost of inference scales linearly with the number of objects being tracked. See §C for detailed discussion around limitations. |

**Relevant Factors**

| | |
|---|---|
| Groups | SAM 3 was developed for open world vocabulary. See §C for limitations on text prompts. |
| Instrumentation and environment | SAM 3 was trained on a diverse set of media pools that vary in instrumentation (e.g. medical instruments, underwater imagery, camera traps) and settings (e.g everyday scenes, driving, robotics, wildlife). |

**Metrics**

We evaluate the performance of SAM 3 using metrics tailored to the specific task. For image tasks, we use the following metrics:

- *PCS in Images with Text*: $cgF_1$, IL_MCC, $pmF_1$, AP. See §F.3 for details.
- *PCS in Images with Exemplars*: $cgF_1$, AP. See §F.3 for details.
- *Instance Segmentation, Box Detection, Few Shot Box Detection*: $cgF_1$, AP
- *Semantic and Interactive Instance Segmentation (SAM 1 task)*: mIoU
- *Counting*: MAE, Accuracy

For video tasks, we use the following metrics:

- *PCS in Videos*: $cgF_1$, pHOTA, mAP. See §G.5 for details.
- *VOS*: $\mathcal{J}\&\mathcal{F}$, $\mathcal{G}$
- *Promptable video segmentation (SAM 2 task)*: offline average $\mathcal{J}\&\mathcal{F}$, online average $\mathcal{J}\&\mathcal{F}$

**Evaluation Data**

| | |
|---|---|
| Data sources | For promptable concept segmentation, SAM 3 is evaluated using the SA-Co benchmark which is composed of: |

- SA-Co/Gold
- SA-Co/Silver
- SA-Co/Bronze
- SA-Co/Bio
- SA-Co/VEval

Each of these subsets is composed of a series of data sources, see §F.2 for details. The SA-Co benchmark annotations are released publicly at `https://github.com/facebookresearch/sam3`

**Training Data**

| | |
|---|---|
| Data source | SAM 3 is trained using the following datasets: |

- SA-Co/HQ
- SA-Co/EXT
- SA-Co/SYN
- SA-Co/VIDEO
- SA-Co/VIDEO-EXT
- SA-1B

SA-Co/EXT, SA-Co/VIDEO, SA-Co/SYN, SA-Co/HQ, SA-Co/VIDEO-EXT are composed of multiple data sources, see Table 26 in §F.1 for the list of data sources for subset. See Table 18 in §D.4.1 for details as to which datasets are used during which training stages.

**Ethical Considerations**

| | |
|---|---|
| Data | See §I.1 for data annotation card |
| Cost and impact of compute | The released SAM 3 was trained on 172k A100 GPU hours and 86k H200 GPU hours. This corresponds to an estimated 142k-176k kWH and emissions between 65.3 and 77.6 metric tons of $CO_2e$ (Patterson et al., 2021; Lacoste et al., 2019). The emissions from training the released SAM 3 model are equivalent to 166k miles driven by an average gasoline-powered passenger vehicle (Agency, 2022). |
| Risks and harms | The behavior of SAM 3 is undefined for opinionated or subjective queries (e.g. *magnificent painting*) and should not be used to make subjective judgments about objects or people. Users should evaluate the safety of SAM 3 tailored to their specific use case. |
| Use cases | We implore users to use their best judgment. |

Table 43: Model card for SAM 3 following the structure in (Mitchell et al., 2019)

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
