# OpenReview forum: "SAM 3: Segment Anything with Concepts"
_ICLR.cc/2026/Conference — ICLR 2026 Poster_

### Official Review · Reviewer_wjcv · 2025-10-27

**Soundness:** 4
**Presentation:** 3
**Contribution:** 4
**Rating:** 8
**Confidence:** 5

**Summary:**

This work proposes a new task called Promptable Concept Segmentation (PCS), which segments target objects through language prompts or image examples. Authors build a scalable data engine to automatically generate training data. They also present a new model, SAM3, to implement PCS. SAM3 has achieved excellent performance in both image and video segmentation tasks.

**Strengths:**

1. Generalized prompts. Compared with previous works, PCS has a wider range of prompts and SAM3 achieves better performance.
2. Data engine. The data engine proposed by authors can effectively expand data, including mask and text, which is beyond the capability of previous works.
3. SAM3 achieves PCS for both images and videos through a detection-then-tracking paradigm.
4. SAM3 achieves sota performances over several tasks and benchmarks.

**Weaknesses:**

1. The formulation of SAM3. This detection-then-tracking approach of SAM3 follows a two-stage format. It inevitably introduces errors and requires hyperparameter in the merging stage. The paradigm of SAM3 could be further adjusted to implement PCS in an end-to-end manner instead of a two-stage one.
2. Complex naming conventions. The naming of training and test data in the paper is overly complex, with numerous similar names. Additionally, due to space limitations, authors did not elaborate on the meanings of different names, which makes this quite confusing.
3. Missed citation. Many of the mentioned datasets or benchmarks are not accompanied by citations in the references.
4. Multi-stage training, which trains different parts of SAM3. A final end-to-end training that fine-tunes the entire model, may further improve performance.

**Questions:**

1. Do you have the plan to release the code and model? I think it is quite import for the community.
2. It seems that the concept of SAM3 is limited to simple text and cannot perform complex reasoning like ReasonSeg. Authors have used an agent-based approach to achieve this. However, authors did not show the performance of using SAM3 directly for complex reasoning, which I find quite curious.

---

> ### Author Response · Authors · 2025-11-24
> **Response to reviewer wjcv**
>
> Thank you for your feedback and insightful comments!
>
> We agree that the dataset naming has become complex. We will try to simplify it for the final version and include citations in the main text  (due to space constraints they were moved to the appendix).
>
> # Question 1
> > Do you have the plan to release the code and model?
>
> Yes! The code, model and evaluation benchmarks will be released.
>
> # Question 2
>
> > authors did not show the performance of using SAM3 directly for complex reasoning,
>
> We have an evaluation of SAM 3 vs SAM 3 Agent in Table 8 in the paper, also reproduced below, on ReasonSeg and on OmniLabel. SAM 3 Agent is significantly better than SAM 3, except on the OmniLabel short-description split (which are mostly short noun phrases that SAM 3 can deal with directly). The longer the text query is, the larger the gap is.
>
> On ReasonSeg (gIOU)
>
> | Model      |         Agent   | Val  | test-all | test-short | test-long |
> |------------|-----------------|------|----------|------------|-----------|
> | SAM 3      | None            | 14.2 | 9.2      | 21.2       | 5.4       |
> | SAM 3 Agent| Gemini 2.5 Pro  | 76.0 | 73.8     | 74.0       | 73.7      |
>
> On OmniLabel (AP)
>
> | Model      |         Agent   | descr | descr-S | descr-M | descr-L |
> |------------|-----------------|-------|---------|---------|---------|
> | SAM 3      | None            | 27.0  | 54.0    | 22.5    | 13.5    |
> | SAM 3 Agent| Gemini 2.5 Pro  | 46.7  | 54.6    | 46.2    | 38.7    |

---

> > ### Comment · Reviewer_wjcv · 2025-11-28
> >
> > Authors' response has addressed my concerns.
> > However, I think authors may have misunderstood the meaning of my Weakness 3. It is acceptable to place citations in the appendix; my point is that some relevant works are not included in the references at all, meaning authors appear to have failed to cite them even in the appendix. Authors should carefully check all citations in the revised manuscript.
> > I will keep my rating, because I think it is engough.

---

> > > ### Author Response · Authors · 2025-12-03
> > > **follow up with additional experiments**
> > >
> > > Thank you for the clarification and bringing this to our attention! It was indeed a misunderstanding on our part, we will carefully check citations for each dataset and benchmark and include them. We also ran some additional experiments and wanted to follow up on our initial response to the following points you raised:
> > >
> > > > Multi-stage training, which trains different parts of SAM3. A final end-to-end training that fine-tunes the entire model, may further improve performance.
> > >
> > > This is a good discussion point. We designed the multi-stage training curriculum of SAM 3 with **efficiency** in mind. Early stages of training (stages 2, 3) establish robust *spatial localization* capabilities, before progressing to learning *spatio-temporal* localization capabilities in videos which is more memory- and compute-intensive.
> > >
> > >
> > > Critically, each component still sees *all* relevant data sources—for example, the detector in stages 2-3 trains on the SA-Co/VIDEO data (on individual frames), as detailed in Table 20. This provides benefits similar to end-to-end fine-tuning while maintaining computational efficiency. To demonstrate the significant impact of this curriculum, we trained a version of SAM 3 excluding SA-Co/VIDEO data in training the detector, showing significantly weaker performance. We acknowledge that an end-to-end finetuning stage may yield further small improvements and leave this for future work.
> > >
> > > | Detector trained on SA-Co/VIDEO | SA-V | YT-T-1B | SmartGlasses | LVVIS | BURST | YTVIS21 | OVIS |
> > > |:------------------------|-----:|--------:|-------------:|------:|------:|--------:|-----:|
> > > | no                      | 31.7 | 48.1    | 35.9         | 29.6  | 39.4  | 47.8    | 47.7 |
> > > | yes                     | 34.9 | 50.9    | 37.9         | 36.3  | 44.5  | 57.4    | 60.5 |
> > >
> > > **Metrics:** SA-V (test cgF1), YT-T-1B (test cgF1), SmartGlasses (test cgF1), LVVIS (test mAP), BURST (test HOTA), YTVIS21 (val mAP), OVIS (val mAP).
> > >
> > > > The formulation of SAM3. This detection-then-tracking approach of SAM3 follows a two-stage format. It inevitably introduces errors and requires hyperparameter in the merging stage. The paradigm of SAM3 could be further adjusted to implement PCS in an end-to-end manner instead of a two-stage one.
> > >
> > >
> > > Yes, SAM 3 embraces a *decoupled design* for detection and tracking, which avoids task conflict between components (the detector needs to be identity-agnostic, while the tracker's main objective is to separate identities in a video). While we acknowledge this formulation has limitations, **this design choice enables modular training with minimal annotation demands**—a highly desirable property for practical applications.
> > > To demonstrate this, we compare below a version of SAM 3 trained with and without SA-FARI (a *fine-grained* multi-animal tracking dataset). Critically, we train *only the detector* on SA-FARI frames (not the tracker, so this data is never used as *video*), yet observe dramatic gains in video grounding performance. This shows that users extending SAM 3 to niche or fine-grained video domains need only annotate a modest number of individual frames (which even be done automatically by our data engine at near human-level, as shown in appendix B.3), finetune the detector, and immediately gain *video grounding capability*.
> > >
> > >
> > > | Method                       | cgF₁ | pHOTA Total | pHOTA Det | pHOTA Ass | TETA |
> > > |------------------------------|------|--------------|------------|------------|-------|
> > > | **SAM 3** (w/o SA-FARI)                  | 14.0 | 48.5        | 28.4       | 83.4       | 39.6  |
> > > | **SAM 3 (FT on SA-FARI)**     | **46.9** | **68.1** | **55.4** | **84.6** | **58.7** |
> > >
> > > Thank you for raising these great discussion points, and for reading our responses!

---

### Official Review · Reviewer_ma9q · 2025-10-29

**Soundness:** 4
**Presentation:** 3
**Contribution:** 3
**Rating:** 6
**Confidence:** 4

**Summary:**

This work introduces SAM 3, a new extension of the "Segment Anything Model" paradigm that formalizes and successfully tackles the promptable concept segmentation task.
This moves the field beyond single-object segmentation to the more complex challenge of detecting, segmenting, and tracking all instances matching a given concept text prompt.
This advance is underpinned by a scalable data engine to generate a massive, high-quality dataset.
The refined model architecture effectively decouples recognition and localization for better open-vocabulary segmentation.
The method demonstrates a good performance gain on this new task and this paper establishes the new and large-scale SA-Co benchmark.

**Strengths:**

- The introduction of the "presence head" is a specific and impactful architectural contribution. This design choice directly addresses the challenge of open-vocabulary detection by decoupling the recognition from the localization, which the ablations show is effective.
- The paper demonstrates compelling quantitative results, achieving a great performance gain on the PCS task and setting a new state-of-the-art on existing benchmarks. These results validate the efficacy of the complete system.
- The SAM 3 model and the large-scale SA-Co benchmark are valuable. This provides a valuable new asset for the community.
- While other works have explored text-prompted segmentation, this paper successfully integrates concept-level segmentation and tracking into a single, unified SAM model. The ability to scale this integration to such a large dataset and achieve robust performance is a notable engineering accomplishment.

**Weaknesses:**

Given that this paper can be viewed as the latest advancement in the SAM series, a critical point of evaluation is the extent of its novelty and extension over prior work (SAM 1 & 2). The contributions can be broadly categorized into three areas: (1) task definition, (2) data benchmark, and (3) model design.

1. On the task definition: The paper extends promptable visual segmentation to promptable concept segmentation. However, the integration of text prompts (in addition to conventional interactive prompts) has already been explored by numerous existing works [1,2,3]. This makes the novelty of this specific task definition, particularly for the SAM series, appear limited.
1. On the data benchmark: What are the specific differences and unique innovations of the data engine workflow (Sec. 4) compared to the pipelines used for SAM 1 & 2? The primary distinction appears to be the incorporation of text, which seems like a necessary adaptation for the new task rather than a fundamental innovation in the data engine itself. Furthermore, existing works [1,2,3] have also constructed large-scale datasets for the similar tasks. The paper fails to adequately discuss or differentiate its data collection process (in terms of workflow and details) from these prior efforts.
1. On the model design: Existing research [4] has shown that SAM models can be sensitive to perturbations in interactive prompts. This paper also lacks an investigation into this aspect. How does SAM 3 perform when faced with perturbations in its text or visual prompts, such as variations in phrasing (expression deviations) or positional shifts (spatial deviations)?

REF:
1. Sa2VA: Marrying SAM2 with LLaVA for Dense Grounded Understanding of Images and Videos, 2025
1. VoCap: Video Object Captioning and Segmentation from Any Prompt, 2025
1. Perceive Anything: Recognize, Explain, Caption, and Segment Anything in Images and Videos, 2025
1. Inspiring the Next Generation of Segment Anything Models: Comprehensively Evaluate SAM and SAM 2 with Diverse Prompts Towards Context-Dependent Concepts under Different Scenes, 2024

**Questions:**

1. What is meant by the use of "group masks" as mentioned in Lines 246-248?
1. For the evaluation of zero-shot capabilities, the authors are advised to include a performance comparison on the MESS [1] benchmark. This benchmark covers a diverse range of target domains, which would provide a more comprehensive perspective on the model's generalization abilities.
1. The paper restricts its analysis to specific object counting benchmarks. This ignores the long-standing, highly challenging, and practical domain of crowd counting [2] (e.g., on datasets like UCF-QNRF,  JHU-Crowd, ShanghaiTech, or UCFCC50). These dense, highly-occluded scenarios are the true stress test for any model claiming robust counting abilities.

REF:
1. What a MESS: Multi-Domain Evaluation of Zero-Shot Semantic Segmentation, 2023
1. Revisiting crowd counting: State-of-the-art, trends, and future perspectives, 2023

---

> ### Author Response · Authors · 2025-11-24
> **Response to reviewer ma9q [Part 1]**
>
> Thank you for your feedback and questions!
>
> # Task Novelty
> > The integration of text prompts (in addition to conventional interactive prompts) has already been explored
>
> To clarify, we do not claim that the Promptable Concept Segmentation (PCS) task is novel. Indeed, the text-prompt version is similar to various forms of open-vocabulary object detection and instance segmentation tasks proposed before. It does differ from referring expression tasks, in two ways:
>
>   1. in our task the text prompts are simple noun phrases with emphasis on broad concepts like “green apples,” rather than referring expressions describing a specific instance like “the green apple sitting on the table”,
>
>   2. In our task the output should contain all instances of the concept, whereas in referring expression benchmarks the output is usually a single mask.
>
> Our contribution lies in the following:
> * **A precise formalization** of the “concept prompt” as any short noun phrase, i.e., a noun with a few modifier nouns or adjectives. This opens the vocabulary to a huge number of options, but still limits the scope to general object detection rather than understanding longer referring expressions and reasoning about the object’s relationships to other scene elements. We also include image exemplars as an alternative prompt modality, and make the task interactive: users can refine predictions with positive or negative exemplars of the concept. Note that exemplars define categories, as opposed to visual prompts which define single instances.
>
> * **New metrics and benchmarks** to drastically expand the vocabulary and measure progress on the task. No benchmark of that scale and quality exists today, severely slowing down progress. Our new, open-sourced SA-Co benchmark has 270K unique concepts, over *50 times* more than existing benchmarks, all exhaustively annotated with multiple instance masks. It also includes many hard negative prompts to evaluate the models’ ability to do open vocabulary recognition (lobster vs crab). Our new classification-gated F1 (cgF1) metric helps evaluate large-vocabulary recognition by adding an image-level MCC score to the traditional F1 score.
>
> * **A major step change in performance.** No baseline available today comes close to rivaling the performance of SAM3 on the breadth of tasks it supports. We are releasing this model under a permissive license to enable the community to further build on it. Similar to how CLIP upleveled all following work on image recognition by providing a high-performing, highly generalizable baseline, we hope that SAM 3 can serve in a similar capacity.
>
> * **A single multi-task model.** SAM 3 seamlessly supports six segmentation tasks in one model:
>   * Visual segmentation in images (with multiple positive or negative clicks)
>   * Visual segmentation and tracking in video (with multiple positive or negative clicks
>   * Text concept segmentation in images
>   * Exemplar concept segmentation in images (with multiple positive or negative exemplars)
>   * Text concept segmentation and tracking in video
>   * Exemplar concept segmentation and tracking in video (with multiple positive or negative exemplars)
>
> To the best of our knowledge, no existing model can support all six tasks. GLEE (Wu et al., 2024a) allows text phrases, referring expressions, and visual prompts for category and instance segmentation in both images and videos. However unlike SAM 3, GLEE does not support exemplar prompts for specifying an object category without text. SAM 3 also allows interactive refinement with multiple exemplars after the initial text prompt. T-Rex2 (Jiang et al., 2024) supports exemplars (positive only), however, does not integrate them with the other tasks, like visual prompts.
>
> # Question 1. Group masks
> > What is meant by the use of "group masks" as mentioned in Lines 246-248?
>
> These are special masks that cover more than a single instance in an image. Normally, annotators create a separate mask for each concept instance in the image. In rare cases, when objects are small, cluttered together and hard to separate (like a pile of leaves or a crowd or people), annotators can use these special group masks to indicate which pixels contain the concept without separating them into instances. In our benchmark, we hold out phrases with group masks from the instance segmentation evaluations, but use them for semantic segmentation evals.

---

> ### Author Response · Authors · 2025-11-24
> **Response to reviewer ma9q [Part 2]**
>
> # Data Engine Novelty
> > What are the specific differences and unique innovations of the data engine workflow (Sec. 4) compared to the pipelines used for SAM 1 & 2?
>
> SAM 1&2 advanced the Promptable *Visual* Segmentation (PVS) task – using points, boxes or masks to *visually* segment a *single* object per prompt. SAM 3's key focus is on the Promptable *Concept* Segmentation (PCS) task (section 2) and addresses the task of *linguistic* open-vocabulary object detection and instance segmentation. For example, to segment and track *all* cars in a video, SAM 2 requires manually clicking on each car, while SAM 3 just needs a single prompt “car”.
>
> Advancing progress on the PCS task thus requires a data engine *fundamentally different* from SAM 1&2. To the best of our knowledge, our data engine is 1) *novel* and, much more importantly, 2) *highly effective* and 3) *efficient*. In more detail,
>
> 1) **Novel**. Previous approaches (e.g. DINO-X, Grounding DINO 1.5, 1.6) in this space that used a data engine to scale up annotations have sophisticated pipelines but share very little detail, while others such as OWLv2 have relied on simple parsing of alt-text associated with images together with a grounding model for localization. To create SAM 3 we built a novel pipeline that includes custom AI mask and exhaustivity verifiers, captioners, parsers, an ontology and hard negative proposers. *None of these components are a part of SAM 1 and 2 data engines* (nor are they detailed by any related work we are aware of).
>
> 2) **Highly Effective**. Beyond novelty, our data engine is highly effective. We ran additional experiments to demonstrate this. We generated several versions of training annotations for a domain not seen in training (“food & drink”), trained SAM 3 models on these datasets, and evaluated them on the test set.
> |    training data  | cgF1 on SA-Co|
> |-------------------|--------------|
> |a) our data engine | 65 |
> |b) no AI verifiers | 50 |
> |c) no ontology/negatives | 47 |
> |d) baseline/init | 45 |
>
> We find using (a) our full data engine with AI mask and exhaustivity verifiers, our ontology and hard negative proposer (65 cgF1) is *dramatically* better than (b) leveraging our ontology and negative proposer alone (50 cgF1), while (c) a baseline not using our ontology or negative proposer performs poorly (47 cgF1), little better than the baseline/initialization (45 cgF1). Baseline (c) is comparable to OWLv2’s data engine. *Our data engine produces dramatically better performance*. In Figure 7a (appendix B.3), we demonstrate *robust scaling* behaviour of our data engine, providing high quality annotations that enable performance on-par with *human-labelled annotations*.
>
> 3) **Efficient**. Our AI annotators are *5x faster* than humans on negative prompts (concepts not present in the image/video) and *36% faster* for positive prompts, even in challenging fine-grained domains. (see appendix B.4)
>
> # Robustness to perturbations
> > How does SAM 3 perform when faced with perturbations in its text or visual prompts, such as variations in phrasing (expression deviations) or positional shifts (spatial deviations)?
>
> A comprehensive study on the sensitivity to perturbation is out of scope for this paper, as there are no existing benchmarks that could give a clear signal. We can however give more details on the expected robustness of the model:
> * Robustness to cultural/geographical shifts: we used the gold standard GEODE dataset as part of our SA-CO/silver benchmark, ensuring that the model doesn’t have degraded performance on common objects when they occur on geographically diverse situations.
>
> * Robustness to image distribution: we evaluate SAM3 on COCO-O (AP$_\text{O}$, Table 1), which studies various domain shifts (sketch, cartoon, extreme weather), and found that it still performs very well
>
> * Robustness to text formulation. We do some data augmentation to add synonyms and vary the articles (eg “a dog”/”the dog”/”dog”). The model is also reasonably robust to typos thanks to the CLIP-style pre-training
>
> * Robustness to perturbations in the input box: we conduct an ablation where we add noise to the GT box provided to the model when doing detection on COCO using image prompts. The noise is scaled by the dimension of the box, and we clamp to ensure a minimal box size of 10 pixel^2. The results are in the table below:
>
> | Noise std                | 0    | 1.0  | 5.0  | 10   | 50   | 100  |
> |--------------------------|------|------|------|------|------|------|
> | COCO AP (exemplar input) | 76.2 | 76.2 | 75.4 | 71.8 | 60.1 | 57.2 |
>
> This shows that SAM3 is robust to mild perturbations, and it takes a really high amount of noise to see a decrease in performance.

---

> ### Author Response · Authors · 2025-11-24
> **Response to reviewer ma9q [Part 3]**
>
> # Question 2: Evaluate on MESS
> > For the evaluation of zero-shot capabilities, the authors are advised to include a performance comparison on the MESS [1] benchmark. This benchmark covers a diverse range of target domains
>
> We thank the reviewer for the suggestion. We would like to point out that in our opinion, the set of evaluations reported in the paper already covers “a diverse range of target domains”. Our SACO splits target self-driving (bdd100k), fine-grained animal detection (inaturalist, fathomnet), medical/biology/micro biology domains (all the datasets in SACO/bio). We also report on ODinW and RF100, which include various industrial domains (amongst others) and on COCO-O to study robustness under domain shift (such as art or extreme weather).
>
> We also note that MESS is a *semantic segmentation* benchmark, and while SAM3 is capable of tackling that, it’s not our main focus. We optimized mainly the instance segmentation task, as well as its interactive variants.
>
> Following the reviewer’s suggestion, we evaluate SAM 3 on MESS. Below we report results using the default mIoU metric. We did not manage to download one of the sub-datasets, so we limit the evaluation to 21 datasets. We compare to CAT-Seg-L, a baseline specific to semantic segmentation (unlike SAM3, it cannot do instance segmentation), as well as Grounded-SAM-L. We note that the H size version of these models perform worse than L, so we only report that one.
>
> | Model          | BDD100K | Dark Zurich | MHP v1 | FoodSeg103 | ATLANTIS | DRAM  | iSAID | WorldFloods | FloodNet | UAVid | Kvasir-Instrument |
> |----------------|---------|-------------|--------|------------|----------|-------|-------|-------------|----------|-------|-------------------|
> | CAT-Seg-L      | 45.8   | 33.1        | 30.0  | 30.5      | 33.6     | 66.5 | 16.1 | 49.9       | 39.8    | 42.0 | 79.4    |
> | Grounded-SAM-L | 42.7   | 21.9       | 28.1  | 10.8      | 17.6    | 60.8  | 12.4 | 33.4        | 19.3    | 39.4 | 47.3    |
> | SAM3           | 65.9   | 45.6        | 49.6  | 37.3      | 44.4    | 79.3 | 32.3  | 34.4       | 29.2    | 53.9 | 49.7             |
>
> *(table continued below)*
>
> | Model       |CHASE DB1 | CryoNuSeg | PAXRay-4 | Corrosion CS | DeepCrack | PST900 | ZeroWaste-f | SUIM  | CUB-200 | CWFID | _Average_ |
> |-----------|----------|--------------|-----------|--------|-------------|-------|---------|-------|---------|-----------|-----------|
> | CAT-Seg-L      | 25.0     |35.1     |54.5     | 16.9        | 31.4     | 25.3  | 30.6       | 53.9 | 9.2    | 39.0    | 37.5   |
> | Grounded-SAM-L  | 25.2     |  38.1   | 44.2    | 20.9        | 58.2     | 21.2  | 16.7       | 14.3  | 0.4    | 38.5 | 29.1   |
> | SAM3 | 31.7     | 75.9     | 37.3    | 30.1        | 87.3     | 21.7  | 32.6       | 47.7 | 33.8    | 53.6 | **46.3**   |
>
> Overall, SAM3’s average performance of 46.3 is *significantly higher* than both CAT-Seg (37.5) and Grounded-SAM (29.1). SAM3 outperforms CAT-Seg in 15 datasets out of 21, and Grounded-SAM-L in 20/21.
> While performance could still be improved on expert domains (especially IR images and x-rays), these strong results showcase SAM3 versatility as a foundation model.

---

> ### Author Response · Authors · 2025-11-24
> **Response to reviewer ma9q [Part 4]**
>
> # Question 3: Crowd counting
> > The paper restricts its analysis to specific object counting benchmarks. This ignores the long-standing, highly challenging, and practical domain of crowd counting
>
> As a *generalist* foundation model, SAM 3 is not designed for extreme counting scenarios and crowd density estimation. However, to help understand its capabilities in more crowded scenarios, we conducted additional experiments on the standard FSC-147 dataset. We compare with a recent expert counting model, CountGD and follow its evaluation protocol.
> Note that as a generalist model SAM 3 only has 200 queries by default (to allow faster inference) which limits the number of objects that can be detected. It's typical for models designed for extreme counting scenarios to use many more queries. We therefore evaluate a version of SAM 3 equipped with 800 queries and find similar performance to CountGD, a counting expert model trained on FSC-147 train, in a *zero-shot* way.
>
> In addition, we finetuned SAM 3 on FSC-147 train and found it can handily outperform the specialist model, even with fewer queries. Note that SAM 3 is solving a *harder* task of *segmenting* target instances, not just counting them.
>
> | Model                | # object queries | MAE   |
> |----------------------|------------------|-------|
> | CountGD (finetuned)  | 900              | 14.8  |
> | SAM 3 (zero-shot)    | 800              | 16.8  |
> | SAM 3 (finetuned)    | 200              | 13.1 |
> | SAM 3 (finetuned)    | 1000             | **11.5** |
>
> ### **Thank you** for reading our responses! Please let us know if they addressed your questions.

---

### Official Review · Reviewer_N5F4 · 2025-10-30

**Soundness:** 4
**Presentation:** 3
**Contribution:** 2
**Rating:** 6
**Confidence:** 5

**Summary:**

This paper proposes SAM 3, an advanced segmentation foundation model aiming to perform “Prompted Concept Segmentation (PCS)” tasks, where text and/or image exemplars serve as prompts for concept understanding and object segmentation. The system integrates large-scale data, modular architecture design, and training techniques into a unified framework, showing strong segmentation performance across multiple benchmarks. The paper presents extensive dataset construction and evaluation.

**Strengths:**

1. Engineering Excellence and Practical Impact.
SAM 3 demonstrates strong engineering quality and productization potential, similar to recent systems such as DeepSeek. Its framework and modular tools are robust and well implemented, showing strong potential for deployment and for industrial or cross-domain applications.

2. Comprehensive Dataset and Benchmarking.
The data engine and dataset organization are well executed, standardized, systematic, and clear. The large-scale data collection and clearly defined evaluation pipeline demonstrate solid engineering effort.

3. High Performance through Scalable Architecture.
The model achieves impressive segmentation results through a reasonable architectural combination, large-scale data, and effective training strategies.

4. Potential for Broader Impact.
The SAM 3 Agent and its associated tools are solid and stable. With additional validation in medical, industrial, or molecular domains, this work could be even more suitable for a Nature-level publication due to its cross-domain applicability and engineering completeness.

**Weaknesses:**

1. Limited Novelty Beyond Existing Referring and Open-Vocabulary Segmentation Frameworks. While SAM 3 extends promptable segmentation to a broader “concept-level” scope, its core mechanism remains largely similar to existing referring and open-vocabulary segmentation approaches. The main novelty lies in system integration, data scaling, and multimodal prompting, rather than in algorithmic or conceptual innovation.
The paper’s strength is its engineering completeness and potential impact, but from a research novelty standpoint, its incremental contribution beyond prior referring segmentation frameworks (e.g., CLIPSeg (CVPR'22 [https://arxiv.org/abs/2112.10003]), SEEM (NeurIPS'23 [https://arxiv.org/pdf/2304.06718]), Grounded-SAM & Grounaded 2 (arXiv [https://arxiv.org/abs/2401.14159]) appears limited.
2. Lack of Theoretical Insight. The paper appears more like a large-scale project than a conceptual or methodological contribution. The performance improvements mainly come from scaling data and model size, rather than introducing new conceptual insights or algorithmic innovations.
3. Limited Definition and Discussion of “Concept”.
The notion of “concept” is underexplained. Since “concept” is an abstract idea, the paper should explicitly define it and clarify how PCS differs from existing works such as Spider (ICML'24 (https://arxiv.org/abs/2405.01002)) and SAM-Eva (arXiv (https://arxiv.org/abs/2412.01240)), which already distinguish between CI (context-independent) and CD (context-dependent) concepts. Without a clear definition, the term “Prompt Concept Segmentation” remains ambiguous. I suggest that the authors expand the Related Work section to provide a more detailed explanation of how “concept” has been defined and studied in prior literature. It would also be valuable to include additional experiments to demonstrate how SAM 3 performs across different types of concepts, particularly distinguishing between context-independent and context-dependent cases.
4. Task Limitation (PCS Inference Scope). The PCS task currently handles single-image or text-based prompts but lacks the ability for batch or generalizable reasoning across multiple images or complex prompts.
5. Lack of Novelty in Data Engine. While the data pipeline is well-organized, it is largely an engineering implementation without notable methodological innovation.
6. Questionable Data Efficiency.
In Table 14, the improvement from using 20% → 100% data is comparable to the smaller-scale increments (10% → 20%, etc.), suggesting poor data efficiency and underutilization of the large dataset. Moreover, there remains a significant gap between the full-data model and the teacher model, implying room for optimization in training or architecture.

**Questions:**

1.    SAM 3 presents itself as a large-scale, highly engineered system integrating architecture design, massive data curation, and multi-modality prompt handling. While its engineering quality and practical completeness are impressive, I am uncertain whether such a system-level project aligns with ICLR’s focus on methodological and theoretical innovation.

   * Would the authors consider that SAM 3, with its solid engineering and cross-domain applicability, might be more suitable for a Nature-type venue, where large, impactful engineering frameworks and cross-domain demonstrations are more appreciated?
   * Has the team considered including case studies or validations in diverse domains (e.g., medical imaging, industrial inspection, materials science, or bioinformatics) to better highlight the model’s broad real-world impact?


2.   I acknowledge the revolutionary impact of SAM (1), which fundamentally redefined segmentation as a promptable and interactive visual understanding task. SAM 2 further extended this to the temporal domain through hierarchical memory and video modeling.
   However, SAM 3 appears to be more of a task-level extension (image -> video -> concept) rather than a technical breakthrough.

   * The progression of input and output forms (point or box to frame sequence to text or image exemplars; single-frame masks to temporal sequences to concept-level masks) seems evolutionary rather than fundamentally new.
   * It is difficult to evaluate SAM 3’s novelty in isolation from SAM (1) and SAM (2). Its advances appear to rely heavily on the established SAM framework and infrastructure.
   * If the entire SAM series (1, 2, and 3) were viewed as a single long-term contribution, I would rate it extremely high, possibly a 10/10. For SAM 3 alone, however, the incremental contribution ratio remains unclear. Could the authors clarify what specific conceptual or architectural innovations distinguish SAM 3 from its predecessors beyond scaling and multi-modality integration?

---

> ### Author Response · Authors · 2025-11-24
> **Response to N5F4 [Part 1]**
>
> Thank you for the detailed feedback and questions!
>
> # Question 2: Clarify conceptual novelty
> > improvements mainly come from scaling data and model size, rather than introducing new conceptual insights or algorithmic innovations…. Could the authors clarify what specific conceptual or architectural innovations distinguish SAM 3 from its predecessors beyond scaling and multi-modality integration?
>
> > “from a research novelty standpoint, its incremental contribution beyond prior referring segmentation frameworks (e.g., CLIPSeg (CVPR'22 [https://arxiv.org/abs/2112.10003]), SEEM (NeurIPS'23 [https://arxiv.org/pdf/2304.06718]), Grounded-SAM & Grounaded 2 (arXiv [https://arxiv.org/abs/2401.14159]) appears limited.”
>
> Our paper makes several key contributions:
> * **New metrics and benchmarks** to drastically expand the vocabulary in open-vocabulary detection and segmentation
> * **A novel data engine** to vastly scale high-quality, exhaustively annotated training data
> * **A novel architecture** to support visual, text, and exemplar prompts in one model (the first to do so)
> * **A major leap** in performance on open-vocabulary instance segmentation, **double** that of existing models
>
> The **key conceptual insight** that enabled the performance leap is the *decoupling* of *recognition* (whether the concept is present in the image) and *localization* (where the concept is in the image) for open-vocabulary detection/segmentation. While prior work conflates these two tasks and thus struggles to improve performance, our analysis reveals that the primary bottleneck in open-vocabulary applications is actually recognition (e.g., confusing a crab with a lobster) rather than localization.
>
> This conceptual insight of decoupling recognition and localization plays a crucial role in all aspects of SAM 3:
>
> * **Metric (novel evaluation)**: We design a new metric, classification gated F1 (cgF1), to separately measure the recognition and localization abilities of the model. We evaluate localization using positive macro F1 (pmF1) on positive media-phrase pairs with at least one ground-truth mask. Recognition is measured with image-level Matthews Correlation Coefficient (IL_MCC) which evaluates binary prediction at the image level (``is the concept present?''). The final metric is `cgF1 = 100 x pmF1 x IL_MCC`. Table 32 clearly shows that the main gap between previous SoTA and human performance is recognition (IL_MCC: OWLv2* 0.58 vs human 0.88), instead of localization (pmF1: DINO-X 71.8 vs human 84.6). Table 14 also reveals this gap: after training with only external datasets, SAM 3’s recognition is only 50% of human IL_MCC, while localization is already at 80% of human pmF1.
>
> * **Data (novel methodology)**: To address the recognition gap (“crab vs lobster” confusion), we annotated 130M hard negative image-phrase pairs in SA-Co/HQ and synthetically generated 1.26B hard negative image-phrase pairs in SA-Co/SYN as training data. Annotating such large-scale *hard negative* data has never been done before, and required two specific research innovations: (a) constructing a visual concept ontology from Wikidata to sample confusing negatives, and (b) creating an automated verifier which achieves human-level performance (Table 25) in filtering these annotations, by fine-tuning Llama 3.2.
>
> * **Detector (novel architecture and training recipe)**: To solve the recognition gap in the model, we propose a novel *presence token* for the recognition task, and traditional object queries for the localization task (traditional DETR methods conflate recognition and localization in a single query). Crucially, we decouple their training losses: the presence token is trained with the image-level binary classification task, and the object tokens only receive training signals from positive image-concept pairs. We show that this decoupled model architecture and loss are critical in Table 9(a) and Table 10, respectively.
>
>
> With these innovations in the metric, data and model, SAM 3 achieves a step change in recognition, approaching human performance, as shown in Table 32 (IL_MCC: OWLv2* 0.58 vs human 0.88 vs SAM 3 0.82), and a significant improvement in localization (pmF1: DINO-X 71.8 vs human 84.6 vs SAM 3 79.7). As far as we know, the key conceptual insight of decoupling recognition and localization and the techniques we propose to achieve this are all novel research contributions.

---

> ### Author Response · Authors · 2025-11-24
> **Response to N5F4 [Part 2]**
>
> # Question 2: Clarify conceptual novelty (cont-d)
>
> ## Novel Detector-Tracker Integration
>
> This is another contribution of our paper. Methods that chain SAM 2 video segmentation with object detection models (e.g., GroundedSAM2 mentioned by Reviewer N5F4) blindly trust per-frame detections, leading to error propagation: Each detector’s false positive (e.g., due to occlusion or blur) will lead to a false positive masklet. Our approach overcomes this by building a masklet detection confidence score across the entire object trajectory.
>
> While our detector's output may fluctuate or hallucinate on isolated frames due to occlusion or blur, its average confidence across a coherent video sequence provides a highly robust signal. This allows SAM 3 to filter noisy detections effectively, achieving high precision without sacrificing recall, as shown in Table 38 (“SAM3 w/o any temporal disambiguation” vs “SAM3”).
>
> To further validate this, we build a strong baseline following GroundedSAM2’s approach on top of SAM3’s detector and tracker. Note that this is a *strictly stronger system* since GroundedSAM2 is a combination of Grounding DINO with SAM 2 (and both of these detectors and trackers are outperformed by the SAM 3 variants as shown in Table 1 and 6 in the paper).
>
> We show a systematic ablation (identical training data, detector and tracker components) of GroundedSAM2’s approach compared to our video masklet inference strategy in Table A below. The results clearly indicate that SAM 3’s approach for masklet integration outperforms GroundedSAM2’s approach.
>
> **Table A**: GroundedSAM2* vs SAM 3 to show the importance of our novel detection-tracking synergy algorithm.
>
> |  | SA-V test |  |  |  | YT-Temporal-1B test |  |  |  | SmartGlasses test |  |  |  |
> | :---- | ----- | ----- | ----- | ----- | ----- | ----- | ----- | ----- | ----- | ----- | ----- | ----- |
> | Model | F1 | Precision | Recall | TETA | F1 | Precision | Recall | TETA | F1 | Precision | Recall | TETA |
> | GroundedSAM2* | 17.3 | 10.1 | 42.0 | 54.5 | 41.5 | 32.5 | 57.2 | 71.7 | 19.0 | 12.8 | 37.0 | 56.7 |
> | SAM 3 | 38.6 | 34.7 | 43.6 | 56.8 | 55.6 | 52.6 | 59.0 | 70.5 | 43.8 | 39.8 | 48.6 | 65.9 |
>
> Note:  F1 denotes F1@conf >= 0.5.
> *We follow GroundedSAM2’s approach (https://github.com/IDEA-Research/Grounded-SAM-2) of combining SAM3’s detector and tracker components.
>
> ## Novelty Beyond Existing Open-Vocabulary Segmentation Frameworks
> > While SAM 3 extends promptable segmentation to a broader “concept-level” scope, its core mechanism remains largely similar to existing … open-vocabulary segmentation approaches.
>
> As described above, the core approach behind SAM 3–decoupling recognition and localization in the model, metric and data–is novel and has not been implemented in any existing framework. To *“extend promptable segmentation to a broader ‘concept-level’ scope”* is highly non-trivial and has not been achieved by prior methods including those that scale up data. SAM 3 represents a ground-breaking improvement in open-vocabulary instance segmentation in images and videos, going far beyond existing models’ capabilities. Our performance is *2 times, 2.5 times, and 3 times* better than the best baseline (OWLv2*) on the SA-CO Gold, Bronze and Silver datasets, respectively, and *5.8 times* better than the best baseline (Gemini) on SA-CO Bio. Moreover, SAM 3 achieves above 80% of human performance on the SA-Co Gold and Video eval splits, demonstrating that it comes close to “just working” on this task.
>
> SAM 3 can handle a *vastly* larger set of open-vocabulary prompts than prior work. Existing segmentation benchmarks have a maximum of 1K unique open-vocabulary prompts, so we created the new SA-CO benchmark to push the model to handle more concepts. Our SA-Co benchmark has *~220x more unique concepts* than existing benchmarks, all exhaustively annotated with multiple instance masks. It also includes many hard negative prompts to evaluate the models’ ability to do open vocabulary recognition (lobster vs crab).

---

> ### Author Response · Authors · 2025-11-24
> **Response to N5F4  [Part 3]**
>
> ## Novelty beyond SAM 1&2
> > If the entire SAM series (1, 2, and 3) were viewed as a single long-term contribution, I would rate it extremely high, possibly a 10/10. For SAM 3 alone, however, the incremental contribution ratio remains unclear.
> > The progression of input and output forms (point or box to frame sequence to text or image exemplars; single-frame masks to temporal sequences to concept-level masks) seems evolutionary rather than fundamentally new.
>
> SAM 1&2 advanced the Promptable *Visual* Segmentation (PVS) task – using points, boxes or masks to *visually* segment a single object per prompt. They did not address the task of *linguistic* open-vocabulary object detection and instance segmentation. For example, to segment and track all cars in a video, SAM 2 requires manually clicking on each car, while SAM 3 just needs a single prompt “car”. SAM 3 not only introduces this new ability to the SAM series, it advances the state-of-the-art on this task to a ground-breaking new level.
>
> We do not claim the PCS task is completely new in the literature. We formalize it as an interactive open-vocabulary instance segmentation task with noun phrase and exemplar prompts. Nevertheless, please note that the PVS task was also not invented by SAM 1. SAM 1 advanced the SOTA on this task to a level where it became practically useful. SAM 3 does the same for open-vocabulary concept segmentation.
>
> Furthermore, the architecture of SAM 3 innovates on SAM 1&2 in several ways:
> * adds a new image-based detector to the SAM 2-based tracker to enable concept prompts (text or exemplars)
> * designs the detector architecture to better support open vocabulary tasks
> * combines the tracker and detector in an effective way
>
> ## Novelty Beyond Existing Referring Segmentation Frameworks
> > from a research novelty standpoint, its incremental contribution beyond prior referring segmentation frameworks (e.g., CLIPSeg (CVPR'22), SEEM (NeurIPS'23 ), Grounded-SAM & Grounded SAM2 appears limited.
>
> We want to clarify that SAM 3 is not a referring segmentation model, and should not be compared directly to methods aimed solely at referring segmentation. SAM 3 is designed to exhaustively detect general object categories, a task we call promptable concept segmentation (PCS). These tasks are very different:
> * Referring expression segmentation (RES)
>    * The goal is to identify a *single* foreground object that matches a text prompt.
>   * The text prompt describes a specific region in the scene, such as “the child sitting in the adult’s lap”
>   * The output is a binary foreground/background segmentation map.
> * Promptable concept segmentation (PCS)
>   * The goal is to segment *many* instances of the concept provided via text prompt, and do so exhaustively
>   * The text prompts are short noun phrases that are category-level, e.g. “children,” “adults”.  Exemplars can also be used as prompts.
>   * The output is *multiple* instance segments with unique IDs. In video, the IDs must be correctly tracked across frames (multi-object tracking).
>
> Models trained on RES do not work well for PCS, and vice versa. Some models are trained on both RES and open-vocab instance segmentation tasks (e.g. SEEM, Grounded SAM), but in our experiments they fall far short of SAM 3 on PCS. As described earlier, the SOTA approaches suffer from the recognition gap for open-vocabulary detection/segmentation, which we solve with SAM 3.

---

> ### Author Response · Authors · 2025-11-24
> **Response to N5F4 [Part 4]**
>
> ## Data engine novelty
> > Lack of Novelty in Data Engine. While the data pipeline is well-organized, it is largely an engineering implementation without notable methodological innovation.
>
> To the best of our knowledge, our data engine is 1) *novel* and, much more importantly, 2) *highly effective* and 3) *efficient*.
>
> 1) **Novel**. Previous approaches (e.g. DINO-X, Grounding DINO 1.5, 1.6) in this space that used a data engine to scale up annotations have sophisticated pipelines but share very little detail, while others such as OWLv2 have relied on simple parsing of alt-text associated with images together with a grounding model for localization. To create SAM 3 we built a novel pipeline that includes custom AI mask and exhaustivity verifiers, captioners, parsers, an ontology and hard negative proposers. If the reviewer can share references to comparable data engines, we are happy to include them in our discussion.
>
> 2) **Highly Effective**. Beyond novelty, our data engine is highly effective. We ran additional experiments to demonstrate this. We generated several versions of training annotations for a domain not seen in training (“food & drink”), trained SAM 3 models on these datasets, and evaluated them on the test set.
> |    training data  | cgF1 on SA-Co|
> |-------------------|--------------|
> |a) our data engine | 65 |
> |b) no AI verifiers | 50 |
> |c) no ontology/negatives | 47 |
> |d) baseline/init | 45 |
>
> We find using (a) our full data engine with AI mask and exhaustivity verifiers, our ontology and hard negative proposer (65 cgF1) is *dramatically* better than (b) leveraging our ontology and negative proposer alone (50 cgF1), while (c) a baseline not using our ontology or negative proposer performs poorly (47 cgF1), little better than the baseline/initialization (45 cgF1). Baseline (c) is comparable to OWLv2’s data engine. *Our data engine produces dramatically better performance*. In Figure 7a (appendix B.3), we demonstrate *robust scaling* behaviour of our data engine, providing high quality annotations that enable performance on-par with *human-labelled annotations*.
>
> 3) **Efficient**. Our AI annotators are *5x faster* than humans on negative prompts (concepts not present in the image/video) and *36% faster* for positive prompts, even in challenging fine-grained domains. (see appendix B.4)
>
> # Definition and Discussion of “Concept”.
> > Since “concept” is an abstract idea, the paper should explicitly define it and clarify how PCS differs from existing works ...which already distinguish between CI (context-independent) and CD (context-dependent) concepts.
>
> Thank you for the question. A “concept” in our work is defined as any visual concept that can be described with a simple noun phrase (NP), such as “yellow school bus,” “cloudy sky,” “a child” (p.1 line 38). We use the term “concept” rather than “object” to indicate semantic categories and distinguish it from visual object segmentation. Practically, our set of concepts is defined by the way we generate them for training: either by parsing image captions into simple noun phrases, or by using the names of Wiki nodes from our ontology (see Table 20 and Fig. 12).
>
> Thanks for pointing out the papers that further distinguish between CI (context-independent) and CD (context-dependent) concepts. We do not make this distinction in our work, nor benchmark such concepts separately, but it is indeed an interesting distinction and possible future follow up evaluation. Anecdotally, SAM 3 can detect shadows and reflections quite well.

---

> ### Author Response · Authors · 2025-11-24
> **Response to N5F4 [Part 5]**
>
> # Question 1a: Suitability for ICLR
> >SAM 3 presents itself as a large-scale, highly engineered system integrating architecture design, massive data curation, and multi-modality prompt handling. While its engineering quality and practical completeness are impressive, I am uncertain whether such a system-level project aligns with ICLR’s focus on methodological and theoretical innovation.
>
> >Would the authors consider that SAM 3, with its solid engineering and cross-domain applicability, might be more suitable for a Nature-type venue, where large, impactful engineering frameworks and cross-domain demonstrations are more appreciated?
>
> We respectfully believe that ICLR, along with similar conferences such as NeurIPS, ICML, and CVPR, is the most appropriate venue for SAM 3. In contrast, Nature may not reach the intended audience for this work. Previous influential projects with significant data contributions, and with less emphasis on architectural or algorithmic novelty have been successfully published and recognized at these conferences. Notable examples include CLIP (Radford et al., ICML’21), GPT-3 (Brown et al., NeurIPS’21, Outstanding Paper), and SAM 1 & 2 (ICCV’23, ICLR’25, both receiving Outstanding Paper Honorable Mentions). These precedents suggest that contributions like SAM 3 are well aligned with the scope and interests of ICLR and related venues.
>
> >I acknowledge the revolutionary impact of SAM (1), which fundamentally redefined segmentation as a promptable and interactive visual understanding task. SAM 2 further extended this to the temporal domain through hierarchical memory and video modeling. However, SAM 3 appears to be more of a task-level extension (image -> video -> concept) rather than a technical breakthrough.
>
> Our work makes a major contribution to the field of machine learning by developing a new foundational model for object-aware visual perception. Most foundational perception models today (e.g. Gemini 2.5) have poor object understanding: they struggle to detect and localize open-vocabulary object categories and generate pixel-accurate masks in images or videos. Foundation models specialized for segmentation also struggle with accurate open-vocabulary instance segmentation, as we show in our experiments on a large-vocabulary benchmark  (see Table 1, e.g. OWLv2+SAM2, Tab. 5, GLEE).
>
> The significance of our technical breakthrough is comparable to that of SAM 1 and SAM 2 (the latter is published in ICLR and highly cited). SAM 1 similarly introduced a very powerful promptable segmentation model. Note that SAM 1 was NOT the first to propose the interactive visual segmentation task. This was already explored a year earlier:
>
> * Konstantin Sofiiuk, Ilya A Petrov, and Anton Konushin. Reviving iterative training with mask guidance for interactive segmentation. ICIP, 2022, https://arxiv.org/pdf/2102.06583
>
> What SAM 1 did was achieve a breakthrough in performance on this task, via a large-scale data collection engine. The amazing thing about SAM 1 is that it generalizes so well, it “just works”. SAM 3 achieves this kind of technical breakthrough for open-vocabulary concept detection/segmentation.

---

> ### Author Response · Authors · 2025-11-24
> **Response to N5F4 [Part 6]**
>
> # Question 1b: Case studies on real-world domains
> > Has the team considered including case studies or validations in diverse domains (e.g., medical imaging, industrial inspection, materials science, or bioinformatics) to better highlight the model’s broad real-world impact?
>
> In the paper we train and evaluate SAM 3 on **29 diverse domains** contained in our SA-Co evaluation benchmark, including:
> _Driving - BDD100k,
> Robotics - DROID,
> Egocentric - Ego4D,
> Food recognition - MyFoodRepo-273,
> Animals - iNaturalist-2017,
> Art - National Gallery of Art,
> Sea animals - Fathomnet,
> Fashion - Fashionpedia,
> Biomedical - Livecell, PanNuke, MedSAM2, SNOW,_
> and many more.
>
> SAM 3 improves results on all of these domains relative to existing models, see Table 1. For example, on the biomedical subset, SA-Co/Bio, SAM 3 beats the best baseline with pmF1=55.4 vs 10.7. We also evaluate zero-shot performance on ODinW13 and RF-100VL which total 113 real-world datasets including industrial applications, see Table 2.
>
> Below we provide an additional case study evaluating SAM 3 zero-shot on a real-world use case, a multi-animal tracking dataset called SA-FARI. The dataset contains 11,609 camera trap videos collected over 10 years (2014-2024) from 741 locations, spanning 99 species categories. We evaluate on the test set, which contains 833 videos with a total duration of 202 minutes and 83 species categories (spider monkey, sloth, agouti, etc.) associated with 1,083 unique masklets.
>
> We compare two baselines, GLEE (Wu et al., 2024a) and LLMDet (Fu et al., 2025) combined with SAM 3’s tracker. All models are prompted with the species category and asked to detect and segment all instances of that species throughout the video. We also fine-tune SAM 3 on SA-FARI and report cgF1, pHOTA and TETA metrics.
>
> | Method                       | cgF₁ | pHOTA Total | pHOTA Det | pHOTA Ass | TETA |
> |------------------------------|------|--------------|------------|------------|-------|
> | **1 GLEE**                   | -0.2 | 7.5          | 1.2        | 49.7       | 22.0  |
> | **2 LLMDet + SAM 3 Tracker** | 2.6  | 41.3         | 21.4       | 80.0       | 30.4  |
> | **3 SAM 3**                  | 14.0 | 48.5         | 28.4       | 83.4       | 39.6  |
> | **5 SAM 3 FT (SA-FARI)**     | **46.9** | **68.1** | **55.4** | **84.6** | **58.7** |
>
> Camera traps are a notoriously difficult detection domain. The results show that SAM 3 has strong zero-shot performance on this dataset, significantly improving on baselines. With fine-tuning, it reaches even better performance, demonstrating its value as a foundational model that can be tuned to specialized domains.
>
> # Questionable Data Efficiency in Table 14
> > In Table 14, the improvement from using 20% → 100% data is comparable to the smaller-scale increments (10% → 20%, etc.), suggesting poor data efficiency and underutilization of the large dataset.
>
> **“Data Inefficiency Issue”**: It is fairly typical for data scaling laws to be *power*-laws [1, 2], so it’s quite *expected* that gains from 20% → 100% are comparable to 10% → 20%, as performance gains are expected to be linear in **log** of #samples. While we agree that it would be nice to have higher data efficiency, this is a much broader and fundamental question for the field of AI [3] and beyond scope for SAM 3 and as such is not a particular weakness of SAM 3.
>
> * Kaplan et al. (2020), Scaling Laws for Neural Language Models.
> * Hoffmann et al. (2022), Training Compute-Optimal Large Language Models.
> * Bahri et al. (2024), Explaining Neural Scaling Laws.
>
> **“Gap to Teacher”**:
> > Questionable Data Efficiency. In Table 14, the improvement from using 20% → 100% data is comparable to the smaller-scale increments (10% → 20%, etc.), suggesting poor data efficiency and underutilization of the large dataset. Moreover, there remains a significant gap between the full-data model and the teacher model, implying room for optimization in training or architecture.
>
> While there is indeed a gap to teacher performance in Table 14, it’s important to note that this is a shorter, *ablation setting* and the gap to teacher performance is significantly lower for the *full model* (Table 1) which reaches 88% of lower bound of teacher (human) performance. Importantly, the “teacher” in this context is **human-level** performance - respectfully, not matching human performance as a weakness would be an extraordinarily high expectation for any submission; on the contrary we are very encouraged that SAM 3 pushes performance to the extent that human-level performance is not far off!
>
> ### **Thank you** for reading our responses! Please let us know if they address your concerns.

---

### Official Review · Reviewer_EmE5 · 2025-11-01

**Soundness:** 3
**Presentation:** 3
**Contribution:** 3
**Rating:** 8
**Confidence:** 4

**Summary:**

This paper introduces SAM3, a unified model that can perform detection on images and tracking across frames based on given concepts or prompts. The main contributions are twofold: proposing the SAM3 method and introducing a data engine pipeline. Unlike SAM2, it can detect multiple objects based on given concepts, extending its capabilities. Furthermore, by incorporating MLLMs, it produces more precise and higher quality datasets. The results demonstrate strong generalization ability across various scenarios.

**Strengths:**

- Exhaustive Analysis
  - The paper demonstrates its effectiveness through extensive analyses provided in the appendix.
  - Each component is evaluated with ablation studies, and the paper even illustrates the impact of AI verification through various experiments.
  - For both image and video datasets, the importance of each and the effects of different dataset sizes are thoroughly explored.
- Open-sourcing
  - The paper open sources key components, including SAM3 and the SA-Co benchmark. In particular, releasing the SA-Co benchmark contributes to the perception research community by providing challenging and diverse samples.

**Weaknesses:**

- Unclear Definition of Terms
  - Throughout the paper, the term **geometric** is frequently used. What is its exact meaning here? At times, *geometric* is distinguished from *visual prompts*, but in other cases, it seems to include their meaning. Although it represents an important concept in the paper, its usage is quite vague.

**Questions:**

- Motion-aware Concept
  - In SAM3, the main focus is on simple noun phrases. For more complex or longer phrases, the paper demonstrates the use of MLLMs to handle such cases. However, in video segmentation, object descriptions often incorporate motion-aware information. In such cases, which approach could be used? Could the SAM3 Agent design still be applied to such scenarios? The current design seems to use the tracking module only to associate objects across frames in a semantic-agnostic manner.

---

> ### Author Response · Authors · 2025-11-24
> **response to reviewer EmE5**
>
> Thank you for your feedback and questions!
>
> > Throughout the paper, the term geometric is frequently used. What is its exact meaning here? At times, geometric is distinguished from visual prompts, but in other cases, it seems to include their meaning. Although it represents an important concept in the paper, its usage is quite vague.
>
> The terms “visual prompts” and “geometric prompts” refer to the same thing: point, box or mask prompts used in visual segmentation. Thank you for pointing out the possible confusion, we will update the paper to use the term “visual prompt” consistently.
>
> >In SAM3, the main focus is on simple noun phrases. For more complex or longer phrases, the paper demonstrates the use of MLLMs to handle such cases. However, in video segmentation, object descriptions often incorporate motion-aware information. In such cases, which approach could be used? Could the SAM3 Agent design still be applied to such scenarios?
>
> Our focus in this work is on simple noun phrases that can be observed in single frames - as we show in this work, even this simpler task is *very challenging* and far from being solved and SAM 3 enables a step-change that can provide a *robust foundation* to solve more complex tasks/queries (e.g. based on motion-cues, “person getting up vs. sitting down").
> We demonstrated an example of such a workflow with SAM 3 Agent for images. In a similar spirit, we expect a Video LLM can use SAM 3 as a tool, to detect and track a person, and then use its own processing to infer the high-level information from motion cues.

---

### Public Comment · ~Siyang_Li1 · 2025-11-14
**VOS task results**

Hi authors,

I've been following the SAM series for a while.
Compared with SAM2, this paper shifts the focus from videos back to images. However, I still have one question about the evaluation on video tasks:

In Table 19 (appendix B.5), with 100% SACO training videos, the model gets DAVIS 91.5 (-0.1); SAV-val 77.4 (+0.4), SAV-test 78.0 (+0.9), which means the additional training videos only helped a little bit on VOS tasks (comparing with only use SAM2 video data).  While in Table 6, SAM3 gets DAVIS 92.0, SAV-val 82.0, SAV-test 84.6. Could you please explain where the improvement, especially the 4%+ on SAV is from? And what's the difference between the models for Table 6 and Table 19?

Thanks very much for your attention! Looking forward to your reply.

---

> ### Author Response · Authors · 2025-11-25
> **Table 19 is a simpler ablation setting**
>
> Thank you for your interest!
>
> The results in Table 19 use an *ablation* setting to show the impact of domain expansion by adding new VOS data. Differences compared to the model used in Table 6 include:
>
> 1. In Table 19, the backbone was pretrained with fewer VOS datasets (frames used for "image" training task). Similarly, for video training, we don’t include the full training set (e.g. MOSEv2 was not included in these ablation runs).
> 2. The model in Table 19 is trained with a much shorter training schedule.
> 3. Memory filtering was not used in Table 19 evaluations.

---

### Meta-Review · Area_Chair_9Gho · 2025-12-25

**Summary:**

The submission presents SAM 3, a unified framework for open-vocabulary concept segmentation and tracking in images and videos. The authors introduce a scalable data engine, a new benchmark (SA-Co), and a model architecture that notably decouples recognition from localization using a "presence head." The reviewers unanimously recognized the strong empirical performance, the engineering excellence of the data pipeline, and the value of the open-sourced benchmark. While the consensus is positive, there is a divergence regarding the depth of the methodological contribution, with some reviewers viewing the work as a robust system-level integration rather than a fundamental algorithmic breakthrough.

**Reviewer Concerns:**

The authors provided a comprehensive rebuttal that effectively addressed the empirical inquiries, particularly regarding robustness to prompt perturbations and the inclusion of semantic segmentation benchmarks like MESS.

However, regarding the methodological contribution, I concur with the reservations raised during the review process. The core architectural innovation of decoupling recognition from localization via a "presence head" is more like a heuristic engineering trick rather than a fundamental theoretical breakthrough. While this design is highly effective for mitigating hallucinations in open-vocabulary settings, it represents a practical workaround ("patching" the recognition gap) rather than a shift in representation learning paradigms. Similarly, the "Promptable Concept Segmentation" task is a logical, evolutionary extension of the SAM framework.

Therefore, the paper's primary value lies in its system-level synthesis, data engine scale, and the resulting community resource, rather than in algorithmic novelty. This profile still supports an acceptance.

**Reviewer Scores:**

The scores are expected to stabilize around the current consensus (two 8s, two 6s). Reviewers EmE5 and wjcv (Scores: 8) will likely maintain their support, valuing the data engine and performance scale. Reviewers N5F4 and ma9q (Scores: 6) are expected to maintain their scores as well; their reservations regarding the "incremental" nature of the innovation are valid conceptual points that the rebuttal (which focused on empirics) cannot fundamentally change. The split reflects a healthy debate on "Utility vs. Novelty," with the decision landing on Accept based on the undeniable utility.

---

### Decision · Program_Chairs · 2026-01-26

Accept (Poster)